# The Good, the Bad and the Ugly: Meta-Analysis of Watermarks, Transferable Attacks and Adversarial Defenses

**Grzegorz Głuch**
UC Berkeley
gluch@berkeley.edu

**Berkant Turan**
Zuse Institute Berlin
turan@zib.de

**Sai Ganesh Nagarajan**
University of Southern Denmark
sgnagarajan@imada.sdu.dk

**Sebastian Pokutta**
Zuse Institute Berlin and TU Berlin
pokutta@zib.de

## Abstract

We formalize and analyze the trade-off between backdoor-based watermarks and adversarial defenses, framing it as an interactive protocol between a verifier and a prover. While previous works have primarily focused on this trade-off, our analysis extends it by identifying transferable attacks as a third, counterintuitive, but necessary option. Our main result shows that for all learning tasks, at least one of the three exists: a *watermark*, an *adversarial defense*, or a *transferable attack*. By transferable attack, we refer to an efficient algorithm that generates queries indistinguishable from the data distribution and capable of fooling *all* efficient defenders. Using cryptographic techniques, specifically fully homomorphic encryption, we construct a transferable attack and prove its necessity in this trade-off. Finally, we show that tasks of bounded VC-dimension allow adversarial defenses against all attackers, while a subclass allows watermarks secure against fast adversaries.

## 1 Introduction

Backdoor attacks and adversarial robustness are closely related: the former embeds hidden behaviors via subtle input changes, while the latter seeks to ensure stable predictions against worst-case input modifications. Recent works [Weng et al., 2020, Sun et al., 2020, Niu et al., 2024, Fowl et al., 2021, Tao et al., 2024] have empirically explored the trade-offs between adversarial robustness and backdoor attacks. Their general observation is that "models that are made to be robust against certain types of adversarial attacks may become more vulnerable to backdoor attacks."

Outside classic methods such as adversarial training Madry et al. [2018], which apply generally, provable defenses against general adversaries are known only for restricted function classes—particularly when the defense focuses on detecting, rather than classifying, attacks. Provided the attacks are not *indistinguishable* from the data distribution, rejection-based methods can effectively defend against arbitrarily crafted adversarial examples (beyond $\ell_p$-norm perturbations) Goldwasser et al. [2020]. Other studies show that backdoors can be planted using cryptographic schemes Goldwasser et al. [2022], making their detection computationally intractable. Conversely, standard post-hoc defenses such as randomized smoothing may inadvertently remove such backdoors Cohen et al. [2019].

Formally analyzing these trade-offs must account for the full spectrum of strategies available to attackers and defenders—each with potentially different computational capacities and resources Bubeck et al. [2019], Garg et al. [2020]—which presents a significant challenge. Recently, Christiano et al. [2024] attempted to characterize function classes for which one can provably defend against

39th Conference on Neural Information Processing Systems (NeurIPS 2025).

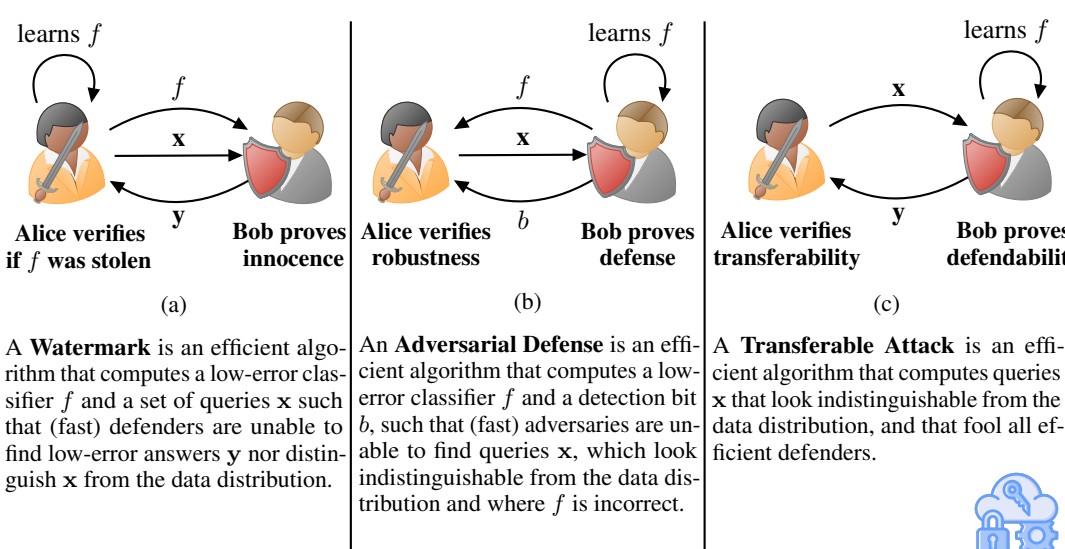

A **Watermark** is an efficient algorithm that computes a low-error classifier $f$ and a set of queries $\mathbf{x}$ such that (fast) defenders are unable to find low-error answers $\mathbf{y}$ nor distinguish $\mathbf{x}$ from the data distribution.

An **Adversarial Defense** is an efficient algorithm that computes a low-error classifier $f$ and a detection bit $b$, such that (fast) adversaries are unable to find queries $\mathbf{x}$, which look indistinguishable from the data distribution and where $f$ is incorrect.

A **Transferable Attack** is an efficient algorithm that computes queries $\mathbf{x}$ that look indistinguishable from the data distribution, and that fool all efficient defenders.

Figure 1: Schematic overview of the interaction structure, along with short, informal versions of our definitions of (a) Watermark (Definition 3), (b) Adversarial Defense (Definition 4), and (c) Transferable Attack (Definition 5), with (c) tied to cryptography (see Section 6).

backdoor attacks: they demonstrate that classes with bounded VC-dimension admit defenses, and they construct classes for which designing a defense is computationally infeasible.

Finally, studying this trade-off is crucial because backdoor attacks can also serve as watermarks in black-box settings Adi et al. [2018], Zhang et al. [2018], Namba and Sakuma [2019]. Understanding their interplay informs us of the limitations and applicability of both defenses and backdoor-based watermarks across different learning tasks. This paper formalizes these notions and undertakes a meta-analysis of them. In doing so, it led us to identify a third scheme—the *transferable attack*, which is an attack that is *indistinguishable* from the data distribution and can fool all models trainable within given resource constraints.

## 1.1 Our Contributions

We study classifcation learning tasks and our main result shows that:

*For every learning task, at least one of the three must exist:*
*A Watermark, an Adversarial Defense, or a Transferable Attack.*

To prove this, we formalize and extend existing definitions of watermarks and adversarial defenses as an interactive protocol between two players—Alice and Bob, (see Figure 1) [Goldwasser and Sipser, 1986]. This protocol always has at least one winner—either Alice can embed an unremovable watermark, Bob can construct a strong adversarial defense, or a third option emerges: a transferable attack.

**Transferable Attack.** To understand transferable attacks, consider the following game. Alice interacts with a player who claims to have a secure model for a learning task $\mathcal{D}, h$, where $\mathcal{D}$ is the data distribution and $h$ is the ground truth. Alice sends queries and observes the responses. She wins if she can generate queries that (i) cause significant errors and (ii) remain indistinguishable from samples drawn from $\mathcal{D}$.[1] Whether she succeeds depends on the computational and data resources available to her and the other player. If Alice can defeat *any* equally-resourced player, we call her queries a *Transferable Attack*. Intuitively, the more challenging a query becomes, the easier it should be to detect—but surprisingly, we show that transferable attacks do exist. Specifically, we prove:

---

[1]We note that what we consider a Transferable Attack is slightly nonstandard - there is no explicit model the attacks on which we consider transferability of. However, we can think that Alice first trains a model, then tries to find adversarial examples for it, and sends those as the queries in the game.

- The existence of a **Transferable Attacks** as defined above. Our construction uses cryptographic techniques, particularly Fully Homomorphic Encryption (FHE) [Gentry, 2009]. This establishes that Transferable Attacks form the third fundamental option in the trade-off.

- That any learning task supporting a Transferable Attack must be computationally complex. More precisely, Transferable Attacks imply the existence of a *cryptographic primitive*.

Notably, Tramèr et al. [2017], suggest a conjecture for the transferability of adversarial attacks: *If two models achieve low error for some task while also exhibiting low robustness to adversarial examples, adversarial examples crafted on one model transfer to the other.* However, they qualify their hypothesis by showing that it is not true in general, but only when the models are of the same class- thus complicating the picture. Our meta-analysis shows that whether the conjecture holds depends crucially on the **computational-resources** available to the attacker and defender. We argue that foregrounding computational resources in the problem of robustness clarifies the landscape considerably.

**Constructions.** We show that the existence of these properties does not depend on any particular algorithm or a model that is used. It depends on the learning task at hand and the computational resources for Alice and Bob. We give examples of learning tasks that provably support Watermarks, Adversarial Defenses and Transferable Attacks thereby justifying our framework. Concretely: (1) The construction of a learning task with a **Transferable Attack**, where the attacker needs strictly *fewer* resources than the defender. (2) We show that learning tasks with bounded VC dimension allow **Adversarial Defenses** against all (even computationally unbounded) attackers, ruling out Transferable Attacks in these settings. (3) We construct a **Watermark** for a class of learning tasks with bounded VC-dimension. Interestingly, in this case, both a Watermark and an Adversarial Defense coexist. Overall, these examples reiterate that the dependence on resources and the learning task are crucial.

**Resource Allocation Implications.** Our theorem can provide a rule of thumb for defenders. If an adversary has computational budget $T$ (e.g., time), then allocating $T^2$ computation on the defender's side suffices (up to constant factors) to construct a defense whenever one exists under our assumptions. Conversely, if a $T^2$-budgeted procedure fails, this provides evidence that the instance admits *transferable attacks*, which in-turn precludes watermarks in our framework. To our knowledge, this is among the first quantitative attacker–defender resource trade-offs stated in a model-agnostic setting, i.e., beyond capacity-bounded regimes (e.g., finite VC dimension).

## 2 Related Work

While most trade-offs between backdoor-based attacks and adversarial defenses have been studied empirically, Pal and Vidal [2020] show (theoretically) that *Fast Gradient Methods* (attacks) and *Randomized Smoothing* (defenses) can form a Nash equilibrium under a restricted additive-noise model. They also provide experiments confirming this on datasets such as MNIST.[2] Our theoretical results generalize their findings to a broader class of attacks and defenses.

### 2.1 Adversarial Robustness

Adversarial robustness research includes techniques like adversarial training [Madry et al., 2018], which improves resilience via adversarial examples, and certified defenses [Raghunathan et al., 2018], which provide provable guarantees within perturbation bounds. Methods such as randomized smoothing [Cohen et al., 2019] extend these guarantees, but mainly as a defense against $\ell_p$ norm perturbations. Moving beyond this, the work of Goldwasser et al. [2020] establish provable and computationally efficient defenses against arbitrary adversarial examples by detection-based defense mechanisms, but on bounded VC-dimension classes as well.

### 2.2 Backdoor-Based Watermarks

In black-box settings, where model auditors lack access to internal parameters, watermarking methods often involve embedding backdoors during training. Techniques by Adi et al. [2018] and

---

[2]Pal and Vidal (2020) consider a game with a slightly different utility than ours.

Zhang et al. [2018] use crafted input patterns as triggers linked to specific outputs, enabling ownership verification by querying the model with these specific inputs. Advanced methods by Merrer et al. [2017] utilize adversarial examples, which are perturbed inputs that yield predefined outputs. Further enhancements by Namba and Sakuma [2019] focus on the robustness of watermarks, ensuring the watermark remains detectable despite model alterations or attacks. In the domain of Natural Language Processing (NLP), backdoor-based watermarks have been studied for Pre-trained Language Models (PLMs)[3], as exemplified by works such as [Gu et al., 2022, Peng et al., 2023] and [Li et al., 2023]. These approaches embed backdoors using rare or common word triggers, ensuring watermark robustness across downstream tasks and resistance to removal techniques like fine-tuning or pruning.

## 2.3 Undetectable Backdoors

A key related work by Goldwasser et al. [2022] shows how a learner can plant undetectable backdoors in any classifier. The authors propose two frameworks: one employing digital signature schemes [Goldwasser et al., 1985] to make backdoored models indistinguishable from the original to any computationally-bounded observer, and another using Random Fourier Features (RFF) [Rahimi and Recht, 2007], which remains undetectable even with full visibility of the model and training data.

In a very recent work, Christiano et al. [2024] introduce a defendability framework that formalizes the interaction between an attacker planting a backdoor and a defender tasked with detecting it. A major difference from our work, is that in their approach, the attacker chooses the distribution, whereas we keep the distribution fixed. This makes defendability in their model harder since the attacker has more control. However, in their framework, the backdoor trigger $x^*$ is sampled from $\mathcal{D}$, so the attacker does not influence it. In contrast, our model allows the attacker to choose specific $x$'s, making defendability in their model easier in this regard. Thus, the definitions are a priori incomparable. However, there are many interesting connections. They show that computationally unbounded defendability is equivalent to PAC learnability, while we, in a similar spirit, show an Adversarial Defense for all tasks with bounded VC-dimension. Using cryptographic tools, they show that the class of polynomial-size circuits is not efficiently defendable, while we use different cryptographic tools to give a Transferable Attack, which rules out a Defense.

## 3 Modeling

A key aspect of our formalization is modeling Alice and Bob while accounting for computational resources. We do so by representing them as families of circuits indexed by a size parameter $n$, a standard approach in complexity theory. Families of Boolean circuits—as used here—are Turing complete and can simulate any algorithm, making them a natural abstraction for studying learning tasks independent of implementation details. Although circuits are less common in computational learning theory than more loosely specified algorithms, this finer granularity is essential for our results.

### 3.1 Learning

**Definition 1** (*Learning Task (Informal)*). Let $\{0,1\}^n$ be an input space[4] A *learning task* $\mathbb{L}$ is defined as a sequence $\{\mathbb{L}_n\}_{n\in\mathbb{N}}$, where each $\mathbb{L}_n$ is a *fixed* distribution over pairs $(\mathcal{D}_n, h_n)$. Concretely, for each $n$, we draw $(\mathcal{D}_n, h_n) \sim \mathbb{L}_n$, where $\mathcal{D}_n$ is a distribution with domain $\{0,1\}^n$, and $h_n : \{0,1\}^n \to \{0,1\}$ is a *ground truth* labeling function.

To every model $f \colon \{0,1\}^n \to \{0,1\}$, we associate $\mathrm{err}(f) := \mathbb{E}_{x\sim\mathcal{D}_n}[f(x) \neq h_n(x)]$. And for $q \in \mathbb{N}, \mathbf{x} \in (\{0,1\}^n)^q$, and predictions $\mathbf{y} \in \{0,1\}^q$, we define the empirical error to be: $\mathrm{err}(\mathbf{x}, \mathbf{y}) := \frac{1}{q} \sum_{i\in[q]} \mathbb{1}_{\{h_n(x_i)\neq y_i\}}$.

**Definition 2** (*Computationally Bounded Learnability (Informal)*). Let $\epsilon, \delta : \mathbb{N} \to (0,1)$ be functions that specify the allowable error and confidence levels for each input size $n$, respectively. A learning

---

[3]We refer readers to Appendix J.3, where we discuss potential avenues for generalizing our framework to generative tasks. We explore the differences between generation and verification.

[4]We work over $\mathbb{F}_2$ (i.e., inputs in $\{0,1\}^n$) for analytic convenience. Any ML pipeline—processing images, tokens, or graphs—executes a finite sequence of arithmetic and logical operations that can be compiled into polynomial-size Boolean circuits.

task $\mathbb{L} = \{\mathbb{L}_n\}_{n \in \mathbb{N}}$ is said to be *learnable* to error $\epsilon(n)$ with confidence $1 - \delta(n)$ and circuit complexity $S(n)$ if there exists a family of circuits $\{C_n\}_{n \in \mathbb{N}}$, where each circuit $C_n$ has size at most $S(n)$, such that for every sufficiently large $n$, the following condition holds:

$$\mathbb{P}_{(\mathcal{D}_n, h_n) \sim \mathbb{L}_n} \left[ \mathrm{err}_{\mathcal{D}_n, h_n}(f_n) \leq \epsilon(n) \right] \geq 1 - \delta(n),$$

where $f_n : \{0,1\}^n \to \{0,1\}$ is the hypothesis computed by the circuit $C_n$ when given sample access to $(\mathcal{D}_n, h_n)$, i.e., $f_n \leftarrow C_n$. In other words, with probability at least $1 - \delta(n)$ over the choice of $(\mathcal{D}_n, h_n)$ drawn from $\mathbb{L}_n$, the circuit $C_n$ successfully computes a function $f_n$ that achieves an error rate of at most $\epsilon(n)$.

Definition 2 is very similar to the standard definition of efficient PAC learnability Kearns and Vazirani [1994]. The main difference is that instead of defining 'efficient' as polynomial in $n$ (and $1/\epsilon, 1/\delta$) we define it as implementable by a circuit of size given by a fixed function $S(n)$. The reason for this increased generality is that we need finer control over sizes than, e.g., polynomial or exponential (see Theorem 1 where the separation between two circuit families is $S(n)$ *versus* $\sqrt{S(n)}$). A second difference is that compared to the standard definition we bound the size of circuits Arora and Barak [2009], not the running time. Assuming a processing unit without parallel execution the two notions can be thought equivalent. Formal definitions and additional details can be found in Appendix B. In the rest of the main part of the paper, we will often omit the parameter $n$ when it is clear context.

**Connections to Existing Models of Learning**   Definition 1 represents a learner's prior knowledge as a distribution over pairs $(\mathcal{D}_n, h_n)$, where $\mathcal{D}_n$ is a distribution on the domain $\{0,1\}^n$ and $h_n : \{0,1\}^n \to \{0,1\}$ is the ground truth. This models a learning task as a distribution over both input distributions and hypotheses, assuming a realizable scenario with a fixed ground truth.

Unlike distribution-specific or restricted family settings Kalai et al. [2008], Feldman et al. [2006], our definition does not limit the underlying support. While standard PAC learning requires generalization across all domain distributions, it often fails to explain the performance of complex models like DNNs, as their rich hypothesis classes make standard PAC bounds ineffective Zhang et al. [2021], Nagarajan and Kolter [2019]. Our definition aims to bridge this gap by providing a formal framework that aligns with contemporary practical learning scenarios.

## 3.2   Interaction

Alice and Bob will engage in interaction. To measure their computational resources, we require a specification of how the model $f_n$ is transmitted between them. We assume that before the interaction starts they agree on a family of function classes $\mathcal{F} = \{\mathcal{F}_n\}_n$ as well as an encoding of them into messages of some length. This modeling implies that $f_n$ are sent *white-box*. One example of such a family is the family of neural networks of a given architecture. See Appendix B for details.

## 3.3   Computational Indistinguishability

A crucial property of interest will be the indistinguishability of distributions. For a pair of distributions $\mathcal{D}^0, \mathcal{D}^1$ consider the following game between a sender and the distinguisher $C$: (1) The sender samples a bit $b \sim U(\{0,1\})$ and then draws a random sample $x \sim \mathcal{D}^b$, (2) $C$ receives $x$ and outputs $\hat{b} := C(x) \in \{0,1\}$. $C$ wins if $\hat{b} = b$. We define the *advantage* of $C$ for *distinguishing* $\mathcal{D}^0$ from $\mathcal{D}^1$

$$\mathbb{P}_{b \sim U(\{0,1\}), x \sim \mathcal{D}^b}[C(x) = b] = \frac{1}{2} + \gamma.$$

For a pair of families of distributions $\mathcal{D}^0 = \{\mathcal{D}_n^0\}_n, \mathcal{D}^1 = \{\mathcal{D}_n^1\}_n$, a function $\gamma : \mathbb{N} \to (0, \frac{1}{2})$, and a size bound $S : \mathbb{N} \to \mathbb{N}$ we say $\mathcal{D}^0, \mathcal{D}^1$ are $\gamma$-*indistinguishable* for circuits of size $S$ if for every $n$, every circuit $C$ (also known as the distinguisher) of size $S(n)$ the *advantage* of $C$ for *distinguishing* $\mathcal{D}_n^0$ from $\mathcal{D}_n^1$ is at most $\gamma(n)$.

# 4   Watermarks, Adversarial Defenses and Transferable Attacks

In our protocols, Alice (**A**, verifier) and Bob (**B**, prover) engage in interactive communication, with distinct roles depending on the specific task. Each protocol is defined with respect to a learning

task $\mathbb{L}$, an error parameter $\varepsilon \in \left(0, \frac{1}{2}\right)$, and circuit size bounds $S_\mathbf{A}$ and $S_\mathbf{B}$, which are functions of $n$. A scheme is successful if the conditions of the protocols are satisfied. We denote the set of such circuits by $\mathrm{SCHEME}(\mathbb{L}, \varepsilon, S_\mathbf{A}(n), S_\mathbf{B}(n))$, where $\mathrm{SCHEME}$ refers to $\mathrm{WATERMARK}$, $\mathrm{DEFENSE}$, or $\mathrm{TRANSFATTACK}$ (see Appendix C for the formal versions of all the definitions).

**Definition 3** (*Watermark, informal*).

A family of circuits $\{\mathbf{A}_n^{\mathrm{WATERMARK}}\}_n$ of sizes $\{S_\mathbf{A}(n)\}_n$, implements a *backdoor-based watermarking scheme* for the learning task $\mathbb{L}$ with error parameter $\epsilon > 0$ if, for every sufficiently large $n$, an interactive protocol in which first $(\mathcal{D}_n, h_n) \sim \mathbb{L}_n$ and then $\mathbf{A}_n^{\mathrm{WATERMARK}}$ computes a classifier $f \colon \{0,1\}^n \to \{0,1\}$ and a sequence of queries $\mathbf{x} \in (\{0,1\}^n)^q$, and a prover $\mathbf{B}_n$ outputs $\mathbf{y} = \mathbf{B}_n(f, \mathbf{x}) \in \{0,1\}^q$, satisfies the following properties:

1. **Correctness:** $f$ has low error, i.e., $\mathrm{err}(f) \leq \epsilon$.
2. **Uniqueness:** There exists a prover $\mathbf{B}_n$, of size $S_\mathbf{A}(n)$, which provides low-error answers, such that $\mathrm{err}(\mathbf{x}, \mathbf{y}) \leq 2\epsilon$.
3. **Unremovability:** For every prover $\mathbf{B}_n$ of size $S_\mathbf{B}(n)$, it holds that $\mathrm{err}(\mathbf{x}, \mathbf{y}) > 2\epsilon$.
4. **Undetectability:** For every prover $\mathbf{B}_n$ of size $S_\mathbf{B}(n)$, the advantage of $\mathbf{B}_n$ in distinguishing the queries $\mathbf{x}$ generated by $\mathbf{A}_n^{\mathrm{WATERMARK}}$ from random queries sampled from $\mathcal{D}_n^q$ is small.

Table 1: Backdoor–based Watermarks (Definition 3).

| Property in Def. 3 | Classical analogue | Why it is needed? |
| --- | --- | --- |
| Correctness | Standard accuracy requirement (e.g., [Adi et al., 2018]) | Ensures watermarking does not degrade task performance. |
| Uniqueness | Verifiability in black-box watermarking | Prevents false positives on independently-trained models. |
| Unremovability | Robustness to pruning / fine-tuning (e.g., [Namba and Sakuma, 2019]) | Captures the usual "cannot be scrubbed" criterion. |
| Undetectability | Stealth requirement (e.g., [Merrer et al., 2017]) | Guarantees watermark triggers look like in-distribution data. |

As summarized in Table 1, uniqueness ensures that watermark verification cannot be triggered by independently trained models. Formally, we require that any $\mathbf{B}_n$ (Bob), who did not use $f$ and trained a model $f_{\mathrm{Scratch}}$ using the specified procedure, must be accepted as distinct. This reflects realistic settings where multiple models could emerge independently.

**Definition 4** (*Adversarial Defense, informal*).

A family of circuits $\{\mathbf{B}_n^{\mathrm{DEFENSE}}\}_n$ of sizes $\{S_\mathbf{B}(n)\}_n$, implements an *adversarial defense* for the learning task $\mathbb{L}$ with error parameter $\epsilon > 0$, if for every sufficiently large $n$, an interactive protocol in which first $(\mathcal{D}_n, h_n) \sim \mathbb{L}_n$ and then $\mathbf{B}_n^{\mathrm{DEFENSE}}$ computes a classifier $f \colon \{0,1\}^n \to \{0,1\}$, while $\mathbf{A}_n$ replies with $\mathbf{x} = \mathbf{A}_n(f)$, where $\mathbf{x} \in (\{0,1\}^n)^q$, and $\mathbf{B}_n^{\mathrm{DEFENSE}}$ outputs $b = \mathbf{B}_n^{\mathrm{DEFENSE}}(f, \mathbf{x}) \in \{0,1\}$, satisfies the following properties:

1. **Correctness:** $f$ has low error, i.e., $\mathrm{err}(f) \leq \epsilon$.
2. **Completeness:** When $\mathbf{x} \sim \mathcal{D}_n^q$, then $b = 0$.
3. **Soundness:** For every $\mathbf{A}_n$ of size $S_\mathbf{A}(n)$, we have $\mathrm{err}(\mathbf{x}, f(\mathbf{x})) \leq 7\epsilon$ or $b = 1$.

The key requirement for a successful defense is the ability to detect when it is being tested (see the soundness and completeness properties in Table 2). To bypass the defense, an $\mathbf{A}_n$ (Alice) must provide samples that are both adversarial, causing the classifier to err, and indistinguishable from samples of $\mathcal{D}_n$.

**Definition 5** (*Transferable Attack, informal*).

A family of circuits $\{\mathbf{A}_n^{\mathrm{TRANSFATTACK}}\}_n$ of sizes $\{S_\mathbf{A}(n)\}_n$, implements a *transferable attack* for the learning task $\mathbb{L}$ with error parameter $\epsilon > 0$, if for every sufficiently large $n$, an interactive protocol in which first $(\mathcal{D}_n, h_n) \sim \mathbb{L}_n$ and then $\mathbf{A}_n^{\mathrm{TRANSFATTACK}}$ computes $\mathbf{x} \in (\{0,1\}^n)^q$ and $\mathbf{B}_n$ outputs $\mathbf{y} = \mathbf{B}_n(\mathbf{x}) \in \{0,1\}^q$ satisfies the following properties:

Table 2: Adversarial Defenses (Definition 4).

| Property in Def. 4 | Classical analogue | Why it is needed? |
|---|---|---|
| Correctness | Baseline test-error requirement in certified / detection-based defences | Ensures the defended model remains useful. |
| Completeness | "No false positive" guarantee in detection frameworks ([Goldwasser et al., 2020]) | Prevents trivial defences that reject everything. |
| Soundness | Detection + robustness guarantee (rejection-based defenses) | Formalises that attacks must both fool and stay indistinguishable. |

1. **Correctness:** Size $S_{\mathbf{B}}(n)$ is sufficient to learn a classifier of low-error, $\mathrm{err}(f) \leq \epsilon$.

2. **Transferability:** For every prover $\mathbf{B}_n$ of size $S_{\mathbf{A}}(n)$, we have $\mathrm{err}(\mathbf{x}, \mathbf{y}) > 2\epsilon$.

3. **Undetectability:** For every prover $\mathbf{B}_n$ of size $S_{\mathbf{B}}(n)$, the advantage of $\mathbf{B}_n$ in distinguishing the queries $\mathbf{x}$ generated by $\mathbf{A}_n^{\text{TRANSFATTACK}}$ from random queries sampled from $\mathcal{D}_n^q$ is small.

Table 3: Transferable Attacks (Definition 5).

| Property in Def. 5 | Classical analogue | Why it is needed? |
|---|---|---|
| Correctness | Baseline learnability precondition | Ensures a meaningful low-error model exists for the attacker to exploit. |
| Transferability | Cross-model adversarial transfer ([Tramèr et al., 2017]) | Captures worst-case attacks that succeed regardless of defender architecture or training procedure. |
| Undetectability | Stealth / indistinguishability ([Goldwasser et al., 2020]) | Guarantees defenders cannot filter the adversarial queries, aligning with cryptographic indistinguishability. |

# 5 Main Result

We are ready to state an informal version of our main theorem (see Appendix D, for the full version). The key idea is to define a *zero-sum game* between $\mathbf{A}_n$ (Alice) and $\mathbf{B}_n$ (Bob), for every $n$, where the actions of each player are all possible circuits that can be realized with size $S_{\mathbf{A}}(n)$ and $S_{\mathbf{B}}(n)$. Notably, this game is finite, but there are exponentially many such actions for each player. We rely on some key properties of such large zero-sum games [Lipton and Young, 1994b] to argue about our main result.

**Theorem 1** (*Main Theorem, informal*). *For every $\epsilon \in \left(0, \frac{1}{2}\right), S : \mathbb{N} \to \mathbb{N}$ and learning task $\mathbb{L}$ learnable to error $\epsilon$ with high confidence with circuit complexity $S(n)$, at least one of these three exists*[5]:

$$\text{WATERMARK}\left(\mathbb{L}, \epsilon, S(n), o\left(\frac{\sqrt{S(n)}}{\log(S(n))}\right)\right),$$

$$\text{DEFENSE}\left(\mathbb{L}, \epsilon, o\left(\frac{\sqrt{S(n)}}{\log(S(n))}\right), O(S(n))\right),$$

$$\text{TRANSFATTACK}\left(\mathbb{L}, \epsilon, S(n), S(n)\right).$$

*Proof (Sketch).* The intuition of the proof relies on the complementary nature of Definitions 3 and 4. Specifically, every attempt to remove a fixed Watermark can be transformed to a potential Adversarial Defense, and vice versa. We define a zero-sum game $\mathcal{G}$ between circuits for watermarking $\mathbf{A}_n$ and circuits attempting to remove a watermark $\mathbf{B}_n$. The set of (pure) strategies of each player are all possible circuits that can be realized with size $S_{\mathbf{A}}(n)$ and $S_{\mathbf{B}}(n)$, and the payoff is determined by the probability that the errors and rejections meet specific requirements. It is well known that this

---

[5]We remark that formally the existence does not hold for all sufficiently large $n$ but only with some 'frequency'. See Theorem 5 for a formal statement.

two-player zero-sum game admits a Nash equilibrium (NE) and the value of the game is unique v. Neumann [1928]. Let $\{\mathbf{A}_n^{\text{NASH}}\}_n$ and $\{\mathbf{B}_n^{\text{NASH}}\}_n$ be the NE strategies of Alice and Bob respectively. For each $n \in \mathbb{N}$, a careful analysis shows that depending on the value of the game, we have a Watermark, an Adversarial Defense, or a Transferable Attack. In the first case, where the expected payoff at the NE is greater than a threshold, we show there is an Adversarial Defense. As an illustration, consider some $n \in \mathbb{N}$, for which we define $\mathbf{B}_n^{\text{DEFENSE}}$ as follows. $\mathbf{B}_n^{\text{DEFENSE}}$ first learns a low-error classifier $f$, then sends $f$ to the party that is attacking the Defense, then receives queries $\mathbf{x}$, and simulates $(\mathbf{y}, b) = \mathbf{B}_n^{\text{NASH}}(f, \mathbf{x})$. The bit $b = 1$ if $\mathbf{B}_n^{\text{NASH}}$ thinks it is attacked. Finally, $\mathbf{B}_n^{\text{DEFENSE}}$ replies with $b' = 1$ if $b = 1$, and if $b = 0$ it replies with $b' = 1$ if the fraction of queries on which $f(\mathbf{x})$ and $\mathbf{y}$ differ is high. Careful analysis shows $\mathbf{B}_n^{\text{DEFENSE}}$ is an Adversarial Defense. In the second case, where the expected payoff at the NE is below the threshold, we have either a Watermark or a Transferable Attack. The full proof can be found in Appendix D. □

# 6 Transferable Attacks and Cryptography

In this section, we show that tasks with Transferable Attacks exist. To construct such examples, we use cryptographic tools. But importantly, the fact that we use cryptography is not coincidental. As a second result of this section, we show that every learning task with a Transferable Attack *implies* a certain cryptographic primitive. One can interpret this as showing that Transferable Attacks exist only for *complex learning tasks*, in the sense of computational complexity theory.

## 6.1 A Cryptography-based Task with a Transferable Attack

Next, we give an example of a cryptography-based learning task with a Transferable Attack. The following is an informal statement of the formal version (Theorem 7) given in Appendix F.

**Theorem 2** (*Transferable Attack for a Cryptography-based Learning Task, informal*). *There exists a learning task $\mathbb{L}^{crypto}$ and $\mathbf{A}$ such that for all sufficiently small $\epsilon$*

$$\mathbf{A} \in \text{TRANSFATTACK}\left(\mathbb{L}^{crypto}, \epsilon, S_{\mathbf{A}} \approx \frac{1}{\epsilon}, S_{\mathbf{B}} = \Omega\left(\frac{1}{\epsilon^2}\right)\right).$$

*Moreover, $\mathbb{L}^{crypto}$ is such that for every $\epsilon$, $\approx \frac{1}{\epsilon}$ time (and $O\left(\frac{1}{\epsilon}\right)$ samples) is enough, and $\Omega\left(\frac{1}{\epsilon}\right)$ samples (and in particular time) is necessary to learn a classifier of error $\epsilon$.*

Notably, the parameters are set so that $\mathbf{A}$ (the party computing $\mathbf{x}$) has a *smaller* circuit size than $\mathbf{B}$ (the party computing $\mathbf{y}$), specifically $\approx 1/\epsilon$ compared to $\Omega(1/\epsilon^2)$. Furthermore, because of the cryptography tools used, this is a setting where a single input maps to multiple outputs, which deviates away from the setting of classification learning tasks considered in Theorem 1.

*Proof (Sketch).* We start with a definition of a learning task that will be later augmented with a cryptographic tool to produce $\mathbb{L}^{crypto}$.

**Lines on Circle Learning Task $\mathbb{L}^{\circ}$ (Figure 2).** We associate the input space $\{0, 1\}^n$ with vertices of a $2^n$ regular polygon inscribed in $\{x \in \mathbb{R}^2 \mid \|x\|_2 = 1\}$. Let $\mathcal{H} := \{h_w \mid w \in \mathbb{R}^2, \|w\|_2 = 1\}$, where $h_w(x) := \text{sgn}(\langle w, x \rangle)$. Let $\mathbb{L}^{\circ}$ be a distribution corresponding to the following process: sample $h_w \sim U(\mathcal{H})$, return $(U(\{0, 1\}^n), h_w)$. Additionally, let $B_w(\alpha) := \{x \in \{0, 1\}^n \mid |\measuredangle(x, w)| \leq \alpha\}$ denote the set of points within an angular distance up to $\alpha$ to $w$.

**Fully Homomorphic Encryption (FHE) (Appendix E).** FHE [Gentry, 2009] allows for computation on encrypted data *without* decrypting it. An FHE scheme allows to encrypt $x$ via an efficient procedure $e_x = \text{FHE.ENC}(x)$, so that later, for any algorithm $C$, it is possible to run $C$ on $x$ *homomorphically*. More concretely, it is possible to produce an encryption of the result of running $C$ on $x$, i.e., $e_{C,x} := \text{FHE.EVAL}(C, e_x)$. Finally, there is a procedure FHE.DEC that, when given a *secret key* sk, can decrypt $e_{C,x}$, i.e., $y := \text{FHE.DEC}(\text{sk}, e_{C,x})$, where $y$ is the result of running $C$ on $x$. Crucially, encryptions of any two messages are indistinguishable for all efficient adversaries.

**Cryptography-based Learning Task $\mathbb{L}^{\text{crypto}}$ (Figure 2).** $\mathbb{L}^{\text{crypto}}$ is derived from *Lines on Circle Learning Task $\mathbb{L}^{\circ}$*. $\mathbb{L}^{\text{crypto}}$ corresponds to the following process: $w \sim U(\{w \in \mathbb{R}^2 \mid \|w\|_2 = 1\})$, return the distribution $\mathcal{D}^w$, which is an equal mixture of two parts $\mathcal{D}^w = \frac{1}{2}\mathcal{D}_{\text{CLEAR}}^w + \frac{1}{2}\mathcal{D}_{\text{ENC}}^w$. The first

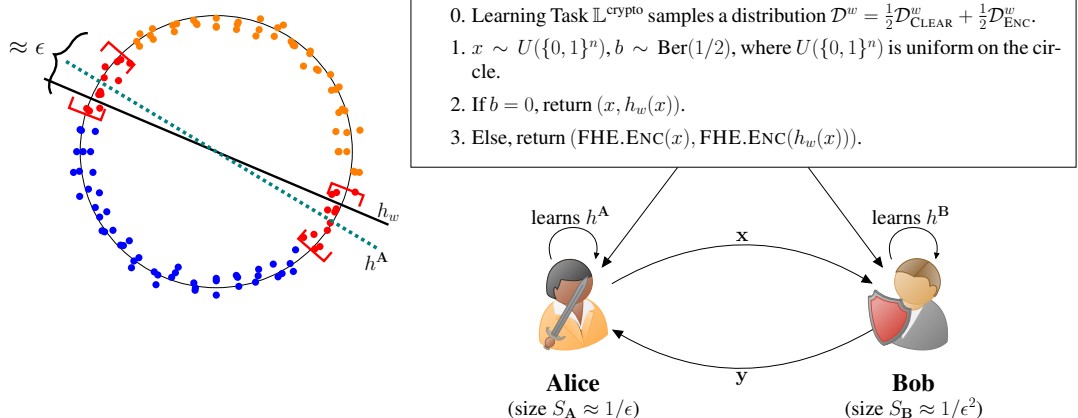

Figure 2: The left part of the figure represents a *Lines on Circle Learning Task* $\mathbb{L}^\circ$ with a ground truth function denoted by $h_w$. On the right, we define a *cryptography-augmented* learning task derived from $\mathbb{L}^\circ$. In its distribution, a "clear" or an "encrypted" sample is observed with equal probability. Given their respective times, both **A** and **B** are able to learn a low-error classifier $h^{\mathbf{A}}$, $h^{\mathbf{B}}$ respectively, by learning only on the *clear samples*. **A** is able to compute a Transferable Attack by computing an encryption of a point close to the decision boundary of her classifier $h^{\mathbf{A}}$.

part, i.e., $\mathcal{D}^w_{\text{CLEAR}}$, is equal to $x \sim U(\{0,1\}^n)$ with the correct label $y = h_w(x)$. The second part, i.e., $\mathcal{D}^w_{\text{ENC}}$, is equal to $x' \sim U(\{0,1\}^n), y' = h_w(x'), (x,y) = (\text{FHE.ENC}(x'), \text{FHE.ENC}(y'))$,[6] which can be thought of as $\mathcal{D}^w_{\text{CLEAR}}$ under an encryption. See Figure 2 for a visual representation. Note that we omitted the size parameter $n$ for simplicity.

**Transferable Attack (Figure 2).** Consider the following attack strategy **A**. First, **A** collects $O(1/\epsilon)$ samples from the distribution $\mathcal{D}^w_{\text{CLEAR}}$ and learns a classifier $h^{\mathbf{A}}_{w'}, \in \mathcal{H}$ that is consistent with these samples. Since the VC-dimension of $\mathcal{H}$ is 2, the hypothesis $h^{\mathbf{A}}_{w'}$ has error at most $\epsilon$ with high probability.[7] Next, **A** samples a point $x_{\text{BND}}$ uniformly at random from a region close to the decision boundary of $h^{\mathbf{A}}_{w'}$, i.e., $x_{\text{BND}} \sim U(B_{w'}(\epsilon))$. Finally, with equal probability, **A** sets as an attack **x** either FHE.ENC$(x_{\text{BND}})$ or a uniformly random point $\mathcal{D}^w_{\text{CLEAR}} = U(\{0,1\}^n)$. We claim[8] that **x** satisfies the properties of a Transferable Attack.

Since $h^{\mathbf{A}}_{w'}$ has a low error with high probability, $x_{\text{BND}}$ is a uniformly random point from an arc containing the boundary of $h_w$ (see Figure 2). The circuit size of **B** is upper-bounded by $\Omega(1/\epsilon^2)$, meaning it can only learn a classifier with error $\gtrsim 10\epsilon^2$ (see Lemma 3 for details). **B**'s can only learn (Lemma 3) a classifier of error, $\gtrsim 10\epsilon^2$. Taking these two facts together, we expect **B** to misclassify $x'$ with probability $\approx \frac{1}{2} \cdot \frac{10\epsilon^2}{\epsilon} = 5\epsilon > 2\epsilon$, where the factor $\frac{1}{2}$ takes into account that we send an encrypted sample only half of the time. This implies *transferability*.

Note that **x** is encrypted with the same probability as in the original distribution because we send FHE.ENC$(x_{\text{BND}})$ and a uniformly random **x** $\sim \mathcal{D}^w_{\text{CLEAR}} = U(\{0,1\}^n)$ with probability $\frac{1}{2}$. Crucially, FHE.ENC$(x_{\text{BND}})$ is indistinguishable, for efficient adversaries, from FHE.ENC$(x)$ for any other $x \in \{0,1\}^n$. This follows from the security of the FHE. Consequently, *undetectability* holds. $\quad\square$

### 6.2 Tasks with Transferable Attacks Imply Cryptography

In this section, we show that a Transferable Attack for any task implies a *cryptographic primitive*.

**EFID Pairs.** In cryptography, an *EFID pair* [Goldreich, 1990] is a pair of ensembles of distributions $\mathcal{D}^0, \mathcal{D}^1$, that are **E**fficiently samplable, statistically **F**ar, and computationally **I**ndistinguishable. By

---

[6]Note that because FHE encryption is probabilistic there are many valid answers for a given $x$.

[7]**A** can also evaluate $h^{\mathbf{A}}_{w'}$ homomorphically (i.e., run FHE.EVAL) on FHE.ENC$(x)$ to obtain FHE.ENC$(y)$ of error $\epsilon$ on $\mathcal{D}^w_{\text{ENC}}$ also. This means that **A** is able to learn a low-error classifier on $\mathcal{D}^w$.

[8]In this proof sketch, we set $q = 1$, i.e., **A** sends only one $x$ to **B**. This is not true for the formal scheme.

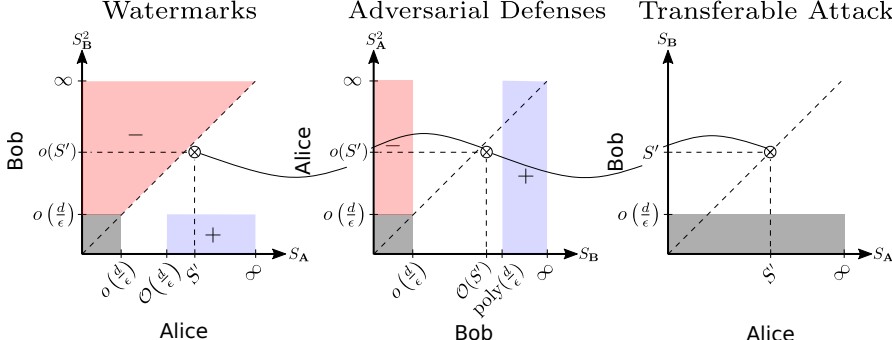

Figure 3: Overview of the taxonomy of learning tasks, illustrating the presence of Watermarks, Adversarial Defenses, and Transferable Attacks for learning tasks of bounded VC dimension. The axes represent the size bound for the parties in the corresponding schemes. The blue regions depict positive results, the red negative, and the gray regimes of parameters which are not of interest. See Lemma 5 and 6 for details about blue regions. The curved line represents a potential application of Theorem 1, which says that at least one of the three points should be blue.

a seminal result [Goldreich, 1990], we know that the existence of EFID pairs is equivalent to the existence of *Pseudorandom Generators* (PRG), which can be used for tasks including encryption and key generation [Goldreich, 1990], which makes EFID pairs a useful primitive. We consider a slight modification of the standard definition of EFID pairs, where instead of defining security to hold against polynomial time adversaries we do it for a fixed size bound function. More concretely, for two size bounds $S, S' : \mathbb{N} \to \mathbb{N}$ we call a pair of ensembles of distributions $(\mathcal{D}^0, \mathcal{D}^1)$ an $(S, S')$-EFID pair if for every $n$ (i) $\mathcal{D}_n^0, \mathcal{D}_n^1$ are samplable by circuits of size $S(n)$, (ii) $\mathcal{D}_n^0, \mathcal{D}_n^1$ are statistically far, (iii) $\mathcal{D}_n^0, \mathcal{D}_n^1$ are indistinguishable for circuits of size $S'(n)$.

**Tasks with Transferable Attacks imply EFID Pairs.** The second result shows that any task with a Transferable Attack implies the existence of a type of EFID pair. This guarantees that any learning task with a Transferable Attack has to be computationally complex. The proof is in Appendix G.

**Theorem 3** (*Transferable Attacks imply EFID pairs, informal*). *For every $\epsilon \in (0, 1), S_{\mathbf{A}}, S_{\mathbf{B}} : \mathbb{N} \to \mathbb{N}, S_{\mathbf{A}} \le S_{\mathbf{B}}$, every learning task $\mathbb{L}$ learnable to error $\epsilon$ with high confidence and circuit complexity $S_{\mathbf{A}}$ if there exists $\text{TRANSATTACK}(\mathbb{L}, \epsilon, S_{\mathbf{A}}, S_{\mathbf{B}})$ then there exists an $(S_{\mathbf{A}}, S_{\mathbf{B}})$-EFID pair.*

We note that it is unclear if the existence of EFID-pairs guaranteed by Theorem 3 implies PRGs because the sampling of $\mathcal{D}^0, \mathcal{D}^1$ requires oracle access to $\mathbb{L}$. Therefore, the standard construction of PRGs from EFID pairs does not automatically transfer.

## 7 Tasks with Watermarks and Adversarial Defenses

As the final pair of results, we present tasks exhibiting Watermarks and Adversarial Defenses. In the first, hypothesis classes with polynomially bounded VC-dimension admit polynomial-size Adversarial Defenses against all attackers. In the second, a learning task of polynomially bounded VC-dimension admits a Watermark secure against fast adversaries. These lemmas highlight the importance of bounding the sizes of $\mathbf{A}$ and $\mathbf{B}$. See Figure 3 for a visual summary; formal statements and proofs appear in Appendices H and I.

**Lemma 1** (*Adversarial Defense for bounded VC-dimension, informal*). *There exists $\mathbf{B}$ such that for every $n$, every learning task $\mathbb{L}$ of VC-dimension $n^9$, every sufficiently small $\epsilon$,*

$$\mathbf{B} \in \text{DEFENSE}\left(\mathbb{L}, \epsilon, S_{\mathbf{A}} = \infty, S_{\mathbf{B}} = \texttt{poly}\left(\tfrac{n}{\epsilon}\right)\right).$$

**Lemma 2** (*Watermark for bounded VC-dimension against fast adversaries, informal*). *For every $d$, there exists a learning task $\mathbb{L}$ of VC-dimension $d$ and $\mathbf{A}$ such that for every sufficiently small $\epsilon$,*

$$\mathbf{A} \in \text{WATERMARK}\left(\mathbb{L}, \ \epsilon, \ q = O\left(\tfrac{1}{\epsilon}\right), \ S_{\mathbf{A}} = O\left(\tfrac{d}{\epsilon}\right), \ S_{\mathbf{B}} = \tfrac{d}{100}\right).$$

---

[9]It means that the ground truth sampled from $\mathbb{L}$ belongs to a class of VC-dimension $n$.

## Acknowledgement

This research was partially supported by the Deutsche Forschungsgemeinschaft (DFG, German Research Foundation) under Germany's Excellence Strategy – The Berlin Mathematics Research Center MATH+ (EXC-2046/1, project ID: 390685689). We thank the anonymous reviewers for their thoughtful comments and suggestions, which improved the paper.

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

# A  Additional Methods in Related Work

This section provides an overview of the main areas relevant to our work: Watermarking techniques, adversarial defenses, and transferable attacks on Deep Neural Networks (DNNs). Each subsection outlines important contributions and the current state of research in these areas, offering additional context and details beyond those covered in the main body

## A.1  Watermarking

Watermarking techniques are crucial for protecting the intellectual property of machine learning models. These techniques can be broadly categorized based on the type of model they target. We review watermarking schemes for both classification and generative models, with a primary focus on classification models, as our work builds upon these methods.

### A.1.1  Watermarking Schemes for classification Models

classification models, which are designed to categorize input data into predefined classes, have been a major focus of watermarking research. The key approaches in this domain can be divided into black-box and white-box approaches.

**Black-Box Setting.**   In the black-box setting, the model owner does not have access to the internal parameters or architecture of the model, but can query the model to observe its outputs. This setting has seen the development of several watermarking techniques, primarily through backdoor-like methods.

Adi et al. [2018] and Zhang et al. [2018] proposed frameworks that embed watermarks using specifically crafted input data (e.g., unique patterns) with predefined outcomes. These watermarks can be verified by feeding these special inputs into the model and checking for the expected outputs, thereby confirming ownership.

Another significant contribution in this domain is by Merrer et al. [2017], who introduced a method that employs adversarial examples to embed the backdoor. Adversarial examples are perturbed inputs that cause the model to produce specific outputs, thus serving as a watermark.

Namba and Sakuma [2019] further enhanced the robustness of black-box watermarking schemes by developing techniques that withstand various model modifications and attacks. These methods ensure that the watermark remains intact and detectable even when the model undergoes transformations.

Provable undetectability of backdoors was achieved in the context of classification tasks by Goldwasser et al. [2022]. Unfortunately, it is known ([Goldwasser et al., 2022]) that some undetectable watermarks are easily removed by simple mechanisms similar to randomized smoothing [Cohen et al., 2019].

The popularity of black-box watermarking is due to its practical applicability, as it does not require access to the model's internal workings. This makes it suitable for scenarios where models are deployed as APIs or services. Our framework builds upon these black-box watermarking techniques.

**White-Box Setting.**   In contrast, the white-box setting assumes that the model owner has full access to the model's parameters and architecture, allowing for direct examination to confirm ownership. The initial methodologies for embedding watermarks into the weights of DNNs were introduced by Uchida et al. [2017] and Nagai et al. [2018]. Uchida et al. [2017] presented a framework for embedding watermarks into the model weights, which can be examined to confirm ownership.

An advancement in white-box watermarking is provided by Darvish Rouhani et al. [2019], who developed a technique to embed an $N$-bit ($N \geq 1$) watermark in DNNs. This technique is both *data-* and *model-dependent*, meaning the watermark is activated only when specific data inputs are fed into the model. For revealing the watermark, activations from intermediate layers are necessary in the case of white-box access, whereas only the final layer's output is needed for black-box scenarios.

Our work does not focus on white-box watermarking techniques. Instead, we concentrate on exploring the interaction between backdoor-like watermarking techniques, adversarial defenses, and transferable attacks. Overall, watermarking through backdooring has become more popular due to its applicability in the black-box setting.

### A.1.2 Watermarking Schemes for Generative Models

Watermarking techniques for generative models have attracted considerable attention with the advent of Large Language Models (LLMs) and other advanced generative models. This increased interest has led to a surge in research and diverse contributions in this area.

**Backdoor-Based Watermarking for Pre-trained Language Models.** In the domain of Natural Language Processing (NLP), backdoor-based watermarks have been increasingly studied for Pre-trained Language Models (PLMs), as exemplified by works such as [Gu et al., 2022] and [Li et al., 2023]. These methods leverage rare or common word triggers to embed watermarks, ensuring that they remain robust across downstream tasks and resilient to removal techniques like fine-tuning or pruning. While these approaches have demonstrated promising results in practical applications, they are primarily empirical, with theoretical aspects of watermarking and robustness requiring further exploration.

**Watermarking the Output of LLMs.** Watermarking the generated text of LLMs is critical for mitigating potential harms. Significant contributions in this domain include [Kirchenbauer et al., 2023], who proposed a watermarking framework that embeds signals into generated text that are invisible to humans but detectable algorithmically. This method promotes the use of a randomized set of "green" tokens during text generation, and detects the watermark without access to the language model API or parameters.

Kuditipudi et al. [2023] introduced robust distortion-free watermarks for language models. Their method ensures that the watermark does not distort the generated text, providing robustness against various text manipulations while maintaining the quality of the output.

Zhao et al. [2023a] presented a provable, robust watermarking technique for AI-generated text. This approach offers strong theoretical guarantees for the robustness of the watermark, making it resilient against attempts to remove or alter it without significantly changing the generated text.

However, Zhang et al. [2023] highlighted vulnerabilities in these watermarking schemes. Their work demonstrates that current watermarking techniques can be effectively broken, raising important considerations for the future development of robust and secure watermarking methods for LLMs.

**Image Generation Models.** Various watermarking techniques have been developed for image generation models to address ethical and legal concerns. Fernandez et al. [2023] introduced a method combining image watermarking with Latent Diffusion Models, embedding invisible watermarks in generated images for future detection. This approach is robust against modifications such as cropping. Wen et al. [2023b] proposed Tree-Ring Watermarking, which embeds a pattern into the initial noise vector during sampling, making the watermark robust to transformations like convolutions and rotations. Jiang et al. [2023] highlighted vulnerabilities in watermarking schemes, showing that human-imperceptible perturbations can evade watermark detection while maintaining visual quality. Zhao et al. [2023c] provided a comprehensive analysis of watermarking techniques for Diffusion Models, offering a recipe for efficiently watermarking models like Stable Diffusion, either through training from scratch or fine-tuning. Additionally, Zhao et al. [2023b] demonstrated that invisible watermarks are vulnerable to regeneration attacks that remove watermarks by adding random noise and reconstructing the image, suggesting a shift towards using semantically similar watermarks for better resilience.

**Audio Generation Models.** Watermarking techniques for audio generators have been developed for robustness against various attacks. Erfani et al. [2017] introduced a spikegram-based method, embedding watermarks in high-amplitude kernels, robust against MP3 compression and other attacks while preserving quality. Liu et al. [2023] proposed DeAR, a deep-learning-based approach resistant to audio re-recording (AR) distortions.

### A.2 Adversarial Defense

The field of adversarial robustness has a rich and extensive literature [Szegedy et al., 2014, Gilmer et al., 2018, Raghunathan et al., 2018, Wong and Kolter, 2018, Engstrom et al., 2017]. Adversarial defenses are essential for ensuring the security and reliability of machine learning models against adversarial attacks that aim to deceive them with carefully crafted inputs.

For classification models, there has been significant progress in developing adversarial defenses. Techniques such as adversarial training [Madry et al., 2018], which involves training the model on adversarial examples, have shown promise in improving robustness. Certified defenses [Raghunathan et al., 2018] provide provable guarantees against adversarial attacks, ensuring that the model's predictions remain unchanged within a specified perturbation bound. Additionally, methods like *randomized smoothing* [Cohen et al., 2019] offer robustness guarantees.

A particularly relevant work for our study is [Goldwasser et al., 2020], which considers a different model for generating adversarial examples. This approach has significant implications for the robustness of watermarking techniques in the face of adversarial attacks.

In the context of LLMs, there is a rapidly growing body of research focused on identifying adversarial examples [Zou et al., 2023, Carlini et al., 2023, Wen et al., 2023a]. This research is closely related to the notion of *jailbreaking* [Andriushchenko et al., 2024, Chao et al., 2023, Mehrotra et al., 2024, Wei et al., 2023], which involves manipulating models to bypass their intended constraints and protections.

### A.3 Transferable Attacks and Transductive Learning

Transferable attacks refer to adversarial examples that are effective across multiple models. Moreover, *transductive learning* has been explored as a means to enhance adversarial robustness, and since our Definition 5 captures some notion of transductive learning in the context of Transferable Attacks, we highlight significant contributions in these areas.

**Adversarial Robustness via Transductive Learning.** Transductive learning [Gammerman et al., 1998] has shown promise in improving the robustness of models by utilizing both training and test data during the learning process. This approach aims to make models more resilient to adversarial perturbations encountered at test time.

One significant contribution is by Goldwasser et al. [2020], which explores learning guarantees in the presence of arbitrary adversarial test examples, providing a foundational framework for transductive robustness. Another notable study by Chen et al. [2021] formalizes transductive robustness and proposes a bilevel attack objective to challenge transductive defenses, presenting both theoretical and empirical support for transductive learning's utility.

Additionally, Montasser et al. [2022] introduce a transductive learning model that adapts to perturbation complexity, achieving a robust error rate proportional to the VC dimension. The method by Wu et al. [2020] improves robustness by dynamically adjusting the network during runtime to mask gradients and cleanse non-robust features, validated through experimental results. Lastly, Tramer et al. [2020] critique the standard of adaptive attacks, demonstrating the need for specific tuning to effectively evaluate and enhance adversarial defenses.

**Transferable Attacks on DNNs.** Transferable attacks exploit the vulnerability of models to adversarial examples that generalize across different models. For classification models, significant works include Liu et al. [2016], which investigates the transferability of adversarial examples and their effectiveness in black-box attack scenarios, [Xie et al., 2018], who propose input diversity techniques to enhance the transferability of adversarial examples across different models, and [Dong et al., 2019], which presents translation-invariant attacks to evade defenses and improve the effectiveness of transferable adversarial examples.

In the context of generative models, including LLMs and other advanced generative architectures, relevant research is rapidly emerging, focusing on the transferability of adversarial attacks. This area is crucial as it aims to understand and mitigate the risks associated with adversarial examples in these powerful models. Notably, Zou et al. [2023] explored universal and transferable adversarial attacks on aligned language models, highlighting the potential vulnerabilities and the need for robust defenses in these systems.

### A.4 Interactive Proof Systems in Machine Learning

*Interactive Proof Systems* [Goldwasser and Sipser, 1986] have recently gained considerable attention in machine learning for their ability to formalize and verify complex interactions between agents, models, or even human participants. A key advancement in this area is the introduction of *Prover-*

*Verifier Games* (PVGs) [Anil et al., 2021], which employ a game-theoretic approach to guide learning agents towards decision-making with verifiable outcomes. Building on PVGs, Kirchner et al. [2024] enhance this framework to improve the legibility of Large Language Models (LLMs) outputs, making them more accessible for human evaluation. Similarly, Wäldchen et al. [2024] apply the prover-verifier setup to offer interpretability guarantees for classifiers. Extending these concepts, self-proving models Amit et al. [2024] introduce generative models that not only produce outputs but also generate proof transcripts to validate their correctness. In the context of AI safety, scalable *debate protocols* [Condon et al., 1993, Irving et al., 2018, Brown-Cohen et al., 2023] leverage interactive proof systems to enable complex decision processes to be broken down into verifiable components, ensuring reliability even under adversarial conditions.

| | | Undetectability | Unremovability | Uniqueness |
|---|---|---|---|---|
| **Classification** | Goldwasser et al. [2022] | ✔ | robust to some smoothing attacks | ✔(E) |
| | Adi et al. [2018], Zhang et al. [2018] | ✔(E) | ✘ | ✔(E) |
| | Merrer et al. [2017] | ✔(E) | robust to fine tunning attacks | ✔(E) |
| **LLMs** | Christ et al. [2023], Kuditipudi et al. [2023] | ✔ | ✘ | ✔ |
| | Zhao et al. [2023a] | ✘ | robust to edit distance attacks only | ✔ |
| | Tiffany Hsu [2023] | ✔(E) | ✘ | ✔ |
| | Kirchenbauer et al. [2023] | ✘ | ✘ | ✔ |

Table 4: Overview of properties across various watermarking schemes. The symbol ✔ denotes properties with formal guarantees or where proof is plausible, whereas ✘ indicates the absence of such guarantees. Entries marked with ✔(E) represent properties observed empirically; these lack formal proof in the corresponding literature, suggesting that deriving such proof may present substantial challenges. The LLM watermarking schemes refer to those applied to text generated by these models.

# B Preliminaries

For $n \in \mathbb{N}$ we define $[n] := \{1, \ldots, n\}$. We say a boolean sequence $a : \mathbb{N} \to \{0, 1\}$ is true with frequency $\alpha \in [0, 1]$ if

$$\liminf_{n \to \infty} \frac{\sum_{i \in [n]} a(i)}{n} \geq \alpha.$$

For two sequences $a, b : \mathbb{N} \to \mathbb{R}$ we say they agree with frequency at least $\alpha \in [0, 1]$ if the sequence $(a \overset{?}{=} b) : \mathbb{N} \to \{0, 1\}$, i.e. $(a \overset{?}{=} b)(n) = \mathbb{1}_{a(n)=b(n)}$, is true with frequency $\alpha$.

**Learning.** For a set $\Omega$, we write $\Delta(\Omega)$ to denote the set of all probability measures defined on the measurable space $(\Omega, \mathcal{F})$, where $\mathcal{F}$ is some fixed $\sigma$-algebra that is implicitly understood. For a parameter $n$, we denote by $\{0, 1\}^n$ the input space and by $\{0, 1\}$ the output space. A *model* is a function $f : \{0, 1\}^n \to \{0, 1\}$.

**Definition 6** (*Learning Task*). A *learning task* $\mathbb{L}$ is a family $\{\mathbb{L}_n\}_{n \in \mathbb{N}}$, where for every $n$, $\mathbb{L}_n$ is an element of $\Delta\left(\Delta(\{0, 1\}^n) \times \{0, 1\}^{\{0,1\}^n}\right)$.

For a *distribution* $\mathcal{D}_n \in \Delta(\{0, 1\}^n)$ and a *ground truth* $h_n : \{0, 1\}^n \to \{0, 1\}$, we define an *error* of $f$ as $\text{err}_{\mathcal{D}_n, h_n}(f) := \mathbb{E}_{x \sim \mathcal{D}_n}[f(x) \neq h(x)]$, where the index of err will often be understood implicitly and omitted in notation. For $\mathcal{D}_n \in \Delta(\{0, 1\}^n), h_n : \{0, 1\}^n \to \{0, 1\}$ we define an *example oracle* $\text{Ex}(\mathcal{D}_n, h_n)$ as an oracle that samples $x \sim \mathcal{D}_n$ and returns $(x, h_n(x))$.

**Interaction.** When $\text{Ex}(\mathcal{D}, h)$ generates $(x, h(x))$ it is encoded as an $n + 1$ bit-string, because $x \in \{0, 1\}^n$ and the label space is $\{0, 1\}$. For a *message space* $\mathcal{M} = \{\mathcal{M}_n\}_n = \{\{0, 1\}^{m(n)}\}_n$ a *representation class* is a collection of mappings $\{\mathcal{R}_n\}_n$, where for every $n$, $\mathcal{R}_n : \mathcal{M}_n \to \{0, 1\}^{\{0,1\}^n}$. Thus, there is a function class corresponding to a representation, i.e., for every $n$ there is a function class $\mathcal{F}_n$, which is an image of $\mathcal{R}_n$. Note that $h_n$ (which is the ground truth) may or may not be in $\mathcal{F}_n$. All function classes considered in this work have an implicit representation class and an underlying message space.

**Computation.** We work with the collection of Boolean circuits over the standard basis $B_2$, the set of all two-bit Boolean functions. The size of a circuit $C$ is measured by its number of gates; let $|C|$ denote the size of $C$. For a circuit family $\mathcal{C} = \{C_n\}_n$ we say it has a circuit complexity $S(n)$ if for every $n$, $|C_n| \leq S(n)$.

For a distribution $\mathcal{D}_n$ over $\{0, 1\}^n$, and a ground truth $h_n : \{0, 1\}^n \to \{0, 1\}$ we denote by $C^{\text{Ex}(\mathcal{D}_n, h_n)}$ a circuit with some[10] number of specified input gates that are initialized with samples $(x, h(x))$ sampled from $x \sim \mathcal{D}_n$. We will also by interested in interaction between circuits. When messages are exchanged between circuits we assume that there are specified input (output) gates that correspond to outgoing (ingoing) messages. Also, when a circuit is randomized we assume there are designated input gates that are initialized with random bits.

**Definition 7** (*Computationally Bounded Learnability*). For $\epsilon, \delta : \mathbb{N} \to (0, 1)$ we say that a learning task $\mathbb{L} = \{\mathbb{L}_n\}_{n \in \mathbb{N}}$ is learnable to error $\epsilon$ with confidence $1 - \delta$ and with circuit complexity $S : \mathbb{N} \to \mathbb{N}$ by a function class $\mathcal{F} = \{\mathcal{F}_n\}_{n \in \mathbb{N}}$ (with a corresponding representation class $\mathcal{R}$), or $(\epsilon, \delta, S, \mathcal{F})$-learnable in short, if there exists a circuit family $\mathcal{C} = \{C_n\}_{n \in \mathbb{N}}$ with complexity $S(n)$ such that for every sufficiently large $n$, with probability $1 - \delta$ over the choice of $(\mathcal{D}_n, h_n) \sim \mathbb{L}_n$, $C_n^{\text{Ex}(\mathcal{D}_n, h_n)}$ computes an $m(n)$ bit message $m_{f_n} \in \mathcal{M}_n$ such that $\mathcal{R}_n(m_{f_n}) \in \mathcal{F}_n$ has error at most $\epsilon$, i.e. for every sufficiently large $n$

$$\mathbb{P}_{(\mathcal{D}_n, h_n) \sim \mathbb{L}_n, m_{f_n} \leftarrow C_n^{\text{Ex}(\mathcal{D}_n, h_n)}} \left[ \text{err}_{\mathcal{D}_n, h_n}(\mathcal{R}_n(m_{f_n})) \leq \epsilon(n) \right] \geq 1 - \delta(n).$$

We often abuse the notation and use $f_n$ to denote both $m_{f_n}$ as well as $\mathcal{R}_n(m_{f_n})$.

## C Formal Definitions

**Definition 8** (*Watermark*). Let $\mathbb{L} = \{\mathbb{L}_n\}_n$ be a learning task, and $\mathcal{F} = \{\mathcal{F}_n\}_n$ a function class. Let $S_{\mathbf{A}}, S_{\mathbf{B}}, q : \mathbb{N} \to \mathbb{N}, \epsilon \in \left(0, \frac{1}{2}\right), l, c, s \in (0, 1), s < c$, where $S_{\mathbf{B}}(n)$ bounds the circuit size of $\mathbf{B}_n$, and $S_{\mathbf{A}}(n)$ the circuit size of $\mathbf{A}_n$, $q(n)$ the number of queries, $\epsilon$ the risk level, $c$ probability that *uniqueness* holds, $s$ probability that *unremovability* and *undetectability* holds, $l$ the learning probability.

We say that a family of circuits $\mathbf{A}^{\text{WATERMARK}} = \{\mathbf{A}_n^{\text{WATERMARK}}\}_n$ *with complexity* $S_{\mathbf{A}}(n)$ implements a watermarking scheme for $\mathbb{L}$ with frequency $\alpha$, denoted by

$$\mathbf{A}^{\text{WATERMARK}} \in_\alpha \text{WATERMARK}\left(\mathbb{L}, \mathcal{F}, \epsilon, q, S_{\mathbf{A}}, S_{\mathbf{B}}, l, c, s\right),$$

if the following is true with frequency $\alpha$ over parameter $n$. An interactive protocol in which first $(\mathcal{D}_n, h_n) \sim \mathbb{L}_n$ and then $\mathbf{A}_n^{\text{WATERMARK}}$ computes $(f, \mathbf{x})$, $f : \{0, 1\}^n \to \{0, 1\}, \mathbf{x} \in (\{0, 1\}^n)^{q(n)}$, and $\mathbf{B}_n$ outputs $\mathbf{y} = \mathbf{B}_n(f, \mathbf{x}), \mathbf{y} \in \{0, 1\}^{q(n)}$, where $f$ is sent using the representation $\mathcal{R}_n$, satisfies the following

- **Correctness** ($f$ has low error). With probability at least $l$

$$\text{err}(f) \leq \epsilon.$$

- **Uniqueness** (models trained from scratch give low-error answers). There exists a circuit $\mathbf{B}_n$ of size $S_{\mathbf{A}}(n)$ such that with probability at least $c$

$$\text{err}(\mathbf{x}, \mathbf{y}) \leq 2\epsilon.$$

---

[10]We will not specify the sample complexity explicitly. In this paper, we focus only on circuit complexity. The sample complexity is an important parameter to analyze and we leave it for future work. We emphasize that the circuit complexity is an upper bound on the sample complexity.

- **Unremovability** (fast $\mathbf{B}_n$ give high-error answers). For every circuit $\mathbf{B}_n$ *of size at most* $S_{\mathbf{B}}(n)$ with probability at most $s$

$$\mathrm{err}(\mathbf{x}, \mathbf{y}) \leq 2\epsilon.$$

- **Undetectability** (fast $\mathbf{B}_n$ cannot detect that they are tested). On average over $(\mathcal{D}_n, h_n) \sim \mathbb{L}_n$, distributions $\mathcal{D}_n^{q(n)}$ and $\mathbf{x} \sim \mathbf{A}_n^{\text{WATERMARK}}$ are $\frac{s}{2}$-indistinguishable for a class of circuits $\mathbf{B}_n$ *of size at most* $S_{\mathbf{B}}(n)$, i.e., for every circuit $\mathbf{B}_n$ of size at most $S_{\mathbf{B}}(n)$ returning one bit,

$$\left| \mathbb{P}_{(\mathcal{D}_n, h_n) \sim \mathbb{L}_n, \mathbf{x}' \sim \mathcal{D}_n^{q(n)}, (f, \mathbf{x}) \leftarrow \mathbf{A}_n^{\text{WATERMARK}}} [\mathbf{B}(f, \mathbf{x}') = 0] - \mathbb{P}_{(\mathcal{D}_n, h_n) \sim \mathbb{L}, (f, \mathbf{x}) \leftarrow \mathbf{A}_n^{\text{WATERMARK}}} [\mathbf{B}(f, \mathbf{x}) = 0] \right| \leq \frac{s}{2}.$$

**Definition 9** (*Adversarial Defense*). Let $\mathbb{L} = \{\mathbb{L}_n\}_n$ be a learning task, and $\mathcal{F} = \{\mathcal{F}_n\}_n$ a function class. Let $S_{\mathbf{A}}, S_{\mathbf{B}}, q : \mathbb{N} \to \mathbb{N}$, $\epsilon \in \left(0, \frac{1}{2}\right)$, $l, c, s \in (0, 1)$, with $s < c$, where $S_{\mathbf{A}}(n)$ bounds the circuit size of $\mathbf{A}_n$, and $S_{\mathbf{B}}(n)$ the circuit size of $\mathbf{B}_n$, $q(n)$ the number of queries, $\epsilon$ the error parameter, $c$ the completeness, $s$ the soundness, and $l$ the learning probability.

We say that a family of circuits $\mathbf{B}^{\text{DEFENSE}} = \{\mathbf{B}_n^{\text{DEFENSE}}\}_n$ *with complexity* $S_{\mathbf{A}}(n)$ implements an adversarial defense for $\mathbb{L}$ with frequency $\alpha$, denoted by

$$\mathbf{B}^{\text{DEFENSE}} \in_\alpha \text{DEFENSE}\left(\mathbb{L}, \mathcal{F}, \epsilon, q, S_{\mathbf{A}}, S_{\mathbf{B}}, l, c, s\right),$$

if the following is true with frequency $\alpha$ over parameter $n$. An interactive protocol in which first $(\mathcal{D}_n, h_n) \sim \mathbb{L}_n$, $\mathbf{B}_n^{\text{DEFENSE}}$ computes $f : \{0, 1\}^n \to \{0, 1\}$, $\mathbf{A}_n$ replies with $\mathbf{x} = \mathbf{A}_n(f_n)$, $\mathbf{x} \in (\{0, 1\}^n)^{q(n)}$, and $\mathbf{B}_n^{\text{DEFENSE}}$ outputs $b = \mathbf{B}_n^{\text{DEFENSE}}(f, \mathbf{x})$, $b \in \{0, 1\}$, satisfies the following:

- **Correctness** ($f_n$ has low error). With probability at least $l$

$$\mathrm{err}(f) \leq \epsilon.$$

- **Completeness** (natural inputs are not flagged as adversarial). When $\mathbf{x} \sim \mathcal{D}_n^{q(n)}$, with probability at least $c$

$$b = 0.$$

- **Soundness** (adversarial inputs are detected). For every circuit $\mathbf{A}_n$ of size at most $S_{\mathbf{A}}(n)$, with probability at most $s$

$$\mathrm{err}(\mathbf{x}, f(\mathbf{x})) > 7\epsilon \text{ and } b = 0.$$

**Definition 10** (*Transferable Attack*). Let $\mathbb{L} = \{\mathbb{L}_n\}_n$ be a learning task and $\mathcal{F} = \{\mathcal{F}_n\}_n$ a function class. Let $S_{\mathbf{A}}, S_{\mathbf{B}}, q : \mathbb{N} \to \mathbb{N}$, $\epsilon \in \left(0, \frac{1}{2}\right)$, and $c, s \in (0, 1)$, with $s < c$, where $S_{\mathbf{A}}(n)$ bounds the circuit size of $\mathbf{A}_n$, and $S_{\mathbf{B}}$ the circuit size of $\mathbf{B}_n$, $q(n)$ the number of queries, $\epsilon$ the error parameter, $c$ the *transferability* probability, and $s$ the *undetectability* probability.

We say that a family of circuits $\mathbf{A}^{\text{TRANSFATTACK}} = \{\mathbf{A}_n^{\text{TRANSFATTACK}}\}$ *with complexity* $S_{\mathbf{A}}(n)$ implements a transferable attack for $\mathbb{L}$ with frequency $\alpha$, denoted by

$$\mathbf{A}^{\text{TRANSFATTACK}} \in_\alpha \text{DEFENSE}\left(\mathbb{L}, \mathcal{F}, \epsilon, q, S_{\mathbf{A}}, S_{\mathbf{B}}, l, c, s\right),$$

if the following is true with frequency $\alpha$ over parameter $n$. An interactive protocol in which first $(\mathcal{D}_n, h_n) \sim \mathbb{L}_n$, $\mathbf{A}_n^{\text{TRANSFATTACK}}$ computes $\mathbf{x} \in (\{0, 1\}^n)^{q(n)}$, and $\mathbf{B}_n$ outputs $\mathbf{y} = \mathbf{B}_n(\mathbf{x})$, $\mathbf{y} \in (\{0, 1\})^{q(n)}$, satisfies the following:

- **Transferability** (fast provers return high-error answers). For every circuit $\mathbf{B}_n$ of size at most $S_{\mathbf{B}}(n)$, with probability at least $c$

$$\mathrm{err}(\mathbf{x}, \mathbf{y}) > 2\epsilon.$$

- **Undetectability** (fast provers cannot detect that they are tested). On average over $(\mathcal{D}_n, h_n) \sim \mathbb{L}_n$, distributions $\mathbf{x} \sim \mathcal{D}_n^{q(n)}$ and $\mathbf{x} := \mathbf{A}_n^{\text{TRANSFATTACK}}$ are $\frac{s}{2}$-indistinguishable for every circuit $\mathbf{B}_n$ of size at most $S_{\mathbf{B}}(n)$, i.e.,

$$\left| \mathbb{P}_{(\mathcal{D}_n, h_n) \sim \mathbb{L}_n, \mathbf{x}' \sim \mathcal{D}_n^{q(n)}} [\mathbf{B}_n(\mathbf{x}') = 0] - \mathbb{P}_{(\mathcal{D}_n, h_n) \sim \mathbb{L}_n} [\mathbf{B}_n(\mathbf{x}) = 0] \right| \leq \frac{s}{2}.$$

# D   Main Theorem

Before proving our main theorem we recall a result from Lipton and Young [1994a] about simple strategies for large zero-sum games.

**Game theory.**   A *two-player zero-sum game* is specified by a payoff matrix $\mathcal{G}$. $\mathcal{G}$ is an $r \times c$ matrix. MIN, the row player, chooses a probability distribution $p_1$ over the rows. MAX, the column player, chooses a probability distribution $p_2$ over the columns. A row $i$ and a column $j$ are drawn from $p_1$ and $p_2$ and MIN pays $\mathcal{G}_{ij}$ to MAX. MIN tries to minimize the expected payment; MAX tries to maximize it.

By the Min-Max Theorem, there exist optimal strategies for both MIN and MAX. Optimal means that playing first and revealing one's mixed strategy is not a disadvantage. Such a pair of strategies is also known as a Nash equilibrium. The expected payoff when both players play optimally is known as the value of the game and is denoted by $\mathcal{V}(\mathcal{G})$.

We will use the following theorem from Lipton and Young [1994a], which says that optimal strategies can be approximated by uniform distributions over sets of pure strategies of size $O(\log(c))$.

**Theorem 4** (Lipton and Young [1994a]). *Let $\mathcal{G}$ be an $r \times c$ payoff matrix for a two-player zero-sum game. For any $\eta \in (0,1)$ and $k \geq \frac{\log(c)}{2\eta^2}$ there exists a multiset of pure strategies for the MIN (row player) of size $k$ such that a mixed strategy $p_1$ that samples uniformly from this multiset satisfies*

$$\max_j \sum_i p_1(i)\mathcal{G}_{ij} \leq \mathcal{V}(\mathcal{G}) + \eta(\mathcal{G}_{max} - \mathcal{G}_{min}),$$

*where $\mathcal{G}_{max}, \mathcal{G}_{min}$ denote the maximum and minimum entry of $\mathcal{G}$ respectively. The symmetric result holds for the MAX player.*

We are ready to prove our main theorem.

**Theorem 5.** *Let $\epsilon \in \left(0, \frac{1}{2}\right), \delta \in \left(0, \frac{1}{48}\right), S : \mathbb{N} \to \mathbb{N}$. For every learning task $\mathbb{L} = \{\mathbb{L}_n\}_n$ learnable to error $\epsilon$ with confidence $1 - \delta$ and circuit complexity $O\left(\frac{\sqrt{S(n)}}{\log(S(n))}\right)$ and for every family of function classes $\mathcal{F} = \{\mathcal{F}_n\}_n$, every query bound $q(n)$ such that $\frac{\sqrt{S(n)}}{\log(S(n))} = \Omega(m(n) + q(n) \cdot n)$ at least one of the three*

$$\text{WATERMARK}\left(\mathbb{L}, \mathcal{F}, \epsilon, q, S(n), o\left(\frac{\sqrt{S(n)}}{\log(S(n))}\right), l = \frac{10}{24}, c = \frac{21}{24}, s = \frac{19}{24}\right),$$

$$\text{DEFENSE}\left(\mathbb{L}, \mathcal{F}, \epsilon, q, o\left(\frac{\sqrt{S(n)}}{\log(S(n))}\right), O(S(n)), l = 1 - \frac{1}{48}, c = \frac{13}{24}, s = \frac{11}{24}\right),$$

$$\text{TRANSFATTACK}\left(\mathbb{L}, \mathcal{F}, \epsilon, q, S(n), S(n), c = \frac{3}{24}, s = \frac{19}{24}\right)$$

*exists with frequency $\frac{1}{3}$.*

*Proof.* Let $\epsilon \in (0, \frac{1}{2})$ and $q : \mathbb{N} \to \mathbb{N}$ be a query bound. Let $\mathbb{L}$ be a learning task learnable to error $\epsilon$ with confidence $1 - \delta$ and complexity $S(n)$.

We will consider every $n$ separately and show that for every $n$, one of the three schemes exists. This automatically implies that one of the schemes exists with frequency at least $\frac{1}{3}$.

Let $s(n) = \Theta\left(\frac{\sqrt{S(n)}}{\log(S(n))}\right)$, where the exact constants will be determined later. Let $\mathfrak{Candidate}_{\mathfrak{W}}(n)$ be a set of $s(n)$-sized circuits computing $(f, \mathbf{x})$. Recall that the execution of a $\mathbf{A}_n \in \mathfrak{C}_{\mathfrak{W}}(n)$ proceeds by first sampling from $\text{Ex}(\mathcal{D}_n, h_n)$ and providing these samples as inputs to $\mathbf{A}_n$ and then running $\mathbf{A}_n$ to obtain $m + q \cdot n$ bits. The first $m$ bits are interpreted as a representation of $f$ (according to $\mathcal{R}_n$), and the following consecutive blocks of $n$ bits each are interpreted as $q$ elements of $\{0,1\}^n$. Similarly, let $\mathfrak{C}_{\mathfrak{D}}(n)$ be a set of $s(n)$-sized circuits accepting as input $(f, \mathbf{x})$ and outputting $(\mathbf{y}, b)$,

where $\mathbf{y} \in \{0,1\}^q, b \in \{0,1\}$. Formally, this is a set of circuits with up to $s(n)$ input gates and $q+1$ output gates. We interpret $\mathfrak{C}_{\mathfrak{W}}(n)$ as candidate algorithms for a watermark, and $\mathfrak{C}_{\mathfrak{D}}(n)$ as candidate algorithms for attacks on watermarks.

For every $n$ define a zero-sum game $\mathcal{G}_n$ between $\mathbf{A}_n \in \mathfrak{C}_{\mathfrak{W}}(n), \mathbf{B}_n \in \mathfrak{C}_{\mathfrak{D}}(n)$. The payoff is given by

$$\mathcal{G}_n(\mathbf{A}_n, \mathbf{B}_n) = \frac{1}{2} \, \mathbb{P}_{(\mathcal{D}_n, h_n) \sim \mathbb{L}_n, (f, \mathbf{x}) := \mathbf{A}_n^{\mathrm{Ex}(\mathcal{D}_n, h_n)}, (\mathbf{y}, b) := \mathbf{B}_n^{\mathrm{Ex}(\mathcal{D}_n, h_n)}} \left[ \mathrm{err}(f) > \epsilon \text{ or } \mathrm{err}(\mathbf{x}, \mathbf{y}) \leq 2\epsilon \text{ or } b = 1 \right]$$

$$+ \frac{1}{2} \, \mathbb{P}_{(\mathcal{D}_n, h_n) \sim \mathbb{L}_n, f := \mathbf{A}_n^{\mathrm{Ex}(\mathcal{D}_n, h_n)}, \mathbf{x} \sim \mathcal{D}_n^{q(n)}, (\mathbf{y}, b) := \mathbf{B}_n^{\mathrm{Ex}(\mathcal{D}_n, h_n)}} \left[ \mathrm{err}(f) > \epsilon \text{ or } \left( \mathrm{err}(\mathbf{x}, \mathbf{y}) \leq 2\epsilon \text{ and } b = 0 \right) \right],$$

where $\mathbf{A}_n$ tries to minimize and $\mathbf{B}_n$ maximize the payoff.

Then the number of possible circuits is bounded by

$$|\mathfrak{C}_{\mathfrak{W}}| \leq (3s(n)^2)^{s(n)} \leq 2^{3s(n) \log(s(n))},$$

because every internal gate of a circuit is one of AND, OR, and NOT, and is connected to 2 gates out of at most $s(n)$ choices.

Applying Theorem 4 to $\mathcal{G}_n$ with $\eta = 2^{-5}$ we get two probability distributions, $p$ over a multiset of pure strategies in $\mathfrak{C}_{\mathfrak{W}}$ and $r$ over a multiset of pure strategies in $\mathfrak{C}_{\mathfrak{D}}$ that lead to a $2^{-5}$-approximate Nash equilibrium. The size $k(n)$ of the multisets is bounded

$$\begin{aligned} k(n) &\leq 2^6 \log \left( |\mathfrak{C}_{\mathfrak{W}}| \right) \\ &\leq O(s(n) \log(s(n))). \end{aligned} \tag{1}$$

Next, observe that the mixed strategy corresponding to the distribution $p$ can be represented by a circuit of size

$$\begin{aligned} k(n) \cdot s(n) &\cdot O(\log(k(n))) \\ &\leq O(s^2(n) \cdot \log^3(s(n))) \qquad \text{By equation (1)} \\ &\leq S(n), \end{aligned}$$

because we can create a circuit that is a collection of $k(n)$ circuits corresponding to the multiset of $p$, where each one is of size $s(n)$ with additional gadgets of size $O(\log(k))$ activating the corresponding gate depending on the randomness determining a strategy. This implies that $p$ can be implemented by a $S(n)$-sized circuit. The same holds for $r$. Let's call the strategy corresponding to $p$, $\mathbf{A}_{\mathrm{Nash}}^n$, and the strategy corresponding to $r$, $\mathbf{B}_{\mathrm{Nash}}^n$.

Consider cases:

**Case $\mathcal{G}(\mathbf{A}_n^{\mathrm{NASH}}, \mathbf{B}_n^{\mathrm{NASH}}) \geq \frac{19}{24}$.** Define $\mathbf{B}_n^{\mathrm{DEFENSE}}$ to work as follows:

1. Simulate the circuit of size $O\left( \frac{\sqrt{S(n)}}{\log(S(n))} \right)$ $\mathbf{L}_n$ that learns $f$, such that

$$\mathbb{P}_{\substack{(\mathcal{D}_n, h_n) \sim \mathbb{L}_n, \\ f \leftarrow \mathbf{L}_n^{\mathrm{Ex}(\mathcal{D}_n, h_n)}}} \left[ \mathrm{err}(f) \leq \epsilon \right] \geq 1 - \frac{1}{48}.$$

2. Send $f$ to $\mathbf{A}_n$.

3. Receive $\mathbf{x}$ from $\mathbf{A}_n$.

4. Simulate $(\mathbf{y}, b) := \mathbf{B}_n^{\mathrm{NASH}}(f, \mathbf{x})$.

5. Return $b' = 1$ if $b = 1$ or $d(f(\mathbf{x}), \mathbf{y}) > 3\epsilon \cdot q(n)$ and $b' = 0$ otherwise,

where $d(\cdot, \cdot)$ is the Hamming distance. $\mathbf{B}_n^{\mathrm{DEFENSE}}$ can be implemented by circuit of size $O(S(n))$, because it simulates a circuit of size $O\left( \frac{\sqrt{S(n)}}{\log(S(n))} \right)$, then simulating $\mathbf{B}_n^{\mathrm{NASH}}$ of size $S(n)$, and computing a predicate $d(f(\mathbf{x}), \mathbf{y}) > 3\epsilon q$, which can be done in size $\log(q(n))$.

We claim that we have:

$$\text{DEFENSE}\left(\mathbb{L}_n, \mathcal{F}_n, \epsilon, q(n), o\left(\frac{\sqrt{S(n)}}{\log(S(n))}\right), O(S(n)), l = 1 - \frac{1}{48}, c = \frac{13}{24}, s = \frac{11}{24}\right). \qquad (2)$$

Assume towards contradiction that completeness or soundness of $\mathbf{B}_n^{\text{DEFENSE}}$ as defined in Definition 9 does not hold.

If completeness of $\mathbf{B}_n^{\text{DEFENSE}}$ does not hold, then

$$\mathbb{P}_{(\mathcal{D}_n, h_n) \sim \mathbb{L}_n, \mathbf{x} \sim \mathcal{D}_n^q}\left[b' = 0\right] < \frac{13}{24}. \qquad (3)$$

Let us compute the payoff of $\mathbf{A}_n$, which first runs $f \leftarrow \mathbf{L}_n^{\text{Ex}(\mathcal{D}_n, h_n)}$ (where $\mathbf{L}_n$ is the learning circuit) and sets $\mathbf{x} \sim \mathcal{D}^q$, in the game $\mathcal{G}_n$, when playing against $\mathbf{B}_n^{\text{NASH}}$

$$\mathcal{G}(\mathbf{A}_n, \mathbf{B}_n^{\text{NASH}})$$

$$= \frac{1}{2}\mathbb{P}_{\substack{(\mathcal{D}_n, h_n) \sim \mathbb{L}_n, \\ (f, \mathbf{x}) \leftarrow \mathbf{A}_n^{\text{Ex}(\mathcal{D}_n, h_n)}}}\left[\text{err}(f) > \epsilon \text{ or err}(\mathbf{x}, \mathbf{y}) \le 2\epsilon \text{ or } b' = 1\right]$$

$$+ \frac{1}{2}\mathbb{P}_{\substack{(\mathcal{D}_n, h_n) \sim \mathbb{L}_n, \\ f \leftarrow \mathbf{A}_n^{\text{Ex}(\mathcal{D}_n, h_n)}, \\ \mathbf{x} \sim \mathcal{D}_n^q}}\left[\text{err}(f) > \epsilon \text{ or } \left(\text{err}(\mathbf{x}, \mathbf{y}) \le 2\epsilon \text{ and } b' = 0\right)\right]$$

$$\le \delta + \frac{1}{2}\mathbb{P}_{\substack{(\mathcal{D}_n, h_n) \sim \mathbb{L}_n, \\ f \leftarrow \mathbf{L}_n^{\text{Ex}(\mathcal{D}_n, h_n)}, \\ \mathbf{x} \sim \mathcal{D}_n^q}}\left[\text{err}(\mathbf{x}, \mathbf{y}) \le 2\epsilon \text{ or } b' = 1\right]$$

$$+ \frac{1}{2}\mathbb{P}_{\substack{(\mathcal{D}_n, h_n) \sim \mathbb{L}_n, \\ f \leftarrow \mathbf{L}_n^{\text{Ex}(\mathcal{D}_n, h_n)}, \\ \mathbf{x} \sim \mathcal{D}_n^q}}\left[\text{err}(\mathbf{x}, \mathbf{y}) \le 2\epsilon \text{ and } b' = 0\right] \qquad \text{Def. of } \mathbf{A}_n, \mathbf{B}_n^{\text{DEFENSE}}, \mathbb{P}\left[\text{err}(f) \le \epsilon\right] \ge \frac{47}{48}$$

$$< \frac{1}{48} + \frac{1}{2} + \frac{\frac{13}{24}}{2} \qquad \text{By equation (3)}$$

$$= \frac{38}{48}$$

$$\le \mathcal{G}(\mathbf{A}_n^{\text{NASH}}, \mathbf{B}_n^{\text{NASH}}), \not{z}$$

where the contradiction is with the properties of Nash equilibria.

Assume that $\mathbf{A}_n$ breaks the soundness of $\mathbf{B}_n^{\text{DEFENSE}}$, which translates to

$$\mathbb{P}_{\substack{(\mathcal{D}_n, h_n) \sim \mathbb{L}_n, \\ \mathbf{x} \leftarrow \mathbf{A}_n(f)}}\left[\text{err}(\mathbf{x}, f(\mathbf{x})) > 7\epsilon \text{ and } b = 0 \text{ and } d(f(\mathbf{x}), \mathbf{y})) > 3\epsilon q\right] > \frac{11}{24}. \qquad (4)$$

Let $\mathbf{A}'_n$ first simulate $f \leftarrow \mathbf{L}_n^{\mathrm{Ex}(\mathcal{D}_n, h_n)}$, then runs $\mathbf{x} \leftarrow \mathbf{A}_n(f)$, and returns $(f, \mathbf{x})$. We have

$$
\begin{aligned}
&\mathcal{G}(\mathbf{A}'_n, \mathbf{B}_n^{\mathrm{NASH}}) \\
&= \frac{1}{2} \, \mathbb{P}_{\substack{(\mathcal{D}_n, h_n) \sim \mathbb{L}_n, \\ (f, \mathbf{x}) \leftarrow \mathbf{A}'_n}} \Big[ \mathrm{err}(f) > \epsilon \text{ or } \mathrm{err}(\mathbf{x}, \mathbf{y}) \le 2\epsilon \text{ or } b' = 1 \Big] \\
&\quad + \frac{1}{2} \, \mathbb{P}_{\substack{(\mathcal{D}_n, h_n) \sim \mathbb{L}_n, \\ f \leftarrow \mathbf{A}'_n, \\ \mathbf{x} \sim \mathcal{D}_n^q}} \Big[ \mathrm{err}(f) > \epsilon \text{ or } \Big( \mathrm{err}(\mathbf{x}, \mathbf{y}) \le 2\epsilon \text{ and } b' = 0 \Big) \Big] \\
&= \frac{1}{2} \, \mathbb{P}_{\substack{(\mathcal{D}_n, h_n) \sim \mathbb{L}_n, \\ f \leftarrow \mathbf{L}_n^{\mathrm{Ex}(\mathcal{D}_n, h_n)}, \\ \mathbf{x} = \mathbf{A}_n(f)}} \Big[ \mathrm{err}(f) > \epsilon \text{ or } \mathrm{err}(\mathbf{x}, \mathbf{y}) \le 2\epsilon \text{ or } b' = 1 \Big] \\
&\quad + \frac{1}{2} \, \mathbb{P}_{\substack{(\mathcal{D}_n, h_n) \sim \mathbb{L}_n, \\ f \leftarrow \mathbf{L}_n^{\mathrm{Ex}(\mathcal{D}_n, h_n)}, \\ \mathbf{x} \sim \mathcal{D}_n^q}} \Big[ \mathrm{err}(f) > \epsilon \text{ or } \Big( \mathrm{err}(\mathbf{x}, \mathbf{y}) \le 2\epsilon \text{ and } b' = 0 \Big) \Big] \qquad \text{By def. of } \mathbf{A}'_n \\
&< \frac{1}{2} + \frac{1 - \frac{11}{24}}{2} \qquad\qquad\qquad\qquad\qquad\qquad \text{By equation (4)} \\
&= \frac{37}{48} \\
&\le \mathcal{G}_n(\mathbf{A}_n^{\mathrm{NASH}}, \mathbf{B}_n^{\mathrm{NASH}}), \, \lightning
\end{aligned}
$$

where the contradiction is with the properties of Nash equilibria. Thus equation (2) holds.

**Case $\mathcal{G}_n(\mathbf{A}_n^{\mathrm{NASH}}, \mathbf{B}_n^{\mathrm{NASH}}) < \frac{19}{24}$.** Consider $\mathbf{B}_n$ that returns $(f(\mathbf{x}), b)$ for a uniformly random $b$. We have

$$
\mathcal{G}_n(\mathbf{A}_n^{\mathrm{NASH}}, \mathbf{B}_n) \ge \left( 1 - \mathbb{P}_{\substack{(\mathcal{D}_n, h_n) \sim \mathbb{L}_n, \\ f \leftarrow \mathbf{A}_n^{\mathrm{NASH}}}} \Big[ \mathrm{err}(f) \le \epsilon \Big] \right) + \mathbb{P}_{\substack{(\mathcal{D}_n, h_n) \sim \mathbb{L}_n, \\ f \leftarrow \mathbf{A}_n^{\mathrm{Nash}}}} \Big[ \mathrm{err}(f) \le \epsilon \Big] \cdot \frac{1}{2},
$$

because when $\mathbf{x} \sim \mathcal{D}_n^q$ and $\mathrm{err}(f) \le \epsilon$ the probability that $\mathrm{err}(\mathbf{x}, \mathbf{y}) \le 2\epsilon$ and $b = 0$ is $\frac{1}{2}$, and similarly when $\mathbf{x} \leftarrow \mathbf{A}_n^{\mathrm{NASH}}$ then the probability that $b = 1$ is equal $\frac{1}{2}$. The assumption that $\mathcal{G}_n(\mathbf{A}_n^{\mathrm{Nash}}, \mathbf{B}_n) < \frac{19}{24}$ and properties of Nash equilibria imply that $\mathbb{P}_{\substack{(\mathcal{D}_n, h_n) \sim \mathbb{L}_n, \\ f \leftarrow \mathbf{A}_n^{\mathrm{Nash}}}}[\mathrm{err}(f) \le \epsilon] \ge \frac{10}{24}$. This implies that *correctness* holds for $\mathbf{A}_n^{\mathrm{Nash}}$ with $l = \frac{10}{24}$.

Next, assume towards contradiction that *unremovability* of $\mathbf{A}_n^{\mathrm{NASH}}$ does not hold, i.e., there is $\mathbf{B}_n$ running in time $o\left( \sqrt{S(n)} / \log(S(n)) \right)$ such that $\mathbb{P}\big[ \mathrm{err}(\mathbf{x}, \mathbf{y}) \le 2\epsilon \big] > \frac{19}{24}$. Consider $\mathbf{B}'_n$ that on input $(f, \mathbf{x})$ returns $(\mathbf{B}_n(f, \mathbf{x}), 0)$. Then by definition of $\mathcal{G}_n$, $\mathcal{G}_n(\mathbf{A}_{\mathrm{NASH}}, \mathbf{B}'_n) > \frac{19}{24}$, which is a contradiction $\lightning$.

Next, assume towards contradiction that *undetectability* of $\mathbf{A}_n^{\mathrm{NASH}}$ does not hold, i.e., there exists $\mathbf{B}_n$ such that it distinguishes $\mathbf{x} \sim \mathcal{D}_n^q$ from $\mathbf{x} \leftarrow \mathbf{A}_n^{\mathrm{NASH}}$ with probability higher than $\frac{19}{24}$. Consider $\mathbf{B}'_n$ that on input $(f, \mathbf{x})$ returns $(f(\mathbf{x}), \mathbf{B}_n(f, \mathbf{x}))$.[11] Then by definition of $\mathcal{G}_n$, $\mathcal{G}_n(\mathbf{A}_n^{\mathrm{NASH}}, \mathbf{B}'_n) > \frac{19}{24}$, which is a contradiction $\lightning$.

There are two further subcases. If $\mathbf{A}_n^{\mathrm{NASH}}$ satisfies *uniqueness* then

$$
\mathbf{A}_n^{\mathrm{NASH}} \in \mathrm{WATERMARK}\left( \mathbb{L}_n, \mathcal{F}_n, \epsilon, q(n), S(n), o\left( \frac{\sqrt{S(n)}}{\log(S(n))} \right), l = \frac{10}{24}, c = \frac{21}{24}, s = \frac{19}{24} \right).
$$

If $\mathbf{A}_n^{\mathrm{NASH}}$ does not satisfy *uniqueness*, then, by definition, every succinctly representable circuit $\mathbf{B}_n$ of size $o\left( \sqrt{S(n)} / \log(S(n)) \right)$ satisfies $\mathrm{err}(\mathbf{x}, \mathbf{y}) \le 2\epsilon$ with probability at most $\frac{21}{24}$. Consider the following $\mathbf{A}_n$. It computes $(f, \mathbf{x}) \leftarrow \mathbf{A}_n^{\mathrm{Nash}}$, ignores $f$ and sends $\mathbf{x}$ to $\mathbf{B}_n$. By the assumption that *uniqueness* is not satisfied for $\mathbf{A}_n^{\mathrm{NASH}}$ *transferability* of Definition 5 holds for $\mathbf{A}_n$ with $c = \frac{3}{24}$. Note

---

[11]Formally $\mathbf{B}_n$ receives as input $(f, \mathbf{x})$ and not only $\mathbf{x}$.

that $\mathbf{B}_n$ in the transferable attack does not receive $f$ but it makes it no easier for it to satisfy the properties. Note that *undetectability* still holds with the same parameter. Thus

$$\mathbf{A}_n^{\text{NASH}} \in \text{TRANSFATTACK}\left(\mathbb{L}_n, \mathcal{F}_n, \epsilon, q(n), S(n), S(n), c = \frac{3}{24}, s = \frac{19}{24}\right).$$

$\square$

# E   Fully Homomorphic Encryption (FHE)

We include a definition of fully homomorphic encryption based on the definition from Goldwasser et al. [2013]. The notion of fully homomorphic encryption was first proposed by Rivest, Adleman and Dertouzos Rivest et al. [1978] in 1978. The first fully homomorphic encryption scheme was proposed in a breakthrough work by Gentry in 2009 Gentry [2009]. A history and recent developments on fully homomorphic encryption is surveyed in [Vaikuntanathan, 2011].

## E.1   Preliminaries

We say that a function $f$ is *negligible* in an input parameter $\lambda$, if for all $d > 0$, there exists $K$ such that for all $\lambda > K$, $f(\lambda) < \lambda^{-d}$. For brevity, we write: for all sufficiently large $\lambda$, $f(\lambda) = \text{negl}(\lambda)$. We say that a function $f$ is *polynomial* in an input parameter $\lambda$, if there exists a polynomial $p$ such that for all $\lambda$, $f(\lambda) \leq p(\lambda)$. We write $f(\lambda) = \text{poly}(\lambda)$. A similar definition holds for $\text{polylog}(\lambda)$. For two polynomials $p, q$, we say $p \leq q$ if for every $\lambda \in \mathbb{N}$, $p(\lambda) \leq q(\lambda)$.

When saying that a Turing machine $\mathcal{A}$ is p.p.t. we mean that $\mathcal{A}$ is a non-uniform probabilistic polynomial-time machine.

## E.2   Definitions

**Definition 11** (Goldwasser et al. [2013]).   A homomorphic (public-key) encryption scheme FHE is a quadruple of polynomial time algorithms (FHE.KEYGEN, FHE.ENC, FHE.DEC, FHE.EVAL) as follows:

- FHE.KEYGEN$(1^\lambda)$ is a probabilistic algorithm that takes as input the security parameter $1^\lambda$ and outputs a public key $pk$ and a secret key $sk$.

- FHE.ENC$(pk, x \in \{0, 1\})$ is a probabilistic algorithm that takes as input the public key $pk$ and an input bit $x$ and outputs a ciphertext $\psi$.

- FHE.DEC$(sk, \psi)$ is a deterministic algorithm that takes as input the secret key $sk$ and a ciphertext $\psi$ and outputs a message $x^* \in \{0, 1\}$.

- FHE.EVAL$(pk, C, \psi_1, \psi_2, \ldots, \psi_n)$ is a deterministic algorithm that takes as input the public key $pk$, some circuit $C$ that takes $n$ bits as input and outputs one bit, as well as $n$ ciphertexts $\psi_1, \ldots, \psi_n$. It outputs a ciphertext $\psi_C$.

**Compactness:** For all security parameters $\lambda$, there exists a polynomial $p(\cdot)$ such that for all input sizes $n$, for all $x_1, \ldots, x_n$, for all $C$, the output length of FHE.EVAL is at most $p(n)$ bits long.

**Definition 12** (*C-homomorphism, Goldwasser et al. [2013]*).   Let $C = \{C_n\}_{n \in \mathbb{N}}$ be a class of boolean circuits, where $C_n$ is a set of boolean circuits taking $n$ bits as input. A scheme FHE is $C$-homomorphic if for every polynomial $n(\cdot)$, for every sufficiently large security parameter $\lambda$, for every circuit $C \in C_n$, and for every input bit sequence $x_1, \ldots, x_n$, where $n = n(\lambda)$,

$$\mathbb{P}\left[\begin{array}{c} (pk, sk) \leftarrow \text{FHE.KEYGEN}(1^\lambda); \\ \psi_i \leftarrow \text{FHE.ENC}(pk, x_i) \text{ for } i = 1 \ldots n; \\ \psi \leftarrow \text{FHE.EVAL}(pk, C, \psi_1, \ldots, \psi_n) : \\ \text{FHE.DEC}(sk, \psi) \neq C(x_1, \ldots, x_n) \end{array}\right] = \text{negl}(\lambda),$$

where the probability is over the coin tosses of FHE.KEYGEN and FHE.ENC.

**Definition 13** (*Fully homomorphic encryption*).   A scheme FHE is fully homomorphic if it is homomorphic for the class of all arithmetic circuits over $\mathbb{GF}(2)$.

**Definition 14** (*Leveled fully homomorphic encryption*). A leveled fully homomorphic encryption scheme is a homomorphic scheme where FHE.KEYGEN receives an additional input $1^d$ and the resulting scheme is homomorphic for all depth-$d$ arithmetic circuits over $\mathbb{GF}(2)$.

**Definition 15** (*IND-CPA security*). A scheme FHE is IND-CPA secure if for any p.p.t. adversary $\mathcal{A}$,

$$\Big| \, \mathbb{P}\big[(pk, sk) \leftarrow \text{FHE.KEYGEN}(1^\lambda) : \mathcal{A}(pk, \text{FHE.ENC}(pk, 0)) = 1\big] +$$
$$- \mathbb{P}\big[(pk, sk) \leftarrow \text{FHE.KEYGEN}(1^\lambda) : \mathcal{A}(pk, \text{FHE.ENC}(pk, 1)) = 1\big] \Big| = \text{negl}(\lambda).$$

We now state the result of Brakerski, Gentry, and Vaikuntanathan [Brakerski et al., 2012] that shows a leveled fully homomorphic encryption scheme based on a standard assumption in cryptography called Learning with Errors [Regev, 2005]:

**Theorem 6** (*Fully Homomorphic Encryption, definition from Goldwasser et al. [2013]*). *Assume that there is a constant $0 < \epsilon < 1$ such that for every sufficiently large $\ell$, the approximate shortest vector problem gapSVP in $\ell$ dimensions is hard to approximate to within a $2^{O(\ell^\epsilon)}$ factor in the worst case. Then, for every $n$ and every polynomial $d = d(n)$, there is an IND-CPA secure $d$-leveled fully homomorphic encryption scheme where encrypting $n$ bits produces ciphertexts of length $poly(n, \lambda, d^{1/\epsilon})$, the size of the circuit for homomorphic evaluation of a function $f$ is $size(C_f) \cdot poly(n, \lambda, d^{1/\epsilon})$ and its depth is $depth(C_f) \cdot poly(\log n, \log d)$.*

## F  Existence of Transferable Attacks

**Learning Theory Preliminaries.** For the next lemma, we will consider a slight generalization of learning tasks to the case where there are many valid outputs for a given input. This can be understood as the case of generative tasks. More concretely, we assume that for the input space $\mathcal{X}_n$ the output space is $\mathcal{Y}_n$ instead of $\{0, 1\}$. It will always be the case that $\mathcal{X}_n$ and $\mathcal{Y}_n$ are equal to $\{0, 1\}^{p(n)}$ for some polynomial $p$. For a distribution $\mathcal{D}_n$ over $\mathcal{X}_n$ we call a function $h : \mathcal{X}_n \times \mathcal{Y}_n \to \{0, 1\}$ an error oracle if the error of a function $f : \mathcal{X}_n \to \mathcal{Y}_n$ is defined as

$$\text{err}(f) := \mathbb{E}_{x \sim \mathcal{D}}[h(x, f(x))],$$

where the randomness of expectation includes the potential randomness of $f$. The example oracle Ex provides access to samples $(x, y) \in \mathcal{X}_n \times \mathcal{Y}_n$, where $x \sim \mathcal{D}_n$ and $y \in \mathcal{Y}_n$ is some $y$ such that $h(x, y) = 0$.

The following learning task will be crucial for our construction.

**Definition 16** (*Lines on a Circle Learning Task $\mathbb{L}^\circ$*). We define $\mathbb{L}^\circ = \{\mathbb{L}_n^\circ\}_n$. For every $n$ we define $\mathcal{X}_n = \{0, 1\}^n$ and associate $\mathcal{X}_n$ with vertices of a $2^n$ regular polygon inscribed in the unit circle $\{x \in \mathbb{R}^2 \mid \|x\|_2 = 1\}$. The output space is $\{-1, +1\}$ for all $n$. Let $\mathcal{H} := \{h_w \mid w \in \mathbb{R}^2, \|w\|_2 = 1\}$, where $h_w(x) := \text{sgn}(\langle w, x \rangle)$. For every $n$, let $\mathbb{L}_n^\circ$ be the distribution corresponding to the following process: sample $h_w \sim U(\mathcal{H})$, return $(U(\mathcal{X}_n), h_w)$. Note that $\mathcal{H}$ has VC-dimension equal to 2 so $\mathbb{L}$ is learnable to error $\epsilon$ with $O(\frac{1}{\epsilon})$ samples for every $n$ and every $\epsilon$.

Moreover, for $n \in \mathbb{N}$ define $B_n^w(\alpha) := \{x \in \mathcal{X}_n \mid |\angle(x, w)| \leq \alpha\}$.

**Lemma 3** (*Learning lower bound for $\mathbb{L}^\circ$*). *Let $n \in \mathbb{N}$. Let $\mathbf{L}_n$ be a learning algorithm for $\mathbb{L}_n^\circ$ (Definition 16) that uses $K$ samples and returns a classifier $f : \mathcal{X}_n \to \{-1, +1\}$. Then*

$$\mathbb{P}_{(\mathcal{D}_n, h_n) \sim \mathbb{L}_n^\circ, f \leftarrow \mathbf{L}^{Ex(\mathcal{D}_n, h_n)}} \left[ \mathbb{P}_{x \sim \mathcal{D}_n}[f(x) \neq h_w(x)] \leq \frac{1}{2K} \right] \leq \frac{3}{100}.$$

*Proof.* Let $n \in \mathbb{N}$. Consider the following algorithm $\mathcal{A}$. It first simulates $\mathbf{L}_n$ on $K$ samples to compute $f$. Next, it performs a smoothing of $f$, i.e., computes

$$f_\eta(x) := \begin{cases} +1, & \text{if } \mathbb{P}_{x' \sim U(B_n^x(2\pi\eta))}[f(x') = +1] > \mathbb{P}_{x' \sim U(B_n^x(2\pi\eta))}[f(x') = -1] \\ -1, & \text{otherwise.} \end{cases}$$

Note that if $\text{err}(f) \leq \eta$ for a ground truth $h_w$ then for every $x \in \mathcal{X}_n \setminus B_n^x(2\pi\eta)$ we have $f_\eta(x) = h_w(x)$. This implies that $\mathcal{A}$ can be adapted to an algorithm that with probability 1 finds $w'$ such that $|\angle(w, w')| \leq \text{err}(f)$.

Assuming towards contradiction that the statement of the lemma does not hold it means that there is an algorithm using $K$ samples that with probability $\frac{3}{100}$ locates $w$ up to angle $\frac{1}{2K}$.

Consider any algorithm $\mathcal{A}$ using $K$ samples. Probability that $\mathcal{A}$ does not see any sample in $B_n^w(2\pi\eta)$ is at least

$$(1 - 4\eta)^K \geq \left( (1 - 4\eta)^{\frac{1}{4\eta}} \right)^{4\eta K} \geq \left( \frac{1}{2e} \right)^{4\eta K},$$

which is bigger than $1 - \frac{3}{100}$ if we set $\eta = \frac{1}{2K}$. But note that if there is no sample in $B_n^w(2\pi\eta)$ then $\mathcal{A}$ cannot locate $w$ up to $\eta$ with certainty. This proves the lemma. $\qquad\square$

**Lemma 4** (*Boosting for $\mathbb{L}^\circ$*)**.** *Let $\eta, \nu \in (0, \frac{1}{4}), n \in \mathbb{N}, \mathbf{L}_n$ be a learning algorithm for $\mathbb{L}_n^\circ$ that uses $K$ samples and outputs $f : \mathcal{X}_n \to \{-1, +1\}$ such that with probability $\delta$*

$$\mathbb{P}_{w\sim U(\mathcal{H}), x\sim U(B_n^w(2\pi\eta))}[f(x) \neq h_w(x)] \leq \nu, \tag{5}$$

*where $\mathcal{H}$ is as defined earlier $\{h_w \mid w \in \mathbb{R}^2, \|w\|_2 = 1\}$. Then there exists a learning algorithm $\mathbf{L}_n'$ for $\mathbb{L}_n^\circ$ that uses $\max\left(K, \frac{9}{\eta}\right)$ samples such that with probability $\delta - \frac{1}{1000}$ returns $f'$ such that*

$$\mathbb{P}_{w\sim U(\mathcal{H}), x\sim U(\mathcal{X}_n)}[f'(x) \neq h_w(x)] \leq 4\eta\nu.$$

*Proof.* Let $n \in \mathbb{N}$. $\mathbf{L}_n'$ first draws $\max\left(K, \frac{9}{\eta}\right)$ samples $Q$ and defines $g : \mathcal{X}_n \to \{-1, +1, \perp\}$ as follows, $g$ maps to $-1$ the smallest continuous interval containing all samples from $Q$ with label $-1$. Similarly $g$ maps to $+1$ the smallest continuous interval containing all samples from $Q$ with label $+1$. The intervals are disjoined by construction. Unmapped points are mapped to $\perp$. Next, $\mathbf{L}_n'$ simulates $\mathbf{L}_n$ with $K$ samples and gets a classifier $f$ that with probability $\delta$ satisfies the assumption of the lemma. Finally, it returns

$$f'(x) := \begin{cases} g(x), & \text{if } g(x) \neq \perp \\ f(x), & \text{otherwise.} \end{cases}$$

Consider 4 arcs defined as the 2 arcs constituting $B_n^w(2\pi\eta)$ divided into 2 parts each by the line $\{x \in \mathbb{R}^2 \mid \langle w, x \rangle = 0\}$. Let $E$ be the event that some of these intervals do not contain a sample from $Q$. Observe that

$$\mathbb{P}[E] \leq 4(1 - \eta)^{\frac{9}{\eta}} \leq \frac{1}{1000}.$$

By the union bound with probability $\delta - \frac{1}{1000}$, $f$ satisfies equation (5) and $E$ does not happen. By definition of $f'$ this gives the statement of the lemma. $\qquad\square$

**Theorem 7** (*Transferable Attack for a Cryptography based Learning Task*)**.** *There exists a learning task $\mathbb{L} = \{\mathbb{L}_\lambda\}_\lambda$ and a function class $\mathcal{F} = \{\mathcal{F}_\lambda\}_\lambda$ such that for every $\epsilon : \mathbb{N} \to \mathbb{N}$ where $1/\epsilon(\lambda)$ is lower-bounded by a sufficiently large polynomial and upper-bounded by some polynomial the following holds.*

*1. $\mathbb{L}$ is $\left(\epsilon, \delta = \frac{1}{10}, S = \frac{10^3}{\epsilon^{1.3}}, \mathcal{F}\right)$-learnable.*

*2. $\mathbb{L}$ is **not** $\left(\epsilon, \delta = \frac{1}{10}, S = \frac{1}{\epsilon}, \mathcal{F}\right)$-learnable*

*3. There exists a circuit family $\mathbf{A} = \{\mathbf{A}_\lambda\}_\lambda$ such that*

$$\mathbf{A} \in_1 \text{TRANSFATTACK} \begin{pmatrix} \mathbb{L}, \mathcal{F}, \epsilon(\lambda), \ q(\lambda) = \frac{16}{\epsilon(\lambda)}, \ S_{\mathbf{A}}(\lambda) = \frac{10^3}{\epsilon^{1.3}(\lambda)}, \\ S_{\mathbf{B}}(\lambda) = \frac{1}{10^2\epsilon^2(\lambda)}, \ c = \frac{9}{10}, \ s = negl(\lambda) \end{pmatrix}.$$

*Proof.* The learning task is based on $\mathbb{L}^\circ = \{\mathbb{L}_n^\circ\}_n$ from Definition 16.

**Setting of Parameters for FHE.** Observe that by assumption of the lemma $p \leq 1/\epsilon \leq r$, for some polynomial $r$, and a polynomial $p$ that we will define later. Let FHE be a fully homomorphic encryption scheme from Theorem 6. We will use the scheme for constant leveled circuits $d = O(1)$. Let $s(n, \lambda, d)$ be the polynomial bounding the size of the encryption of inputs of length $n$ with $\lambda$ security as well as bounding the size of the circuit for homomorphic evaluation, which is guaranteed to exist by Theorem 6. Let $\beta \in (0, 1)$ and $p$ be a polynomial such that

$$s\left(n^\beta, \lambda, d\right) \leq (n \cdot p(\lambda))^{0.1}, \tag{6}$$

which exist because $s$ is a polynomial.

We define $n(\lambda) := \lfloor p^{1/\beta}(\lambda) \rfloor^{12}$ for the length of inputs in the FHE scheme. Observe that for every $\lambda$

$$s(n(\lambda), \lambda, d) \leq (p(\lambda) \cdot p(\lambda))^{0.1} \qquad \text{By equation (6)}$$

$$\leq \frac{1}{\epsilon(\lambda)^{0.2}} \qquad \text{By } \epsilon(\lambda) \in \left(\frac{1}{r(\lambda)}, \frac{1}{p(\lambda)}\right). \tag{7}$$

**Learning Task.** The learning task will be parametrized by $\lambda$, i.e. $\mathbb{L} = \{\mathbb{L}_\lambda\}_\lambda$.

Let $\lambda \in \mathbb{N}$. We define $\mathbb{D}_\lambda := \{\mathcal{D}_\lambda^{(\text{pk,sk})}\}_{(\text{pk,sk})}$, $\mathcal{H}_\lambda := \{h_\lambda^{(\text{pk,sk,w})}\}_{(\text{pk,sk,w})}$ (for $\mathcal{D}_\lambda^{(\text{pk,sk})}$ and $h_\lambda^{(\text{pk,sk,w})}$ to be defined later), where they are indexed by valid public/secret key pairs of the FHE and $w \in \{x \in \mathbb{R}^2 \mid \|x\|_2 = 1\}$. Let $\mathbb{L}_\lambda$ be defined as corresponding to the following process: sample $(\text{pk,sk}, w) \sim \text{FHE.KEYGEN}(1^\lambda) \times U(\{x \in \mathbb{R}^2 \mid \|x\|_2 = 1\})$, return $\left(\mathcal{D}_\lambda^{(\text{pk,sk})}, h_\lambda^{(\text{pk,sk,w})}\right)$.

For a valid (pk,sk) pair we define $\mathcal{D}^{(\text{pk,sk})}$ as the result of the following process: $x \sim U(\{0, 1\}^{n(\lambda)})$, with probability $\frac{1}{2}$ return $(0, x, \text{pk})$ and with probability $\frac{1}{2}$ return $(1, \text{FHE.ENC}(\text{pk}, x), \text{pk})$, where the first element of the triple describes if the $x$ is encrypted or not. Formally, in the case that the first element of the triple is 0 one needs to add a padding of size $s(n(\lambda), \lambda, d) - n(\lambda)$ so that descriptions have the same size in both cases.[13]

For a valid (pk,sk) pair and $w \in \{x \in \mathbb{R}^2 \mid \|x\|_2 = 1\}$ we define $h^{(\text{pk,sk,w})}((b, x, \text{pk}), y)$ as a result of the following algorithm: if $b = 0$ return $\mathbb{1}_{h_w(x)=y}$, otherwise let $x_{\text{DEC}} \leftarrow \text{FHE.DEC}(\text{sk}, x), y_{\text{DEC}} \leftarrow \text{FHE.DEC}(\text{sk}, y)$ and if $x_{\text{DEC}}, y_{\text{DEC}} \neq \perp$ (decryption is succesful) return $\mathbb{1}_{h_w(x_{\text{DEC}})=y_{\text{DEC}}}$ and return 1 otherwise.

**Note 1** ($\Omega(\frac{1}{\epsilon})$-*sample learning lower bound.*). *By construction any learner using $K$ samples for $\mathbb{L}_\lambda$ (for any $\lambda$) can be transformed (potentially computationally inefficiently) into a learner using $K$ samples for $\mathbb{L}_{n(\lambda)}^\circ$ (Defnition 16) that returns a classifier of the same error. This, together with a lower bound for learning from Lemma 3 proves point 2 of the lemma.*

**Definition of A (Algorithm 1).** $\mathbf{A}_\lambda$ draws $N(\lambda)$ samples $Q = \{((b_i, x_i, \text{pk}), y_i)\}_{i \in [N]}$ for $N(\lambda) := \frac{900}{\epsilon(\lambda)}$.

Next, $\mathbf{A}_\lambda$ chooses a subset $Q_{\text{CLEAR}} \subseteq Q$ of samples for which $b_i = 0$. It trains a classifier $f_{w'}(\cdot) := \text{sgn}(\langle w', \cdot \rangle)$ on $Q_{\text{CLEAR}}$ by returning any $f_{w'}$ consistent with $Q_{\text{CLEAR}}$. This can be done in time

$$N(\lambda) \cdot n(\lambda) \leq \frac{900}{\epsilon(\lambda)} \cdot p^{1/\beta}(\lambda) \leq \frac{900}{\epsilon^{1.1}(\lambda)} \tag{8}$$

by keeping track of the smallest interval containing all samples in $Q_{\text{CLEAR}}$ labeled with $+1$ and then returning any $f_{w'}$ consistent with this interval.

**Note 2** ($O(\frac{1}{\epsilon^{1.3}})$-*time learning upper bound.*). *First note that $\mathbf{A}_\lambda$ learns well, i.e., with probability at least $1 - 2\left(1 - \frac{\epsilon(\lambda)}{100}\right)^{\frac{900}{\epsilon(\lambda)}} \geq 1 - \frac{1}{1000}$,*

$$|\angle(w, w')| \leq \frac{2\pi\epsilon(\lambda)}{100} \tag{9}$$

---

[12]Note that this setting allows to represent points in $\{x \in \mathbb{R}^2 \mid \|x\|_2 = 1\}$ up to $2^{-p^{1/\beta}(\lambda)}$ precision and this precision is better than $\frac{1}{r(\lambda)}$ for every polynomial $r$ for sufficiently large $\lambda$. This implies that this precision is enough to allow for learning up to error $\epsilon$, because of the setting $\epsilon(\lambda) \geq \frac{1}{r(\lambda)}$.

[13]Note that the domain of the distributions is not $\{0, 1\}^\lambda$, i.e. $\mathcal{X}_\lambda \neq \{0, 1\}^\lambda$.

---

**Algorithm 1** TRANSFATTACK(Ex$(\mathcal{D}_\lambda, h_\lambda), \epsilon, \lambda$)

---

1: **Input:** Access to the example oracle Ex$(\mathcal{D}_\lambda, h_\lambda)$, where $(\mathcal{D}_\lambda, h_\lambda) \sim \mathbb{L}_\lambda$, error level $\epsilon : \mathbb{N} \to \mathbb{N}$, and the security parameter $\lambda$.

2: $N := 900/\epsilon(\lambda), q := 16/\epsilon(\lambda)$
3: $Q = \{((b_i, x_i, \text{pk}), y_i)\}_{i \in [N]} \sim (\mathcal{D}_\lambda)^{N(\lambda)}$            ▷ $N(\lambda)$ i.i.d. samples from $\mathcal{D}_\lambda$
4: $Q_{\text{CLEAR}} = \{((b, x, \text{pk}), y) \in Q : b = 0\}$           ▷ $Q_{\text{CLEAR}} \subseteq Q$ of unencrypted $x$'s
5: $f_{w'}(\cdot) := \text{sgn}(\langle w', \cdot \rangle) \leftarrow$ a line consistent with samples from $Q_{\text{CLEAR}}$   ▷ $f_{w'} : \mathcal{X}_n \to \{-1, +1\}$
6: $\{x'_i\}_{i \in [q(\lambda)]} \sim U\left((\mathcal{X}_{n(\lambda)})^{q(\lambda)}\right)$
7: $S \sim U(2^{[q(\lambda)]})$                       ▷ $S \subseteq [q(\lambda)]$ a uniformly random subset
8: $E_{\text{BND}} ;= \emptyset$
9: **for** $i \in [q(\lambda) - |S|]$ **do**
10:     $x_{\text{BND}} \sim U(B^{w'}_{n(\lambda)}(2\pi(\epsilon(\lambda) + \frac{\epsilon(\lambda)}{100})))$       ▷ $x_{\text{BND}}$ is close to the decision boundary of $f_{w'}$
11:     $E_{\text{BND}} := E_{\text{BND}} \cup \{\text{FHE.ENC}(\text{pk}, x_{\text{BND}})\}$
12: **end for**
13: $\mathbf{x} := \{(0, x'_i, \text{pk}) \mid i \in [q(\lambda)] \setminus S\} \cup \{(1, x', \text{pk}) \mid x' \in E_{\text{BND}}\}$

14: **Return x**

---

*Moreover, $f_{w'}(x)$ can be implemented by a circuit $C_{f_{w'}}$ that compares $x$ with the endpoints of the interval. This can be done by a constant leveled circuit. Moreover $C_{f_{w'}}$ can be evaluated with* FHE.EVAL *in time*

$$size(C_{f_{w'}})s(n(\lambda), \lambda, d) \leq 10n \cdot s(n(\lambda), \lambda, d) \leq 10p^{1/\beta}(\lambda)s(n(\lambda), \lambda, d) \leq \frac{10}{\epsilon^{0.3}(\lambda)},$$

*where the last inequality follows from equation (7). This proves point 1 of the lemma.*

Next, $\mathbf{A}_\lambda$ prepares $\mathbf{x}$ as follows. It samples $q(\lambda) = \frac{16}{\epsilon(\lambda)}$ points $\{x'_i\}_{i \in [q]}$ from $\{0, 1\}^{n(\lambda)}$ uniformly at random. It chooses a uniformly random subset $S \subseteq [q(\lambda)]$. Next, $\mathbf{A}_\lambda$ generates $q(\lambda) - |S|$ inputs using the following process: $x_{\text{BND}} \sim U(B^{w'}_{n(\lambda)}(2\pi(\epsilon(\lambda) + \frac{\epsilon(\lambda)}{100})))$ ($x_{\text{BND}}$ is close to the decision boundary of $f_{w'}$), return FHE.ENC$(\text{pk}, x_{\text{BND}})$. Call the set of $q(\lambda) - |S|$ points $E_{\text{BND}}$. $\mathbf{A}_\lambda$ defines:

$$\mathbf{x} := \{(0, x'_i, \text{pk}) \mid i \in [q] \setminus S\} \cup \{(1, x', \text{pk}) \mid x' \in E_{\text{BND}}\}.$$

The running time of this phase is dominated by evaluations of FHE.EVAL, which takes

$$q(\lambda) \cdot s(n(\lambda), \lambda, d) \leq \frac{16}{\epsilon(\lambda)} \cdot \frac{1}{\epsilon^{0.2}(\lambda)} \leq \frac{16}{\epsilon^{1.2}(\lambda)}, \tag{10}$$

where the first inequality follows from equation (7). Taking the sum of equation (8) and equation (10) we get that $\mathbf{A}_\lambda$ can be implemented by a circuit of size $\frac{10^3}{\epsilon^{1.3}(\lambda)}$.

**$\mathbf{A}_\lambda$ Constitutes a Transferable Attack.** Now, consider $\mathbf{B}_\lambda$ of size $S_{\mathbf{B}}(\lambda) = \frac{1}{\epsilon^2(\lambda)}$. By the assumption $S_{\mathbf{B}}(\lambda) \leq r(\lambda)$, which implies that the security guarantees of FHE hold for $\mathbf{B}_\lambda$.

We claim that $\mathbf{x}$ is indistinguishable from $\mathcal{D}_\lambda^{(\text{pk,sk})}$ for $\mathbf{B}_\lambda$. Observe that by construction the distribution of ratio of encrypted and not encrypted $x$'s in $\mathbf{x}$ is identical to that of $\mathcal{D}_\lambda^{(\text{pk,sk})}$. Moreover, the distribution of unencrypted $x$'s is identical to that of $\mathcal{D}_\lambda^{(\text{pk,sk})}$ by construction. Finally, by the IND-CPA security[14] of FHE and the fact that the size of $\mathbf{B}_\lambda$ is bounded by some polynomial in $\lambda$, FHE.ENC$(\text{pk}, x_{\text{BND}})$ is distinguishable from $x \sim \mathcal{X}_n$, FHE.ENC$(\text{pk}, x)$ with advantage at most negl$(\lambda)$. Thus *undetectability* holds with near perfect soundness $s = \frac{1}{2} + \text{negl}(\lambda)$.

Next, we claim that $\mathbf{B}_\lambda$ can't return low-error answers on $\mathbf{x}$.

Assume towards contradiction that with probability $\frac{5}{100}$

$$\mathbb{P}_{\substack{w \sim U(\{z \in \mathbb{R}^2 \mid \|z\|_2 = 1\}), \\ x \sim U(B^w_{n(\lambda)}(2\pi\epsilon(\lambda)))}}[f(x) \neq h_w(x)] \leq 10\epsilon(\lambda). \tag{11}$$

---

[14]Note that we need security of FHE in the nonuniform model of computation.

We can apply Lemma 4 to get that there exists a learner using $\frac{1}{100\epsilon^2(\lambda)} + \frac{9}{\epsilon(\lambda)} \leq \frac{1}{90\epsilon^2(\lambda)}$ samples that with probability $\frac{4}{100}$ returns $f'$ such that

$$\mathbb{P}_{\substack{w \sim U(\{z \in \mathbb{R}^2 \ | \ \|z\|_2 = 1\}), \\ x \sim U(\{0,1\}^{n(\lambda)})}}[f'(x) \neq h_w(x)] \leq 40\epsilon^2(\lambda). \tag{12}$$

Applying Lemma 3 to equation (12) we know that

$$40\epsilon^2 \geq \frac{1}{2(\frac{1}{90\epsilon^2(\lambda)})},$$

which is a contradiction. Thus equation (11) does not hold and in consequence, using equation (9), with probability $1 - \frac{6}{100}$

$$\mathbb{P}_{\substack{w \sim U(\{z \in \mathbb{R}^2 \ | \ \|z\|_2 = 1\}), \\ x \sim U(B_{n(\lambda)}^{w'}(2\pi(\epsilon(\lambda) + \frac{\epsilon(\lambda)}{10})))}} [f(x) \neq h_w(x)] \geq \frac{10}{14} \cdot 10\epsilon(\lambda) \geq 7\epsilon(\lambda), \tag{13}$$

where crucially $x$ is sampled from $U(B_{n(\lambda)}^{w'})$ and not $U(B_{n(\lambda)}^{w})$. By Fact 1 we know that $|S| \geq \frac{q(\lambda)}{3}$ with probability at least

$$1 - 2e^{-\frac{q(\lambda)}{72}} = 1 - 2e^{-\frac{1}{8\epsilon(\lambda)}} \geq 1 - \frac{1}{1000}.$$

Using the setting of $q(\lambda) = \frac{16}{\epsilon(\lambda)}$ and applying the Chernoff bound and the union bound we get from equation (13) that with probability at least $1 - \frac{1}{10}$ the error $\mathrm{err}(\mathbf{x}, \mathbf{y})$ is larger than $2\epsilon(\lambda)$.

$\square$

**Note 3.** *We want to emphasize that it is crucial (for our construction) that the distribution has both an encrypted and an unencrypted part.*

*As mentioned before, if there was no $\mathcal{D}_{\mathrm{CLEAR}}$ then $\mathbf{A}_\lambda$ would see only samples of the form*

$$(\mathrm{FHE.ENC}(x), \mathrm{FHE.ENC}(y))$$

*and would not know which of them lie close to the boundary of $h_w$, and so it would not be able to choose tricky samples. $\mathbf{A}_\lambda$ would be able to learn a low-error classifier, but only under the encryption. More concretely, $\mathbf{A}_\lambda$ would be able to homomorphically evaluate a circuit that, given a training set and a test point, learns a good classifier and classifies the test point with it. However, it would* not *be able to, with high probability, generate $\mathrm{FHE.ENC}(x)$, for $x$ close to the boundary as it would not know (in the clear) where the decision boundary is.*

*If there was no $\mathcal{D}_{\mathrm{ENC}}$ then everything would happen in the clear and so $\mathbf{B}$ would be able to distinguish $x$'s that appear too close to the boundary.*

**Fact 1** (*Chernoff-Hoeffding*). Let $X_1, \ldots, X_k$ be independent Bernoulli variables with parameter $p$. Then for every $0 < \epsilon < 1$

$$\mathbb{P}\left[\left|\frac{1}{k}\sum_{i=1}^{k} X_i - p\right| > \epsilon\right] \leq 2e^{-\frac{\epsilon^2 k}{2}}$$

and

$$\mathbb{P}\left[\frac{1}{k}\sum_{i=1}^{k} X_i \leq (1-\epsilon)p\right] \leq e^{-\frac{\epsilon^2 kp}{2}}.$$

Also for every $\delta > 0$

$$\mathbb{P}\left[\frac{1}{k}\sum_{i=1}^{k} X_i > (1+\delta)p\right] \leq e^{-\frac{\delta^2 kp}{2+\delta}}$$

# G  Transferable Attacks Imply Cryptography

## G.1  EFID Pairs

The typical way in which security of EFID pairs is defined, e.g., in [Goldreich, 1990], is that they should be secure against all polynomial-time algorithms. However, for the case of pseudorandom generators (PRGs), which are known to be equivalent (in the standard definition) to EFIDs pairs, more granular notions of security were considered. For instance, in [Nisan, 1990] the existence of PRGs secure against adversaries running in time bounded by a fixed, in contrast to all, polynomial, was studied. In a similar spirit, we consider EFID pairs that are secure against adversaries with fixed circuit complexity bounds.

**Definition 17** (*Total Variation*). For two distrbutions $\mathcal{D}_0, \mathcal{D}_1$ over a finite domain $\{0,1\}^n$ we define their *total variation distance* as

$$\triangle(\mathcal{D}_0, \mathcal{D}_1) := \sum_{x \in \{0,1\}^n} \frac{1}{2} |\mathcal{D}_0(x) - \mathcal{D}_1(x)|.$$

**Definition 18** (*EFID pairs*). For parameters $\eta, \delta : \mathbb{N} \to (0,1)$ and circuit complexity bounds $S, S' : \mathbb{N} \to \mathbb{N}$ we call a pair of ensembles of distributions $(\mathcal{D}^0 = \{\mathcal{D}_n^0\}_n, \mathcal{D}^1 = \{\mathcal{D}_n^1\}_n)$ over domain $\mathcal{X} = \{\mathcal{X}_n\}_n$ an $(S, S', \eta, \delta)$-EFID pair if for every $n$

1. The circuit complexity of sampling $\mathcal{D}^0$ and $\mathcal{D}^1$ is at most $S$,

2. For every $n$, $\triangle(\mathcal{D}_n^0, \mathcal{D}_n^1) \geq \eta(n)$,

3. For every $n$, $\mathcal{D}_n^0, \mathcal{D}_n^1$ are $\delta(n)$-indistinguishable for circuits with complexity $S'(n)$.

Observe that Definition 18 is a generalization of the standard definition. Indeed, for every EFID pair $(\mathcal{D}^0, \mathcal{D}^1)$ according to the standard definition there exists an inverse polynomial function $\eta$ and a polynomial $S$ such that for all polynomials $S'$ there exists a negligible function $\delta$ such that $(\mathcal{D}^0, \mathcal{D}^1)$ is an $(S, S', \eta, \delta)$-EFID pair.

## G.2  Transferable Attacks imply EFID pairs

**Theorem 8** (*Tasks with Transferable Attacks Imply EFID pairs*). *For every* $\epsilon \in (0,1), q \in \mathbb{N}, S_\mathbf{A}, S_\mathbf{B} : \mathbb{N} \to \mathbb{N}$ *such that* $S_\mathbf{A} \leq S_\mathbf{B}$, *every learning task* $\mathbb{L}$ *learnable to error* $\epsilon$ *with confidence* $p$ *and circuit complexity* $S_\mathbf{A}$, *every* $c, s \in (0,1)$ *if*

$$\textsc{TransfAttack}\left(\mathbb{L}, \epsilon, q, S_\mathbf{A}, S_\mathbf{B}, c, s\right)$$

*exists with frequency* $\frac{1}{3}$ *then there exist* $S_\mathbf{A}', S_\mathbf{B}' : \mathbb{N} \to \mathbb{N}$ *that agree with* $S_\mathbf{A}$ *and* $S_\mathbf{B}$ *respectively with frequency* $\frac{1}{3}$ *and there exists*

$$\left(S_\mathbf{A}', S_\mathbf{B}', \frac{1}{2}\left(p + c - 1 - e^{-\frac{\epsilon q}{3}}\right), \frac{s}{2}\right) - \textit{EFID pair.}$$

*Proof.* Let $\epsilon, S_\mathbf{A}, S_\mathbf{B}, q, c, s, p, \mathbb{L}$ be as in the assumption of the theorem. Additionally let $\mathbf{A} = \{\mathbf{A}_n\}_n$ be a family of circuits certifying that a Transferable Attack exists with frequency $\frac{1}{3}$ for $\mathbb{L}$.

For every $n$, define $\mathcal{D}_n^0 := \mathcal{D}_n^q$, where we recall that $q$ is the number of samples $\mathbf{A}_n$ sends in the attack. Define $\mathcal{D}_n^1$ to be the distribution of $\mathbf{x} := \mathbf{A}_n$. Note that $\mathbf{x} \in (\mathcal{X}_n)^q$.

Let $a : \mathbb{N} \to \{0,1\}$ be a sequence certifying that a Transferable Attack exists with frequency $\frac{1}{3}$. Let $n$ be such that $a(n) = 1$. Observe that $\mathcal{D}_n^0, \mathcal{D}_n^1$ are samplable with circuit complexity $S_\mathbf{A}(n)$ because $\mathbf{A}_n$ complexity is bounded by $S_\mathbf{A}(n)$. Secondly, $\mathcal{D}_n^0, \mathcal{D}_n^1$ are $\frac{s}{2}$-indistinguishable for $S_\mathbf{B}(n)$-sized adversaries by *undetectability* of $\mathbf{A}_n$. Finally, the fact that $\mathcal{D}_n^0, \mathcal{D}_n^1$ are statistically far follows from *transferability*. Indeed, the following procedure accepting input $\mathbf{x} \in (\{0,1\}^n)^q$ is a distinguisher:

1. Run the learner (the existence of which is guaranteed by the assumption of the theorem) to obtain $f$.

2. $\mathbf{y} := f(\mathbf{x})$.

3. If $\text{err}(\mathbf{x}, \mathbf{y}) \leq 2\epsilon$ return 0, otherwise return 1.

If $\mathbf{x} \sim \mathcal{D}^0 = \mathcal{D}^q$ then $\text{err}(f) \leq \epsilon$ with probability $p$. By Fact 1 and the union bound we also know that $\text{err}(\mathbf{x}, \mathbf{y}) \leq 2\epsilon$ with probability $p - e^{-\frac{\epsilon q}{3}}$ and so, the distinguisher will return 0 with probability $p - e^{-\frac{\epsilon q}{3}}$. On the other hand, if $\mathbf{x} \sim \mathcal{D}^1 = \mathbf{A}$ we know from *transferability* of $\mathbf{A}_n$ that every algorithm running in time $S_\mathbf{B}(n)$ will return $\mathbf{y}$ such that $\text{err}(\mathbf{x}, \mathbf{y}) > 2\epsilon$ with probability at least $c$. By the assumption that $S_\mathbf{B}(n) \geq S_\mathbf{A}(n)$ we know that $\text{err}(\mathbf{x}, f(\mathbf{x})) > 2\epsilon$ with probability at least $c$ also. Consequently, the distinguisher will return 1 with probability at least $c$ in this case. By the properties of total variation this implies that $\triangle(\mathcal{D}_n^0, \mathcal{D}_n^1) \geq \frac{1}{2}(p + c - 1 - e^{-\frac{\epsilon q}{3}})$.

We define a pair of families of distributions $\widehat{\mathcal{D}}^0, \widehat{\mathcal{D}}^1$ and functions $S'_\mathbf{A}, S'_\mathbf{B}$ as follows. For every $n$ such that $a(n) = 1$ we define $\widehat{\mathcal{D}}_n^0 = \mathcal{D}_n^0, \widehat{\mathcal{D}}^1 = \mathcal{D}_n^1, S'_\mathbf{A}(n) = S_\mathbf{A}(n), S'_\mathbf{B}(n) = S_\mathbf{B}(n)$. For every $n$ sich that $a(n) = 0$ we define $\widehat{\mathcal{D}}_n^0 = \mathcal{D}_k^0$ for the smallest $k > n$ such that $a(k) = 1$, and $S'_\mathbf{A}(n) = S_\mathbf{A}(k)$ And analogously for $\widehat{\mathcal{D}}_n^1$ and $S'_\mathbf{B}$.

Simple verification yields that $\widehat{\mathcal{D}}_n^0, \widehat{\mathcal{D}}_n^1$ is an $(S'_\mathbf{A}, S'_\mathbf{B}, \frac{1}{2}(p + c - 1 - e^{-\frac{\epsilon q}{3}}), \frac{s}{2})$-EFID pair.

**Note 4** (*Setting of parameters*). *Observe that if $p \approx 1$, i.e., it is possible to almost surely learn $f$ in time $S_\mathbf{A}$ such that $\text{err}(f) \leq \epsilon$, $c$ is a constant, $q = \Omega(\frac{1}{\epsilon})$ then $\eta$ in the parameters for the EFID is a constant and so $\triangle(\mathcal{D}^0, \mathcal{D}^1)$ is a constant.*

**Note 5.** *We want to emphasize that our distinguisher crucially uses the error oracle in its last step. So it is possible that it is not implementable for all circuit complexity bounds!*

$\square$

# H  Adversarial Defenses exist

Our result is based on [Goldwasser et al., 2020]. Before we state and prove our result we give an overview of the learning model considered in [Goldwasser et al., 2020]. The authors give a defense against *arbitrary examples* in a transductive model with rejections. In contrast, our model does not allow rejections, but we do require indistinguishability.

## H.1  Transductive Learning With Rejections.

In [Goldwasser et al., 2020] the authors consider a model, where a learner $\mathbf{L}$ receives a training set of labeled samples from the original distribution $(\mathbf{x}_\mathcal{D}, \mathbf{y}_\mathcal{D} = h(\mathbf{x}_\mathcal{D})), \mathbf{x} \sim \mathcal{D}^N, \mathbf{y}_\mathcal{D} \in \{-1, +1\}^N$, where $h$ is the ground truth, together with a test set $\mathbf{x}_T \in (\{0, 1\}^n)^q$. Next, $\mathbf{L}$ uses $(\mathbf{x}_\mathcal{D}, \mathbf{y}_\mathcal{D}, \mathbf{x}_T)$ to compute $\mathbf{y}_T \in \{-1, +1, \square\}^q$, where $\square$ represents that $\mathbf{L}$ abstains (rejects) from classifying the corresponding $x$.

Before we define when learning is successful, we will need some notation. For $q \in \mathbb{N}, \mathbf{x} \in (\{0, 1\}^n)^q, \mathbf{y} \in \{-1, +1, \square\}^q$ we define

$$\text{err}(\mathbf{x}, \mathbf{y}) := \frac{1}{q} \sum_{i \in [q]} \mathbb{1}_{\left\{h(x_i) \neq y_i, y_i \neq \square, h(x_i) \neq \perp\right\}}, \quad \square(\mathbf{y}) := \frac{1}{q}\left|\left\{i \in [q] : y_i = \square\right\}\right|,$$

which means that we count $(x, y) \in \{0, 1\}^n \times \{-1, +1, \square\}$ as an error if $h$ is well defined on $x$, $y$ is not an abstention and $h(x) \neq y$.

Learning is successful if it satisfies two properties.

- If $\mathbf{x}_T \sim \mathcal{D}^q$ then with high probability $\text{err}(\mathbf{x}_T, \mathbf{y}_T)$ and $\square(\mathbf{y}_T)$ are small.
- For *every* $\mathbf{x}_T \in (\{0, 1\}^n)^q$ with high probability $\text{err}(\mathbf{x}_T, \mathbf{y}_T)$ is small.[15]

---

[15]Note that, crucially, in this case $\square(\mathbf{y}_T)$ might be very high, e.g., equal to 1.

The formal guarantee of a result from Goldwasser et al. [2020] are given in Theorem 9. Let's call this model Transductive Learning with Rejections (TLR).

Note the differences between TLR and our definition of Adversarial Defenses. To compare the two models we associate the learner $\mathbf{L}$ from TLR with $\mathbf{B}$ in our setup, and the party producing $\mathbf{x}_T$ with $\mathbf{A}$ in our definition. First, in TLR, $\mathbf{B}$ does not send $f$ to $\mathbf{A}$. Secondly, and most importantly, we do not allow $\mathbf{B}$ to reply with rejections ($\square$) but instead require that $\mathbf{B}$ can "distinguish" that it is being tested (see soundness of Definition 9). Finally, there are no apriori time bounds on either $\mathbf{A}$ or $\mathbf{B}$ in TLR. The models are similar but a priori incomparable and any result for TLR needs to be carefully analyzed before being used to prove that it is an Adversarial Defense.

### H.2 Formal guarantee for Transductive Learning with Rejections (TLR)

Theorem 5.3 from Goldwasser et al. [2020] adapted to our notation reads.

**Theorem 9** (*TLR guarantee (Goldwasser et al. [2020])*). *For any $N \in \mathbb{N}, \epsilon \in (0,1), h \in \mathcal{H}$ and distribution $\mathcal{D}$ over $\{0,1\}^n$:*

$$\mathbb{P}_{\mathbf{x}_{\mathcal{D}}, \mathbf{x}'_{\mathcal{D}} \sim \mathcal{D}^N} \left[ \forall \, \mathbf{x}_T \in \{0,1\}^{n N} : err(\mathbf{x}_T, f(\mathbf{x}_T)) \leq \epsilon^* \wedge \square \left( f \left( \mathbf{x}'_{\mathcal{D}} \right) \right) \leq \epsilon^* \right] \geq 1 - \epsilon,$$

*where $\epsilon^* = \sqrt{\frac{2d}{N} \log(2N) + \frac{1}{N} \log \left( \frac{1}{\epsilon} \right)}$ and $f = \text{REJECTRON}(\mathbf{x}_{\mathcal{D}}, h(\mathbf{x}_{\mathcal{D}}), \mathbf{x}_T, \epsilon^*)$, where $f : \{0,1\}^n \to \{-1, +1, \square\}$ and $d$ denotes the VC-dimension on $\mathcal{H}$. REJECTRON is defined in Figure 2. in [Goldwasser et al., 2020].*

REJECTRON is an algorithm that accepts a labeled training set $(\mathbf{x}_{\mathcal{D}}, h(\mathbf{x}_{\mathcal{D}}))$ and a test set $\mathbf{x}_T$ and returns a classifier $f$, which might reject some inputs. The learning is successful if with a high probability $f$ rejects a small fraction of $\mathcal{D}^N$ and for every $\mathbf{x}_T \in \{0,1\}^{n N}$ the error on labeled $x$'s in $\mathbf{x}_T$ is small.

### H.3 Adversarial Defense for bounded VC-dimension

We are ready to state the main result of this section.

**Lemma 5** (*Adversarial Defense for bounded VC-dimension*). *Let $\{\mathcal{H}_n\}_n$ be a family of hypothesis classes such that there exists a polynomial $p$ such that for every $n$, $\mathcal{H}_n$ has a VC-dimension bounded by $p(n)$. There exists a family of circuits $\mathbf{B} = \{\mathbf{B}_n\}_n$ such that for every $\mathbb{L}$ satisfying for every $n$ that the support of the marginal of $\mathbb{L}_n$ is contained in $\mathcal{H}_n$, i.e., the ground truth sampled from $\mathbb{L}$ are always in $\mathcal{H}$, such that for every sufficiently small $\epsilon$,*

$$\mathbf{B} \in_1 \text{DEFENSE} \left( \mathbb{L}, \epsilon, q = \frac{\texttt{poly}(n)}{\epsilon^3}, S_{\mathbf{A}} = \infty, S_{\mathbf{B}} = \texttt{poly} \left( \frac{n}{\epsilon} \right), l = 1 - \epsilon, c = 1 - \epsilon, s = \epsilon \right).$$

Note that, by the PAC learning bound, this is a setting of parameters, where $\mathbf{B}$ has enough time to learn a classifier of error $\epsilon$. By slightly abusing the notation, we write $S_{\mathbf{A}} = \infty$, meaning that the defense is secure against *all* adversaries regardless of their running time.

*Proof.* The proof is based on an algorithm from Goldwasser et al. [2020].

**Construction of B.** Let $\epsilon \in (0,1), n \in N, d(n)$ be the VC-dimension of $\mathcal{H}_n$ and

$$N := \frac{d \log^2(d)}{\epsilon^3}.$$

Let $q := N$. First, $\mathbf{B}$, draws $N$ labeled samples $(\mathbf{x}_{\text{FRESH}}, h(\mathbf{x}_{\text{FRESH}}))$. Next, it finds $f \in \mathcal{H}$ consistent with them and sends $f$ to $\mathbf{A}$. Importantly this computation is the same as the first step of REJECTRON.

Next, $\mathbf{B}$ receives as input $\mathbf{x} \in (\{0,1\}^n)^q$ from $\mathbf{A}$. $\mathbf{B}$. Let $\epsilon^* := \sqrt{\frac{2d}{N} \log(2N) + \frac{1}{N} \log \left( \frac{1}{\epsilon} \right)}$. Next $\mathbf{B}$ runs $f' = \text{REJECTRON}(\mathbf{x}_{\text{FRESH}}, h(\mathbf{x}_{\text{FRESH}}), \mathbf{x}, \epsilon^*)$, where REJECTRON is starting from the second step of the algorithm (Figure 2 [Goldwasser et al., 2020]). Importantly, for every $x \in \{0,1\}^n$, if $f'(x) \neq \square$ then $f(x) = f'(x)$. In words, $f'$ is equal to $f$ everywhere where $f'$ does not reject.

Finally $\mathbf{B}$ returns 1 if $\square(f'(\mathbf{x})) > \frac{2}{3} \epsilon$, and returns 0 otherwise.

**B is a Defense.**   First, by the standard PAC theorem, with probability at least $1 - \epsilon$, $\text{err}(f) \leq \frac{\epsilon}{2}$. This means that *correctness* holds with probability $l = 1 - \epsilon$.

Note that with our setting of $N$,

$$\epsilon^* \leq \frac{\epsilon}{2}.$$

Theorem 9 guarantees that

- if $\mathbf{x} \in \mathcal{D}^q$ then with probability at least $1 - \epsilon$,

$$\square(f'(\mathbf{x})) \leq \frac{\epsilon}{2}.$$

  which in turn implies that with the same probability $\mathbf{B}$ returns $b = 0$. This implies that *completeness* holds with probability $1 - \epsilon$.

- for every $\mathbf{x} \in (\{0,1\}^n)^q$ with probability at least $1 - \epsilon$,

$$\text{err}(\mathbf{x}, f'(\mathbf{x})) \leq \frac{\epsilon}{2}.$$

  To compute soundness we want to upper bound the probability that $\text{err}(\mathbf{x}, f(\mathbf{x})) > 2\epsilon$[16] and $b = 0$. By construction of $\mathbf{B}$ if $b = 0$ then $\square(f'(\mathbf{x})) \leq \frac{2\epsilon}{3}$, which means that with probability at least $1 - \epsilon$

$$\text{err}(\mathbf{x}, \mathbf{y}) \leq \frac{2\epsilon}{3} + \frac{\epsilon}{2} < 2\epsilon \text{ or } b = 1.$$

  This translates to *soundness* holding with $s = \epsilon$.

REJECTRON can be implemented by a circuit of size polynomial in $N$ and makes $O(\frac{1}{\epsilon})$ calls to an Empirical Risk Minimizer on $\mathcal{H}$ (that we assume can be implemented by a circuit of size polynomial in $d$), which implies the promised circuit complexity. $\square$

## I   Watermarks exist

**Lemma 6** (*Watermark for bounded VC-dimension against fast adversaries*)**.** *There exists a family of hypothesis classes $\{\mathcal{H}_d\}_d$ such that for every $d$, $\mathcal{H}_d$ has VC-dimension $d$ and a family of distributions $\{\mathcal{D}_d\}_d$ such that for every $\epsilon \in \left(\frac{10000}{d}, \frac{1}{8}\right)$ there exists a family of circuits $\mathbf{A} = \{\mathbf{A}_d\}_d$ and a family of function classes $\mathcal{F}$ for which the following conditions hold. For every learning $\mathbb{L} = \{\mathbb{L}_d\}_d$ that for every $d$ samples $\mathcal{D}_d$ always and $h_d \in \mathcal{H}_d$,*

$$\mathbf{A} \in_1 \text{WATERMARK}\begin{pmatrix} \mathbb{L}, \mathcal{F}, \epsilon, q = O\left(\frac{1}{\epsilon}\right), S_{\mathbf{A}} = O\left(\frac{d}{\epsilon}\right), \\ S_{\mathbf{B}} = \frac{d}{100}, l = 1 - \frac{1}{100}, c = 1 - \frac{2}{100}, s = \frac{56}{100} \end{pmatrix}.$$

Note that the setting of parameters is such that $\mathbf{A}$ can learn (with high probability) a classifier of error $\epsilon$, but $\mathbf{B}$ is *not* able to learn a low-error classifier within its allotted circuit size $S_{\mathbf{B}}$. This contrasts with Lemma 5, where $\mathbf{B}$ has a sufficiently large circuit size to learn. This is the regime of interest for Watermarks, where the scheme is expected to be secure against $\mathbf{B}$ with limited circuit complexity.

*Proof.* Let $\mathcal{D}$ be the uniform distribution over $[N]$ for $N = 100d^2$, where recall that $[N] = \{1, \ldots, N\}$. Let $\mathcal{H}$ be the concept class of functions that have exactly $d$ +1's in $[N]$. Note that $\mathcal{H}$ has VC-dimension $d$. Let $h \in \mathcal{H}$ be the ground truth.

---

[16]Note that we measure the error of $f$ not $f'$.

**Construction of A.** **A** works as follows. It draws $n = O\left(\frac{d}{\epsilon}\right)$ samples from $\mathcal{D}$ labeled with $h$. Let's call them $\mathbf{x}_{\text{TRAIN}}$. Let

$$A := \{x \in [N] : \mathbf{x}_{\text{TRAIN}}, h(x) = +1\}, B := \{x \in [N] : x \in \mathbf{x}_{\text{TRAIN}}, h(x) = -1\}.$$

**A** takes a uniformly random subset $A_w \subseteq A$ of size $q$. It defines sets

$$A' := A \setminus A_w, \ B' := B \cup A_w.$$

**A** computes $f$ consistent with the training set $\{(x, +1) : x \in A'\} \cup \{(x, -1) : x \in B'\}$. **A** samples $S \sim \mathcal{D}^q$. It defines the watermark to be $\mathbf{x} := A_w$ with probability $\frac{1}{2}$ and $\mathbf{x} := S$ with probability $\frac{1}{2}$.

**A** sends $(f, \mathbf{x})$ to **B**. **A** can be implemented with circuit complexity $O\left(\frac{d}{\epsilon}\right)$.

**A is a Watermark.** We claim that $(f, \mathbf{x})$ constitutes a watermark.

It is possible to construct a watermark of prescribed size, i.e., find a subset $A_w$ of a given size, only if $|A| \geq q$. The probability that a single sample from $\mathcal{D}$ is labeled $+1$ is $\frac{d}{N}$, so by the Chernoff bound (Fact 1) $|A|, |B| > \frac{dn}{2N} \geq q$ with probability $1 - \frac{1}{100}$, where we used that $n = O\left(\frac{d}{\epsilon}\right), N = 100d^2, q = O(\frac{1}{\epsilon})$.

**Correctness.** Let $h'(x) := h(x)$ if $x \in [N] \setminus A_w$ and $h'(x) := -h(x)$ otherwise. Note that $h'$ has exactly $d - q$ +1's in $[N]$. By construction, $f$ is a classifier consistent with $h'$. By the PAC theorem we know that with probability $1 - \frac{1}{100}$, $f$ has an error at most $\epsilon$ wrt to $h'$ (because the hypothesis class of functions with *at most* $d$ +1's has a VC dimension of $O(d)$). $h'$ differs from $h$ on $q$ points, so

$$\text{err}(f) \leq \epsilon + q/N = O\left(\epsilon + \frac{1}{\epsilon d^2}\right) = O(\epsilon). \tag{14}$$

with probability $1 - \frac{1}{100}$, which implies that *correctness* is satisfied with $l = 1 - \frac{1}{100}$.

**Distinguishing of x and $\mathcal{D}^q$.** Note that the distribution of $A_w$ is the same as the distribution of a uniformly random subset of $[N]$ of size $q$ (when taking into account the randomness of the choice of $h \sim U(\mathcal{H})$). Observe that the probability that drawing $q$ i.i.d. samples from $U([N])$ we encounter repetitions is at most

$$\frac{1}{N} + \frac{2}{N} + \cdots + \frac{q}{N} \leq \frac{3q^2}{N} \leq \frac{1}{100},$$

because $q < \frac{d}{100} < \frac{\sqrt{N}}{10}$. This means that $\frac{1}{100}$ is an information-theoretic upper bound on the distinguishing advantage between $\mathbf{x} = A_w$ and $\mathcal{D}^q$.

Moreover, **B** has access to at most $t$ samples and the probability that the set of samples **B** draws from $\mathcal{D}^t$ and $A_w$ have empty intersection is at least $1 - \frac{1}{100}$. It is because it is at least $(1 - \frac{t}{N})^t \geq (1 - \frac{1}{\sqrt{N}})^{\sqrt{N}/10} \geq 1 - \frac{1}{100}$, where we used that $t < \frac{\sqrt{N}}{10}$.[17]

Note that by construction $f$ maps all elements of $A_w$ to $-1$. The probability over the choice of $F \sim \mathcal{D}^q$ that $F \subseteq h^{-1}(\{-1\})$, i.e., all elements of $F$ have true label $-1$, is at least

$$\left(1 - \frac{d}{N}\right)^q \geq 1 - \frac{1}{100}.$$

The three above observations and the union bound imply that the distinguishing advantage for distinguishing $\mathbf{x}$ from $\mathcal{D}^q$ of **B** is at most $\frac{4}{100}$ and so the *undetectability* holds with $s = \frac{8}{100}$.

**Unremovability.** Assume, towards contradiction with *unremovability*, that **B** can find $\mathbf{y}$ that with probability $s' = \frac{1}{2} + \frac{6}{100}$ satisfies $\text{err}(\mathbf{x}, \mathbf{y}) \leq 2\epsilon$. Notice, that $\text{err}(A_w, f(A_w)) = 1$ by construction.

Consider an algorithm $\mathcal{A}$ for distinguishing $A_w$ from $\mathcal{D}^q$. Upon receiving $(f, \mathbf{x})$ it first runs $\mathbf{y} = \mathbf{B}(f, \mathbf{x})$ and returns 1 iff $d(\mathbf{y}, f(\mathbf{x})) \geq \frac{q}{2}$. We know that the distinguishing advantage is at most $\frac{1}{2} + \frac{4}{100}$, so

$$\frac{1}{2}\mathbb{P}_{\mathbf{x}:=A_w}[\mathcal{A}(f, \mathbf{x}) = 1] + \frac{1}{2}\mathbb{P}_{\mathbf{x}\sim\mathcal{D}^q}[\mathcal{A}(f, \mathbf{x}) = 0] \leq \frac{1}{2} + \frac{4}{100}.$$

---

[17]If the sets were not disjoint then **B** could see it as suspicious because $f$ makes mistakes on all of $A_w$.

But also note that

$$s' \leq \mathbb{P}_{\mathbf{x} \sim \mathbf{A}}[\mathrm{err}(\mathbf{x}, \mathbf{y}) \leq 2\epsilon]$$

$$\leq \frac{1}{2}\mathbb{P}_{\mathbf{x}:=A_w}[d(\mathbf{y}, f(\mathbf{x})) \geq (1 - 2\epsilon)q] + \frac{1}{2}\mathbb{P}_{\mathbf{x} \sim \mathcal{D}^q}[d(\mathbf{y}, f(\mathbf{x})) \leq (2\epsilon + \mathrm{err}(f))q]$$

$$\leq \frac{1}{2}\mathbb{P}_{\mathbf{x}:=A_w}[d(\mathbf{y}, f(\mathbf{x})) \geq q/2] + \frac{1}{2}\mathbb{P}_{\mathbf{x} \sim \mathcal{D}^q}[d(\mathbf{y}, f(\mathbf{x})) \leq q/2] + \frac{1}{100}$$

$$\leq \frac{1}{2}\mathbb{P}_{\mathbf{x}:=A_w}[\mathcal{A}(f, \mathbf{x}) = 1] + \frac{1}{2}\mathbb{P}_{\mathbf{x} \sim \mathcal{D}^q}[\mathcal{A}(f, \mathbf{x}) = 0] + \frac{1}{100}.$$

Combining the two above equations we get a contradiction and thus the *unremovability* holds with $s' = \frac{1}{2} + \frac{6}{100}$.

**Uniqueness.** The following **B** certifies *uniqueness*. It draws $O\left(\frac{d}{\epsilon}\right)$ samples from $\mathcal{D}$, let's call them $\mathbf{x}'_{\mathrm{TRAIN}}$ and trains $f'$ consistent with it. By the PAC theorem $\mathrm{err}(f') \leq \epsilon$ with probability at least $1 - \frac{1}{100}$. Next upon receiving $\mathbf{x} \in \{0,1\}^{nq} = [N]^q$ it returns $y = f'(\mathbf{x})$. By the fact that $\mathbf{x}$ is a random subset of $[N]$ of size $q$ by the Chernoff bound, the union bound we know that $\mathrm{err}(\mathbf{x}, \mathbf{y}) = \mathrm{err}(\mathbf{x}, f'(\mathbf{x})) \leq 2\epsilon$ with probability at least $1 - \frac{2}{100}$ over the choice of $h$. This proves *uniqueness*. $\square$

## J    Future Directions

Below we provide some interesting technical and conceptual future directions.

### J.1    Alternate Viewpoint for Task Complexity

Here we briefly note a connection to the **Platonic Representation Hypothesis** (PRH) Huh et al. [2024], which posits that as models grow in capacity, their learned representations become increasingly similar—hence properties (and failures) transfer more readily across models. Theorem 3 in our work shows that transferable attacks arise only when the underlying learning task is computationally hard (in the EFID sense). If PRH's convergence of representations holds, one should indeed expect greater transferability of adversarial examples; our result then suggests that such transferability is an indicator of *computational complexity* of the task. In this view, a (plausibly) necessary condition—and a partial explanation—of PRH is that frontier models are solving **increasingly difficult** problems, which in turn induces representation similarities and transfer. Making these connections precise could be a promising direction for future work.

### J.2    Practical Aspects of our Main Theorem

At a fundamental level, families of Boolean circuits are Turing complete, meaning they can simulate any algorithm that a Turing machine can. This makes them expressive enough to capture any computable algorithm and a natural abstraction for studying the inherent properties of learning tasks, independent of specific algorithms. The existence result can guide practical efforts by informing where to search. A key part of our proof is the formulation of a zero-sum game between a "watermarking agent" and a "defense agent," where the game's value indicates which of the three properties exists. Moreover, an optimal strategy in this game corresponds to an actual implementation of a watermark, defense, or transferable attack.

Though finding a Nash equilibrium in such a game seems computationally challenging—since the action spaces involve all circuits of a given size—recent iterative algorithms for large-scale games Lanctot et al. [2017], McAleer et al. [2021], Adam et al. [2021] offer promising approaches. These methods work by evaluating only parts of the game at each step, discovering good strategies over time. Recall that our model captures examples like Pal and Vidal [2020].

In practical settings, circuits can be replaced with standard ML models (e.g., deep neural networks). This opens the door to algorithms that (1) determine whether a given task admits a defense, watermark, or transferable attack, and (2) produce an implementation accordingly. In summary, our theory shows that for a given task, one only needs to set up the appropriate loss and apply these iterative algorithms to extract the desired property. We believe this represents an exciting future direction at the intersection of large-scale algorithmic game theory and AI security.

## J.3 Beyond Classification

Inspired by Theorem 2, we conjecture a possibility of generalizing our results to generative learning tasks. Instead of a ground truth function, one could consider a ground truth quality oracle $Q$, which measures the quality of every input and output pair. This model introduces new phenomena *not* present in the case of classification. For example, the task of *generation*, i.e., producing a high-quality output $y$ on input $x$, is decoupled from the task of *verification*, i.e., evaluating the quality of $y$ as output for $x$. By decoupled, we mean that there is no clear formal reduction from one task to the other. Conversely, for classification, where the space of possible outputs is small, the two tasks are equivalent. Without going into details, this decoupling is the reason why the proof of Theorem 1 does not automatically transfer to the generative case.

This decoupling introduces new complexities, but it also suggests that considering new definitions may be beneficial. For example, because generation and verification are equivalent for classification tasks, we allowed neither **A** nor **B** access to $h$, as it would trivialize the definitions. However, a modification of the Definition 8 (Watermark), where access to $Q$ is given to **B** could be investigated in the generative case. Interestingly, such a setting was considered in [Zhang et al., 2023], where access to $Q$ was crucial for mounting a provable attack on "all" strong watermarks. As we alluded to earlier, Theorem 2 can be seen as an example of a task, where generation is easy but verification is hard – the opposite to what Zhang et al. [2023] posits. Furthermore, recent work Gluch and Goldwasser [2025] proved that, in the context of adversarial robustness and safety, there exist generative learning tasks for which *verification* (or *detection*, i.e., identifying adversarial examples) is strictly harder than *generation* (or *mitigation*, i.e., allocating additional inference time to correct outputs of the base model).

We hope that careful formalizations of the interaction and capabilities of all parties might give insights into not only the schemes considered in this work, but also problems like weak-to-strong generalization [Burns et al., 2024] or scalable oversight [Brown-Cohen et al., 2023].

