# OpenReview forum: "The Good, the Bad and the Ugly: Meta-Analysis of Watermarks, Transferable Attacks and Adversarial Defenses"
_NeurIPS.cc/2025/Conference — NeurIPS 2025 poster_

### Official Review · Reviewer_9QUZ · 2025-06-13

**Clarity:** 1
**Significance:** 2
**Originality:** 4
**Rating:** 4
**Confidence:** 3

**Summary:**

This paper shows that for all learning tasks, at least one of the following three algorithms exists, Watermark, Transferable Attacks, and Adversarial Defenses. They give rigorous definitions of these algorithms by formulating them as two-player interaction protocols, and prove their main result based on circuit complexity.

**Questions:**

1.	What is the relationship between the Undetectability of Watermark in your definition and the pseudorandomness of error-correcting codes[a]?

2.	Can these results be expanded to continuous data space?

3.	In Section 3.3, the probability is $1/2+\gamma$, why $\gamma$ be $[0,1]$?



Ref:

[a] Christ and Gunn. Pseudorandom Error-Correcting Codes. 2024.

**Ethical Concerns:**

["NO or VERY MINOR ethics concerns only"]

**Final Justification:**

The rebuttal has addressed my concerns. I tend to be more positive on this paper.

**Limitations:**

They talked about some limitations on extending these ideas to generative tasks. But more perspective should be considered. (See weaknesses and questions, like $F_2$ field, potential misalignment of definitions.)

**Paper Formatting Concerns:**

No.

**Quality:**

3

**Strengths And Weaknesses:**

Strengths:

1.	This paper provides comprehensive analysis of Watermarks, Adversarial Defenses and Transferable Attacks under a cryptographic perspective.

2.	Relationship between Transferable Attacks and complex learning tasks based on cryptography has been well-studied.


Weaknesses:

1.	The definitions are very obscure, it’s hard for readers to understand. There are no explanations of why these definitions are reasonable, and no discussions on the relationship between these definitions and the well-known concepts of Watermarking, Transferable Attacks and Adversarial Defenses. For example, in the definition of Watermark, what is $B_n$? Is it a decoder that try to obtain the injected watermarked messages? In the definition of Adversarial Defense, why soundness means adversarial inputs are detected? Does this definition align more with the adversarial detection rather than defense? Also in the definition of Transferable Attacks, why it is defined in this way? I don’t think this align well with the well-known transferable attacks, that an attack can be transferable across models or datasets. Overall, these definitions are abstract, it is very hard for readers to connect them with real-world scenarios. I suggest the authors to provide more discussions, explanations of these definitions, and their potential alignment with well-known definitions of these aspects.

2.	All these definitions and results are based on the $F_2$ field. This may be not practical for modern deep networks and embeddings.

3.	Authors say this paper focus on classification tasks. However, not any experimental results can validate the soundness of their definitions and theorems. I suggest the authors to conduct some simple experiments on MNIST, or at least some hand-wavy evaluation on certain human-crafted datasets (with some multivariate Gaussian distributions I guess, as often used in other theoretical works). It is worthy to note that, NeurIPS is a conference consider more on practicality, even for a theoretical paper, some simple experiments are welcome and necessary.

---

> ### Author Rebuttal · Authors · 2025-07-31
>
> # Response to Reviewer 9QUZ
>
> We thank the reviewer for their careful evaluation and recognition of our contributions. Below, we address each concern in turn.
>
> ## Response to W1 - Explaining Definitions 3–5 in familiar terms
>
> ---
>
> ### 1.  What is $B_n$ in Definition 3?
> $B_n$ (or Bob) is a party that tries to prove that it is *not* using the (watermarked) model provided by $A_n$ (Alice).
> It either is dishonest and is using the watermarked model or is honest and trained its own model from scratch.
>
> Alice, on the other hand, has planted a backdoor into her own (open-sourced) model, causing deliberate errors whenever a specific trigger is included in the input. The existence of such an honest $B_n$ ensures that a watermark verification cannot (Uniqueness property in Definition 3) falsely accuse any legitimate (non-watermarked) model.
>
> Answering your question directly, if $B_n$ is dishonest then you are right - it tries to remove the watermark.
> However, we clarify that in our watermarking notion, the outputs of the model themselves do not carry hidden messages or explicit watermarks. Rather, Alice verifies ownership because her watermarked model deliberately answers incorrectly on very specific trigger inputs (the backdoors), while an honest $B_n$—a model
> trained independently from scratch—would answer correctly. We refer the reviewer to our answer to Question 1 below, where we discuss this difference in more detail.
>
> ---
>
> ### 2. Why does “Soundness” in Definition 4 mean the attack is detected?
>
> This is a good observation. Yes, one can (partly) interpret Definition 4 as a "defense against adversarial examples *by detection*".
> However, note that if $x$ is not detected as adversarial, then the soundness property in Definition 4 guarantees that $f$ has low error on $x$, i.e., it successfully defended against the adversary.
> Additionally, our notion of a defense is very strong, as compared to $l_p$ bounded perturbation model, as it needs to hold against all (efficient) adversaries.
> This implies that a defense needs to hold against an adversary that provides samples that are clearly far from the support of the distribution.
> It is unreasonable to require $f$ to answer correctly on such samples and thus detection (rejection) is necessary.
> Just to reemphasise, a similar rejection-based definition of a defense was considered in Goldwasser et al., 2020.
>
>
> **To recapitulate Adversarial Defenses (Definition 4, informally):**
>
> | **Property in Def. 4** | **Classical analogue** | **Why it is needed?** |
> |------------------------|------------------------|------------------------|
> | Correctness | Baseline test‑error requirement in certified / detection‑based defences | Ensures the defended model remains useful. |
> | Completeness | “No false positive” guarantee in detection frameworks (*Goldwasser et al., 2020*) | Prevents trivial defences that reject everything. |
> | Soundness | Detection+robustness guarantee (rejection‑based defenses) | Formalises that attacks must both fool and stay indistinguishable. |
>
>
> ---
>
> **3. Why define Transferable Attack this way?**
> We agree that *transferability* is usually understood as an adversarial input crafted on a *surrogate model* that also fools a different,
> unseen *target model*. Our notion builds directly on this intuition but lifts it from individual models to learning tasks.
>
> In our framework, Alice trains one surrogate model and wonders whether inputs that fool her model also fool *all* other models, i.e., models that a different party (e.g., Bob) could train on the same task within the given resource bound. This preserves the classical idea of transferability but generalizes it to a task-level, resource-aware notion. Crucially, the result shows that transferability itself depends on the comparative computational power of attacker and defender, a nuance often overlooked in empirical work.
>
> We will include a description of this perspective directly after Definition 5. Thank you for raising this point.
>
> ---
>
> We believe these edits will make each definition’s role and its link to real-world concepts transparent.
>
> ## Response to W2 - Use of the $\mathbb{F}_2$ (bit) model.
>
> We work over $\mathbb{F}_2$ only for analytic convenience: Boolean circuits are the standard complexity-theoretic abstractions because *every* finite-precision operations used in machine learning, e.g., addition, matrix multiplication, activation functions, can be implemented by a circuit of AND/OR/NOT gates with a small overhead.
> We further emphasize that modern deep networks and embeddings are implemented using finite precision arithmetic at the hardware level (e.g.\ 8- or 16-bit integer/FLOAT16 arithmetic). The particular choice of gates and basic operations affects the size of circuits only by constants.
>
> Thus, the  $\mathbb{F}_2$ can be assumed with a loss of generality for our results applied to modern deep-learning systems.
>
> We also refer the reviewer to our answer to Question 2 for a related reply.
>
> ## Response to W3 – From Theory to Practice (experiments)
>
> As the reviewer noted, our focus is theoretical. In our main result we consider a very general class of algorithms, i.e., all circuits of given size, and because of that, experiments are very challenging to run (due to the number of them). However, there is related work that provides experiments for a similar setting, albeit for more restricted classes.
> For example, Pal and Vidal, 2020 show (first theoretically) that Fast Gradient Methods (as adversarial attacks) and Randomized Smoothing (as defenses) can form a Nash equilibrium under a restricted additive noise model.
> Additionally, they provide *experiments* showing that their theoretical result does hold for some datasets (e.g., MNIST).
> The experiments in Pal and Vidal, 2020 consider a game with a slightly different utility from the one used by us.
> If the experiments were run for our game, then depending on the utility obtained at the equilibrium, our approach would extract one of the three schemes (defense, watermark, transferable attack).
>
> To go more into detail, a key part of our proof is the formulation of a zero-sum game between a "watermarking agent" and a "defense agent," where the game's value indicates which of the three properties exists. Moreover, an optimal strategy in this game corresponds to an actual implementation of a watermark, defense, or transferable attack.
>
> Though finding a Nash equilibrium in such a game seems computationally challenging—since the action spaces involve all circuits of a given size—recent iterative algorithms for large-scale games [Lanctot et al., 2017; McAleer et al., 2021; Adam et al., 2021] offer promising approaches. These methods work by evaluating only parts of the game at each step, discovering good strategies over time.
>
> In practical settings, circuits can be replaced with standard ML models (e.g., deep neural networks). This opens the door to algorithms that (1) determine whether a given task admits a defense, watermark, or transferable attack, and (2) produce an implementation accordingly. In summary, our theory shows that for a given task, one only needs to set up the appropriate loss and apply these iterative algorithms to extract the desired property. We believe this represents an exciting future direction at the intersection of large-scale algorithmic game theory and AI security.
>
>
> ## Response to Q1 - What is the relationship between the Undetectability of Watermark in your definition and the pseudorandomness of error-correcting codes [1]?
> [1] Christ and Gunn. Pseudorandom Error-Correcting Codes. 2024.
>
> The main difference is that we watermark a model, while [1] watermarks the outputs of a model - so the goals are different but related.
> Intuitively, [1] wants to produce *outputs* with a hidden watermark, while our work produces a *model* such that if queried in a certain (backdoor) way the answer would "identify" the model "exactly".
> Our requirement is stronger in the following ways:
> - We give the adversary white-box access to the model while [1] allows only black-box access,
> - our watermark is unremovable in a stronger than [1] sense, i.e., no efficient algorithm is able to "remove it", while [1] allows only a certain class of perturbations to be applied on the outputs, namely constant fraction edit distance. Note, that even a computationally weak adversary can easily modify an output of an LLM that produces a modified version with a high edit distance, e.g., paraphrasing or "translate to French and back to English" attacks are very effective at removing watermarks.
> - both schemes require that watermarked models/outputs are not falsely detected - our uniqueness property, and soundness in [1].
>
> ## Response to Q2 - Can these results be expanded to continuous data space?
> Thank you for raising this point, as it allows us to explain how our results apply more broadly.
>
> Any practical ML pipeline—whether it processes 256×256 RGB images, token sequences, or graphs—executes a finite sequence of arithmetic and logical operations. Bit-level compilation therefore yields a polynomial-size Boolean circuit (e.g., a ReLU network with `p` weights compiles to `O(p log q)` gates when each weight is quantised to `q` bits). Because our theorems depend only on the *ratio* of prover to verifier circuit sizes, the dimensionality or structure of the data does not change the trade-offs.
>
> ## Response to Q3 - $\gamma \in [0,1]$ ?
>
> We thank the reviewer for finding this typo. $\gamma$ should belong to $[0,1/2]$.
>
> ---
>
> ### References:
>
> Lanctot, Marc, et al. "A unified game-theoretic approach to multiagent reinforcement learning." NeurIPS 2017.
>
> McAleer, Stephen, et al. "XDO: A double oracle algorithm for extensive-form games." NeurIPS 2021.
>
> Adam, Lukáš, et al. "Double oracle algorithm for computing equilibria in continuous games." AAAI 2021.
>
> Pal, Ambar, and René Vidal. "A game theoretic analysis of additive adversarial attacks and defenses." NeurIPS 2020.

---

> > ### Comment · Reviewer_9QUZ · 2025-08-04
> >
> > Thanks for your rebuttal. The answers have addressed most of my concerns. Therefore, I will increase my rating to 4. I encourage the authors to explain their definitions more clearly in the revised version.

---

> > > ### Author Response · Authors · 2025-08-08
> > > **Thank You for your Update**
> > >
> > > Thank you for raising your score and providing useful suggestions to improve the clarity of the paper. For the final version, we will add additional explanations of the definitions in the main paper and the appendix.

---

### Official Review · Reviewer_j4Va · 2025-06-23

**Clarity:** 3
**Significance:** 3
**Originality:** 3
**Rating:** 5
**Confidence:** 3

**Summary:**

This paper extends prior work exploring provable properties of learning tasks and tradeoffs between backdoor attacks and adversarial robustness. It proves that, for any learning task, a watermark, adversarial defense, or transferable task must exist. This result is rigorously derived through a series of proofs.

**Questions:**

- It seems the only type of watermark considered in this paper is a backdoor-based watermark. If this is the case, why don't the authors generalize their result to showing that "for every learning task, at least one of the three must exist: a backdoor, an adversarial defense, or a transferable attack"? There are many proposed types of watermarks now, particularly for LLMs (lines 993-1026 of supplementary), so making a broad claim about "watermarks" when in fact what is meant is a backdoor was confusing.

- In Figure 1, who ``owns'' the model? It seems like model ownership transfers back and forth between Alice and Bob -- in "watermark" setting, Alice owns the model and proves ownership by embedding a strong watermark; in the "defense" setting, Bob owns the model and is defending it; and in the "transferable" setting, Bob again owns the model and is defending it. A simple but helpful change to the model would be updating the protocol so one player is always the attacker and one is always the defender.

- Can these results generalize beyond classifiers?

**Ethical Concerns:**

["NO or VERY MINOR ethics concerns only"]

**Final Justification:**

The rebuttal adequately addressed my concerns. Assuming the promised changes are made to the paper, I remain in favor of acceptance and am updating my score to a 5.

**Limitations:**

The paper provides an interesting theoretical result but fails to connect this result to important real-world use cases. The paper would be greatly improved by the inclusion of a discussion setting that contextualizes the theoretical finding and helps the reader understand its significance.

**Quality:**

3

**Strengths And Weaknesses:**

Strengths:
- Strong mathematical formalism of complex ideas. Proofs are communicated clearly and concisely.
- Establishes novel results on transferable attacks and their connection to the computational complexity of the learning task. Side note on this - the statement on lines 257-258 ("One can interpret this as showing that Transferable Attacks exist
only for complex learning tasks, in the sense of computational complexity theory") made me think of the Platonic Representation Hypothesis (PRH) paper (https://arxiv.org/pdf/2405.07987), which claims that increasingly complex models will become more similar (e.g. more transferable properties between the models). A key limitation of the PRH paper is that it does not back up this claim, but rather derives it from intuition. I wonder if your result here about transferability as a result of task complexity could provide some theoretical basis for this claim. Interesting future work, perhaps.

Weaknesses:
- Paper claims that result holds for any learning task, but proofs and definitions appear specific to classifiers. Claims should be adjusted appropriately.
- Paper lacks discussion to connect findings to any real-world impacts. The theoretical result is interesting, but I finished the paper wondering why the result matters.

---

> ### Author Rebuttal · Authors · 2025-07-31
>
> # Response to Reviewer j4Va
>
> Thank you for your review and for recognizing the strengths and contributions of our work. We address your comments individually below.
>
> ---
>
> ## Response to Connection to Platonic Representation Hypothesis (PRH)
>
> Thank you for drawing the connection to the *Platonic Representation Hypothesis* (PRH). This is indeed an interesting connection to our work! We also see a connection to Ilyas et al. 2019, that, at least morally, claims that adversarial examples might transfer between very accurate models.
>
> Our Theorem 3 shows that a *transferable attack* exists only when the underlying learning task is computationally "hard" (in the EFID sense)—that is.
> If the representations of models converge (as observed in PRH), then we might expect the adversarial examples to transfer and our results imply that the underlying learning task is "computationally complex".
> This shows that a perhaps necessary condition, and thus a partial explanation of PRH, for models' representations converging is that frontier models solve increasingly difficult problems.
>
> Overall, we agree that this bridge between task complexity and representation similarity is a promising direction. In the final version, we will add a short remark to **Section 7 (Future Work)** referencing PRH as an empirical hypothesis that our complexity-based formulation may help formalize and test.
>
> We appreciate the reviewer’s insight and will highlight this connection to stimulate further research.
>
> ## Response to W1 – Scope of the result
>
> Thank you for pointing out the wording issue. Throughout the paper, we use the term “learning task” exactly as defined in **Definition 1** (lines 128–131), i.e., a binary or multi-class classification problem. All proofs are carried out in that setting.
>
> To avoid overstatement, we will:
>
> - Change “any learning task” to **“any (binary) classification task”** in the Abstract and in the bullet list of Section 1.1.
>
> As noted in our response to Reviewer f8Ls (W3), we already discuss in *Related Work* (lines 98–102) how backdoor-based watermarks have been applied to generative models, and we clarify in Appendix E how the framework could be adapted beyond classification.
>
>
> ## Response to W2/L1 – Real world-impact and Key Take aways:
>
> ### Real world Impact
>
> We would like to clarify that trade-offs between backdoor-based attacks and adversarial defenses have been studied in several prior works, primarily through empirical analysis. These works generally observe that models made robust to certain adversarial attacks often become more vulnerable to backdoor-based attacks, highlighting the nuanced and complex nature of such trade-offs.
>
> For example, (Pal and Vidal, 2020) show (first theoretically) that Fast Gradient Methods (as adversarial attacks) and Randomized Smoothing (as defenses) can form a Nash equilibrium under a restricted additive noise model.
> Additionally, they provide *experiments* showing that their theoretical result does hold for some datasets (e.g., MNIST).
> The experiments in Pal and Vidal, 2020 consider a game with a slightly different utility from the one used by us.
> If the experiments were run for our game, then depending on the utility obtained at the equilibrium, our approach would extract one of the three schemes (defense, watermark, transferable attack).
> Our theoretical contributions can be seen as a generalization of this result to a broader class of attacks and defenses.
>
>
> ### Computational Resources
>
> Our theorem also offers insight into the computational resources required to successfully construct a watermark or a defense. Specifically, if one aims to defend against adversaries with a computational budget of
> $T$ units (interpretable as compute time), our result suggests that by increasing one’s own resources to $T^2$, it is likely to succeed—provided the learning task admits a defense. Conversely, if such an effort fails, it may indicate that the task permits transferable attacks, which in turn could preclude the existence of a watermark as well. This perspective could serve as a practical guideline for how to allocate resources before concluding that a task may not admit a defense. To our best knowledge, we provide one of the first quantitive bounds on the resources of attackers and defenders in very general settings (e.g., beyond bounded VC-dim etc.)."
>
> ### Key Take away
>
> Most prior works have focused primarily on the trade-off between backdoor attacks and adversarial defenses, often via empirical studies. Our work shows that to fully characterize this landscape, it is essential to include transferable attacks—a point we believe constitutes one of our key technical contributions.
>
> More concretely, we show that there exist learning tasks where one can fail to be both robust (via adversarial defenses) and verifiable (via backdoor-based watermarks)—this is precisely the case captured in Theorem 2 through the construction of transferable attacks.
>
> However, this is not the only scenario we consider. In fact, we prove a more general result: every learning task must admit at least one of the three schemes. For example, we show that there exist restricted classes of learning tasks with bounded VC-dimension where robustness and verifiability can both be achieved simultaneously (Appendix I and Appendix J, Lemmas 3 and 4). For such tasks, transferable attacks do not arise.
>
>
>
> ## Response to Q1 – Why we focus on Backdoor-based Watermarks
>
> As detailed in Appendix A.1.1, we focus on *backdoor-based watermarking* because it is currently the only family of methods that satisfies all three properties required for secure black-box ownership verification, as formalized in Definition 3:
>
> 1. **No accuracy loss** on the main task,
> 2. **Robustness to removal**, and
> 3. **Undetectable verification** (i.e., trigger inputs are indistinguishable from normal data).
>
> Other approaches—such as *white-box watermarking*, which embeds hidden signals in model weights—are fundamentally different in nature. While they may be suitable for analysis, they are not applicable in settings where the model is only accessible via an API.
>
> Because *black-box verification* is the more realistic scenario in practice—especially when models are misused to offer services via an interface—we focus on techniques suitable for that setting. In this realm, the research community has largely converged on *backdoor-based watermarking* as the most viable approach.
>
> For completeness, we discuss other techniques such as white-box watermarking in Appendix A.1.1.
>
> We hope this clarifies why our framework centers on backdoor-based watermarking and how it aligns with the dominant practical and research focus in secure model ownership verification.
>
> ## Response to Q2 – Who owns the model in Fig. 1?
> Unfortunately, "owner" is not the most precise word to be used in this setting, however, we try to answer your question using this terminology.
>
>
> Bob is always the **model owner / defender**, and Alice is always the **adversary / verifier**. Their roles remain fixed throughout the paper; only Alice’s objective varies across the three scenarios:
>
> | Setting         | Alice’s Goal                                      | Bob’s Goal (Ownership Fixed)         |
> |-----------------|----------------------------------------------------|--------------------------------------|
> | (a) Watermark   | Verify that Bob is using *her* watermarked model | Prove that the model is not watermarked but trained from scratch      |
> | (b) Defence     | Induce errors undetected                          | Reject or correct adversarial inputs |
> | (c) Transferable| Find a single input that fools *every* Bob-trainable model | Same as (b)                  |
>
> Bob is always the party required to answer Alice’s queries, which is why he is consistently considered the “owner” of the model across all settings. It is possible though, that in case of ownership theft, that Bob is using the watermarked model by Alice in Definition 3.
>
> We will edit the caption to read (new text in *italics*):
>
> > **Figure 1**: Schematic overview of the interaction structure, along with short, informal versions of
> > our definitions of (a) Watermark (Definition 3), (b) Adversarial Defense (Definition 4), and (c)
> > Transferable Attack (Definition 5), with (c) tied to cryptography (see Section 6).
> > *Throughout this figure, Alice (the verifier) plays the role of the adversary, while Bob (the prover) plays the role of the defender.*
>
> We hope this clarifies the roles. Thank you for bringing this to our attention.
>
> ## Response to Q3 – Generalization Beyond Classifiers
>
> Yes—two routes are discussed in the manuscript:
>
> 1. **Fine-tune generative models into a small classification task.**
>    Pre-trained language models (PLMs), for example, can be lightly fine-tuned so a back-door watermark verifies ownership; we cite such PLM watermarks in Related Work (lines 98–102).
>
> 2. **Direct non-classification extension.**
>    Appendix E outlines how our verifier–prover game carries over when fine-tuning is not possible, by redefining correctness and detection on the model’s native output space.
>
> ---
>
> We thank the reviewer for their insightful feedback, particularly regarding the Platonic Representation Hypothesis (PRH), which aligns well with our framework. We hope our clarifications and revisions address the concerns raised.
>
> ### References:
>
> Lanctot, Marc, et al. "A unified game-theoretic approach to multiagent reinforcement learning." NeurIPS 2017.
>
> McAleer, Stephen, et al. "XDO: A double oracle algorithm for extensive-form games." NeurIPS 2021.
>
> Adam, Lukáš, et al. "Double oracle algorithm for computing equilibria in continuous games." AAAI 2021.
>
> Pal, Ambar, and René Vidal. "A game theoretic analysis of additive adversarial attacks and defenses." NeurIPS 2020.
>
> Ilyas et al. 2019, "Adversarial Examples Are Not Bugs, They Are Features"

---

> > ### Comment · Reviewer_j4Va · 2025-08-01
> > **Response to rebuttal**
> >
> > Thanks to the authors for their thorough and thoughtful response to my review. Assuming the promised changes are made to the paper, I remain in favor of acceptance and am updating my score to a 5.

---

> > > ### Author Response · Authors · 2025-08-08
> > > **Thank you for your Update**
> > >
> > > Thank you for raising your score and providing useful suggestions to improve the clarity of the paper. We will integrate the relevant points from our rebuttal into the final version accordingly.

---

### Official Review · Reviewer_je1d · 2025-06-25

**Clarity:** 1
**Significance:** 2
**Originality:** 2
**Rating:** 3
**Confidence:** 3

**Summary:**

In black-box settings, verifying a model is done through watermarking, which requires embedding a backdoor during training. This inherently conflicts with adversarial robustness because it sacrifices the model vulnerability for model verifiability. Thus, there’s a fundamental trade-off between verifiability and robustness. While previous works explored this trade-off (either empirically or theoretically) only between the two components (watermark vs. adversarial defense), this paper extends this by adding a third component, ‘transferable attacks’. It formally proves that for any discriminative learning task, at least one of the three phenomena: watermark, adversarial defense, or transferable attack must exist, revealing a deeper impossibility. This is proven using three mathematical notions: game theory, circuit complexity, and probabilistic proof. For demonstration, the authors construct a learning task with certain assumptions (encrypted data from the attacker, limited resource bound for the defender), showing that transferable attacks actually occur in practice.

**Questions:**

Please address the technical contribution and experimental result issues.

**Ethical Concerns:**

["NO or VERY MINOR ethics concerns only"]

**Final Justification:**

The rebuttal addresses my concerns to some degree, but I still think practicality is an important issue that must be addressed by the paper. If the other reviewers find the claims useful, I wouldn't argue against accepting this paper.

**Limitations:**

yes

**Quality:**

2

**Strengths And Weaknesses:**

### **Strengths**

-	The authors propose a generalizable theoretical framework that applies to any discriminative learning task, given a resource bound.

-	The authors provide a helpful illustration to better understand each scenario (watermark, adversarial defense, and transferable attack).

-	The authors utilize a diverse set of ideas taken from game theory, circuit complexity, and cryptography to support their claims.

### **Weaknesses**

-	The term ‘transferable attack’ used in this paper is defined differently than in the usual sense, which is very confusing. As far as I know, the term “transferability” in model security refers to an adversarial input crafted using one model (surrogate) being effective to another model (target). On the other hand, “transferable attack” in this paper refers to a set of queries that can fool all possible defenders within a given resource bound. Even though the authors note the difference in the footnote of section 1.1, I carefully suggest using a different naming to avoid confusion, considering that the definition of “transferable attack” is crucial for understanding the paper.

-	The clarity of the writing and the overall structure is difficult to follow. For example, it’s hard to grasp the main proposal or argument of the paper even after reading the abstract and the introduction. What is the main argument made in the paper? Why is it important to analyze these components (watermarks, adversarial defense, transferable attack) using a theoretical framework? In my understanding, the paper is trying to show the inherent limitation of a general learning task where it’s impossible for a model to be robust and verifiable at the same time, and if one ability is not dominant then there must exist a universally transferable attack.  However, this is not clear even after several readings of the manuscript. Also, since the paper assumes deep familiarity with computational learning theory, complexity theory, and cryptographic primitives, providing a high level introduction to these notions in advance would be better for a broader ML audience.

-	The technical contribution of the paper is limited. Although a thorough analysis is always necessary and meaningful, the analysis made in this paper does not seem to have a new finding or enhance the explainability of the problem in any aspect. What is the main takeaway of this paper? What can the readers learn from this?
-	The paper is entirely theoretical. While that is acceptable if the contribution is foundational (as it arguably is), the lack of even toy experimental demonstrations or case studies makes the concepts less tangible. No attempt is made to show how this framework might be instantiated in modern neural network architectures or datasets.

---

> ### Author Rebuttal · Authors · 2025-07-31
>
> # Response to Reviewer je1d
>
> Thank you for your review and for recognizing the strengths and contributions of our work. We address your comments individually below.
>
>
> ## Response to W1 – Use of the term “transferable attack”
>
> We agree that “transferability” is commonly understood as the ability of an adversarial input—crafted on a surrogate model—to fool a different, unseen target model. Our notion builds directly on this intuition, but generalizes it by shifting the focus from individual models to entire learning tasks.
>
> Specifically, in our framework, the surrogate model is trained by Alice herself and represents *one possible instantiation* of the surrogate model. The key question is whether adversarial inputs crafted on Alice’s model transfer to *all other models* that could be trained by any other party (e.g., Bob) on the same task. This formulation captures the original idea of transferability while lifting it to a task-level abstraction that allows for formal analysis.
>
> We believe this generalization is not only conceptually aligned with the standard usage but also necessary to rigorously study transferability. Importantly, our framework shows that transferability depends critically on the computational power of both the attacker and the defender—a subtlety that is often overlooked in empirical work.
>
> We acknowledge the risk of confusion and will add a clarifying remark early in Section 1.1, emphasizing that our use of “transferable attack” formalizes and extends the classical notion in a task-level, resource-bounded setting.
>
> We thank the reviewer for their thoughtful perspective on terminology, and we hope this response clarifies our intended usage.
>
>  ## Response to W2 – Clarity and Key takeaways
>
> We thank the reviewer for this important comment and believe that their intuition is largely correct. As noted, we do show that there exist learning tasks where one can fail to be both robust (via adversarial defenses) and verifiable (via backdoor-based watermarks)—this is precisely the case captured in Theorem 2 through the construction of transferable attacks.
>
> However, this is not the only scenario we consider. In fact, we prove a more general result: every learning task must admit at least one of the three schemes. For example, we show that there exist restricted classes of learning tasks with bounded VC-dimension where robustness and verifiability can both be achieved simultaneously (Appendix I and Appendix J, Lemmas 3 and 4). For such tasks, transferable attacks do not arise.
>
> Most prior works have focused primarily on the trade-off between backdoor attacks and adversarial defenses, often via empirical studies. Our work shows that to fully characterize this landscape, it is essential to include transferable attacks—a point we believe constitutes one of our key technical contributions. Furthermore, please look at the response to W3 below, for more practical take-aways and implications.
>
> Finally, due to space constraints, we were unable to include all relevant preliminaries in the main text. However, we will ensure that additional background material is included in the appendix to make the paper more accessible to a broader ML audience.
>
> ## Response to W3 – Implications of Our Results
>
> 1. ### Existing Results
>
>
> We would like to clarify that trade-offs between backdoor-based attacks and adversarial defenses have been studied in several prior works, primarily through empirical analysis. These works generally observe that models made robust to certain adversarial attacks often become more vulnerable to backdoor-based attacks, highlighting the nuanced and complex nature of such trade-offs.
>
> For example, (Pal and Vidal, 2020) show (first theoretically) that Fast Gradient Methods (as adversarial attacks) and Randomized Smoothing (as defenses) can form a Nash equilibrium under a restricted additive noise model.
> Additionally, they provide *experiments* showing that their theoretical result does hold for some datasets (e.g., MNIST).
> The experiments in Pal and Vidal, 2020 consider a game with a slightly different utility from the one used by us.
> If the experiments were run for our game, then depending on the utility obtained at the equilibrium, our approach would extract one of the three schemes (defense, watermark, transferable attack).
> Our theoretical contributions can be seen as a generalization of this result to a broader class of attacks and defenses.
>
>
> 2. ### Computational Resources
>
> Our theorem also offers insight into the computational resources required to successfully construct a watermark or a defense. Specifically, if one aims to defend against adversaries with a computational budget of
> $T$ units (interpretable as compute time), our result suggests that by increasing one’s own resources to $T^2$, it is likely to succeed—provided the learning task admits a defense. Conversely, if such an effort fails, it may indicate that the task permits transferable attacks, which in turn could preclude the existence of a watermark as well. This perspective could serve as a practical guideline for how to allocate resources before concluding that a task may not admit a defense. To our best knowledge, we provide one of the first quantitive bounds on the resources of attackers and defenders in very general settings (e.g., beyond bounded VC-dim etc.).
>
>
> We respectfully disagree that the paper offers limited new insight, and hope that the clarifications and proposed revisions make its contributions and takeaways more apparent.
>
>
> ## Response to W4 – From Theory to Practice
>
> A key part of our proof is the formulation of a zero-sum game between a "watermarking agent" and a "defense agent," where the game's value indicates which of the three properties exists. Moreover, an optimal strategy in this game corresponds to an actual implementation of a watermark, defense, or transferable attack.
>
> Though finding a Nash equilibrium in such a game seems computationally challenging—since the action spaces involve all circuits of a given size—recent iterative algorithms for large-scale games [Lanctot et al., 2017; McAleer et al., 2021; Adam et al., 2021] offer promising approaches. These methods work by evaluating only parts of the game at each step, discovering good strategies over time. Recall that our model captures examples like (Pal and Vidal, 2020).
>
> In practical settings, circuits can be replaced with standard ML models (e.g., deep neural networks). This opens the door to algorithms that (1) determine whether a given task admits a defense, watermark, or transferable attack, and (2) produce an implementation accordingly. In summary, our theory shows that for a given task, one only needs to set up the appropriate loss and apply these iterative algorithms to extract the desired property. We believe this represents an exciting future direction at the intersection of large-scale algorithmic game theory and AI security.
>
>
>
>
> ---
>
> We sincerely appreciate the reviewer’s critical perspective and constructive remarks. We hope that the above clarifications and proposed revisions effectively address the concerns raised and contribute to improving the paper’s clarity.
>
> ---
>
> ### References:
>
> Lanctot, Marc, et al. "A unified game-theoretic approach to multiagent reinforcement learning." NeurIPS 2017.
>
> McAleer, Stephen, et al. "XDO: A double oracle algorithm for extensive-form games." NeurIPS 2021.
>
> Adam, Lukáš, et al. "Double oracle algorithm for computing equilibria in continuous games." AAAI 2021.
>
> Pal, Ambar, and René Vidal. "A game theoretic analysis of additive adversarial attacks and defenses." NeurIPS 2020.

---

> > ### Comment · Reviewer_je1d · 2025-08-07
> >
> > Thanks for the rebuttal.
> > I have re-read the paper along with the other reviews.
> > It seems that there's consensus on the strengths and weaknesses, with different weights on each of them.
> > The rebuttal addresses my concerns to some degree, but I still think practicality is an important issue that must be addressed by the paper.
> > If the other reviewers find the claims useful, I wouldn't argue against accepting this paper.
> > In addition, even with the new descriptions provided in the rebuttal, the explanations seem to be unfriendly to many readers in the community.
> > If the paper is to be accepted, I wish the authors would go through a thorough revision.

---

> > > ### Author Response · Authors · 2025-08-08
> > > **Thank you for the Update**
> > >
> > > We sincerely thank the reviewer for their careful reading and constructive suggestions to improve the paper. We appreciate the concerns regarding clarity and practicality, and in the final version, we will incorporate the relevant discussions from the rebuttal and expand the preliminaries in the appendix to reach a wider ML audience.

---

### Official Review · Reviewer_adhX · 2025-06-29

**Clarity:** 3
**Significance:** 3
**Originality:** 3
**Rating:** 4
**Confidence:** 3

**Summary:**

This paper presents a unified theoretical framework to study the trade-offs between backdoor-based watermarking, adversarial robustness, and a novel third axis: transferable attacks. By modeling the interaction between a verifier and a prover as a game, the authors prove that for any learning task, at least one of these mechanisms must exist. They formalize each mechanism, demonstrate examples of transferable attacks using fully homomorphic encryption, and link the existence of transferable attacks to the existence of cryptographic primitives (EFID pairs). Overall, this paper introduces transferable attacks into the trade-off between watermarks and defences, providing a new perspective compared to existing works.

**Questions:**

Question
1. The input space is defined as {0,1}^n, but most current classification tasks operate on continuous spaces (e.g., \mathbb{R}^n). Can this framework be extended to more realistic or complex input spaces?

2. Why are Definitions 3 and 4 considered complementary? Intuitively, it seems that adversarial defence and transferable attacks are complementary. Or perhaps the complementarity is between defence and the union of transferable attacks and watermarks?

3. In Definition 3, it is stated that there exists a B_n such that err(x,y)< 2\epsilon. What if there are two such B_n​, or three, or up to n−1different B_n satisfying this condition? Do these cases need to be analyzed within your framework?

4. Intuitively, if a model is adversarially robust, then adversarial examples (transferable attacks in this paper) should not exist. This seems to imply that for any model, either adversarial robustness holds or transferable attacks exist. In that case, why is watermarking needed? Could you clarify how watermarking fits into this framework and how it relates to the intuitive dichotomy between robustness and transferable attacks?

**Ethical Concerns:**

["NO or VERY MINOR ethics concerns only"]

**Quality:**

3

**Strengths And Weaknesses:**

- Strengths
1. The paper addresses a significant research problem by exploring the relationship between backdoor-based watermarking and adversarial robustness. It innovatively introduces the concept of transferable attacks as a novel perspective to analyze this relationship.

2. The Lines-on-Circle combined with Fully Homomorphic Encryption example provides the first rigorous demonstration that transferable attacks can succeed using substantially less computational power compared to what is required for defence.

- Weaknesses
1. The work is exclusively theoretical. Without empirical validation using real-world models or datasets, the practical implications and manifestations of the proposed trilemma in real deep learning systems remain unclear.

2. The analysis is conducted in the idealized Boolean-circuit world. It remains unclear how (or whether) the same argument can scale to high-dimensional, structured data such as images, text, or graphs.

3. The exemplar task relies on FHE, which is far removed from common computer-vision or NLP pipelines. It remains open whether similar transferable attacks can be instantiated without heavyweight cryptography.

4. The notation in lines 206 and 217 seems incomplete or unclear. It should likely be corrected to \mathbf{x} \in ({0,1}^{n})^q rather than \mathbf{x} \in {0,1}^{nq}.

---

> ### Author Rebuttal · Authors · 2025-07-31
>
> Thank you for your review and recognizing our contributions. We address your points below.
>
>
> ## Response to W1- Practical Implications
>
> 1. ### Existing Results
>
> We would like to clarify that trade-offs between backdoor-based attacks and adversarial defenses have been studied in several prior works, primarily through empirical analysis. These works generally observe that models made robust to certain adversarial attacks often become more vulnerable to backdoor-based attacks, highlighting the nuanced and complex nature of such trade-offs.
>
> For example, (Pal and Vidal, 2020) show (first theoretically) that Fast Gradient Methods (as adversarial attacks) and Randomized Smoothing (as defenses) can form a Nash equilibrium under a restricted additive noise model.
> Additionally, they provide *experiments* showing that their theoretical result does hold for some datasets (e.g., MNIST).
> The experiments in Pal and Vidal, 2020 consider a game with a slightly different utility from the one used by us.
> If the experiments were run for our game, then depending on the utility obtained at the equilibrium, our approach would extract one of the three schemes (defense, watermark, transferable attack).
> Our theoretical contributions can be seen as a generalization of this result to a broader class of attacks and defenses.
>
>
> 2. ### Computational Resources
>
> Our theorem also offers insight into the computational resources required to successfully construct a watermark or a defense. Specifically, if one aims to defend against adversaries with a computational budget of
> $T$ units (interpretable as compute time), our result suggests that by increasing one’s own resources to $T^2$
>  , it is likely to succeed—provided the learning task admits a defense. Conversely, if such an effort fails, it may indicate that the task permits transferable attacks, which in turn could preclude the existence of a watermark as well. This perspective could serve as a practical guideline for how to allocate resources before concluding that a task may not admit a defense. To our best knowledge, we provide one of the first quantitative bounds on the resources of attackers and defenders in very general settings (e.g., beyond bounded VC-dim etc.).
>
>
> ## Response to W2- Boolean Circuits
>
> At a fundamental level, families of Boolean circuits—as used in our work—are Turing complete, meaning they can simulate any algorithm that a Turing machine can. This makes them expressive enough to capture any computable algorithm and a natural abstraction for studying the inherent properties of learning tasks, independent of specific algorithms. One of our contributions is to show that, for every learning task, at least one of the following must exist: a watermark, an adversarial defense, or a transferable attack.
>
> While, as the reviewer notes, this may not have immediate practical utility, our existence result can guide practical efforts by informing where to search. A key part of our proof is the formulation of a zero-sum game between a "watermarking agent" and a "defense agent," where the game's value indicates which of the three properties exists. Moreover, an optimal strategy in this game corresponds to an actual implementation of a watermark, defense, or transferable attack.
>
> Though finding a Nash equilibrium in such a game seems computationally challenging—since the action spaces involve all circuits of a given size—recent iterative algorithms for large-scale games [Lanctot et al., 2017; McAleer et al., 2021; Adam et al., 2021] offer promising approaches. These methods work by evaluating only parts of the game at each step, discovering good strategies over time. Recall that our model captures examples like (Pal and Vidal, 2020).
>
> In practical settings, circuits can be replaced with standard ML models (e.g., deep neural networks). This opens the door to algorithms that (1) determine whether a given task admits a defense, watermark, or transferable attack, and (2) produce an implementation accordingly. In summary, our theory shows that for a given task, one only needs to set up the appropriate loss and apply these iterative algorithms to extract the desired property. We believe this represents an exciting future direction at the intersection of large-scale algorithmic game theory and AI security.
>
>  ## Response to W3 – Why the exemplar task uses FHE
>
> Section 6 proves that *transferable attacks* imply the existence of EFID pairs (Theorem 3). This shows that some computational assumptions are *necessary* for the existence of transferable attacks. It is possible that a weaker assumption than FHE is enough to construct transferable attacks. We left this question for future work.
>
> The key reason why some computational assumptions are necessary is the undetectability property of transferable attacks, i.e., the adversarial examples should be *indistinguishable* from normal samples. Standard arguments from computational complexity imply that if adversarial examples can be *efficiently* generated and are indistinguishable, then computational assumptions are necessary.
>
> ## Response to notation comment (lines 206 & 217)
>
> We thank the reviewer for pointing out this typo.
> But additionally, the two representations are equivalent:
>
> ({0,1}^n)^q = {0,1}^n × ... × {0,1}^n (q times) ≅ {0,1}^{nq}
>
> so the original `x ∈ {0,1}^{nq}` is correct.
> However, we agree that `({0,1}^n)^q` makes the “tuple of `q` inputs” interpretation clearer.
> We will therefore *adopt the reviewer’s notation* in lines 206, 217.
>
> ## Response to Q1 – Binary vs. continuous input spaces
>
> Thank you for raising this point, as it allows us to explain how our results apply more broadly.
>
> Any practical ML pipeline—whether it processes 256×256 RGB images, token sequences, or graphs—executes a finite sequence of arithmetic and logical operations. Bit-level compilation therefore yields a polynomial-size Boolean circuit (e.g., a ReLU network with `p` weights compiles to `O(p log q)` gates when each weight is quantised to `q` bits). Because our theorems depend only on the *ratio* of prover to verifier circuit sizes, the dimensionality or structure of the data does not change the trade-offs.
>
> To answer your question directly, our framework already encompasses the continuous input spaces.
>
> ## Response to Q2 – What we mean by “complementary”
>
> We use “complementary” in **two ways**:
>
> 1. **At the mechanism level:**
>    - A **Watermark** (Definition 3) allows the defender to prove authorship by planting backdoors that can later be triggered to cause specific errors, serving as evidence of ownership.
>    - A **defense** (Definition 4) helps the defender detect or reject adversarial manipulations.
>    These serve different purposes and can be applied together.
>
> 2. **At the outcome level:**
>    Theorem 1 shows that for any task, either the defender can enforce a defense or the attacker succeeds with a Watermark or a Transferable Attack (Definition 5).
>    So the true complement of “having a defense” is exactly what the reviewer noted: *the union of Watermark and Transferable Attack*.
>
>
> We hope this clarifies what we mean by “complementary” in this context.
>
> ## Response to Q3 – Multiplicity of $B_n$ in Definition 3
>
> Thank you for pointing this out.
>
> The existential clause $\exists$ $B_n$  such that $err(x, y) < 2 \epsilon$  is included solely to prevent *false positives*: it guarantees that a (non-watermarked) model trained from scratch $f_{Scratch}$ answers the watermark queries (containing the trigger) correctly. Meanwhile, the **Correctness** condition ($err \leq \epsilon$)  ensures we compare only models of similar utility to the watermarked one.
>
> If two, ten, or $n-1$ different models trained from scratch also satisfy this inequality, the condition is still met—the definition depends only on the *existence* of such models, not on the number of them.
>
> To answer your question more directly: yes, our framework is deliberately designed to capture and formalize this through the **Uniqueness** property in Definition 3, which ensures that the watermark does not accidentally generalize across independently trained models.
>
> We hope this clarifies the intent behind the definition.
>
>
> ## Response to Q4 – Clarifying the distinction between robustness, transferable attacks, and watermarking
>
> Thank you for this thoughtful question.
>
> The key subtlety lies in what happens when the model is not adversarial robustness. In that case, two distinct phenomena can occur:
>
> - A **Watermark** is a backdoor trigger that reliably causes *one specific model* to err, while remaining correctly handled by non-watermarked models. This is enforced by the **Uniqueness** property in Definition 3 and ensures that the trigger is model-specific (e.g., can be used for ownership verification).
>
> - A **Transferable Attack**, by contrast, causes *all models* (within some bounded computational class) to fail on the same input. It breaks robustness in a universal way and, unlike a watermark, is not tied to one specific model.
>
> Informally, a watermark trigger might unintentionally cause errors in all models, violating Uniqueness and becoming a transferable attack instead. In such a scenario, adversarial examples are inherently transferable, making proper watermarking impossible.
>
> Thus, while robustness and transferable attacks appear as a dichotomy, watermarking creates a third case: a model vulnerable only to specific, non-transferable triggers, enabling the coexistence of all three definitions in the trilemma.
>
> ---
>
> ### References:
>
> Lanctot, Marc, et al. "A unified game-theoretic approach to multiagent reinforcement learning." NeurIPS 2017.
>
> McAleer, Stephen, et al. "XDO: A double oracle algorithm for extensive-form games." NeurIPS 2021.
>
> Adam, Lukáš, et al. "Double oracle algorithm for computing equilibria in continuous games." AAAI 2021.
>
> Pal, Ambar, and René Vidal. "A game theoretic analysis of additive adversarial attacks and defenses." NeurIPS 2020.

---

### Official Review · Reviewer_f8Ls · 2025-07-03

**Clarity:** 2
**Significance:** 3
**Originality:** 3
**Rating:** 5
**Confidence:** 2

**Summary:**

This paper presents a unified framework for analyzing the trade-offs between backdoor-based watermarks, adversarial defenses, and transferable attacks in machine learning. The authors formalize these concepts as interactive protocols between two players, modeled as families of circuits to account for their computational resources. The key contribution is a theorem stating that for any learning task, at least one of these three properties must exist. This result is proven by formulating the interaction as a zero-sum game between the players and analyzing its Nash equilibrium.

The paper also establishes a connection between transferable attacks and cryptographic primitives, specifically showing that the existence of transferable attacks implies the existence of EFID pairs of distributions. This connection suggests that tasks admitting transferable attacks must be computationally complex. Additionally, the paper provides concrete examples of learning tasks that support watermarks and adversarial defenses, particularly for hypothesis classes with bounded VC dimension. These examples demonstrate how different security properties can exist or coexist depending on the characteristics of the learning task and the available computational resources.

**Questions:**

The paper presents a strong theoretical framework, but the practical implications of the results are not thoroughly discussed. Could the authors provide more guidance on how practitioners in machine learning security should interpret and apply these results? Specifically: a) How might the trade-offs identified in the paper influence the design of secure machine learning systems in practice? b) Can the authors provide examples or scenarios where their framework could inform decision-making in real-world ML security situations?

**Ethical Concerns:**

["NO or VERY MINOR ethics concerns only"]

**Final Justification:**

After considering the authors' response and discussions, I maintain my recommendation in favor of acceptance.

My primary concern prior to the discussion phase was the paper's clarity and accessibility. I suggested better connecting the work to more practical research in ML security. The authors' response to these concerns was convincing. They've outlined plans to improve connections to established concepts in backdoor watermarks, adversarial defences, and transferable attacks. While the work remains primarily theoretical, they've made a commendable effort to discuss its practical implications, which should enhance its relevance for a more general ML audience.

The paper's technical depth and the significance of its main theorem remain its core strengths. These aspects, combined with the authors' commitment to improving accessibility, support its acceptance in my view.

**Limitations:**

The authors have addressed limitations by acknowledging their focus on classification tasks. However, two additional limitations could be discussed:

- The implications of modeling Alice and Bob as circuits and whether this accurately represents real-world adversaries and defenders. Are there scenarios where this model might break down?

- The paper introduces new definitions for watermarks, adversarial defenses, and transferable attacks. It would be beneficial to discuss potential limitations of these definitions, especially in relation to “working definitions” used in the field.

**Paper Formatting Concerns:**

None identified

**Quality:**

4

**Strengths And Weaknesses:**

**Strengths**

S1. The paper demonstrates technical depth by employing diverse theoretical tools. It leverages game theory (zero-sum games and Nash equilibria) to prove its main theorem, uses cryptographic concepts (EFID pairs) to analyze transferable attacks, applies computational complexity theory to model resource-bounded adversaries, and connects to learning theory (VC-dimension) in analyzing watermarks and defenses.

S2. The main theorem, stating that at least one of the three properties (watermark, adversarial defense, or transferable attack) must exist for any learning task, appears to be a significant contribution. It provides fundamental insights into the nature of machine learning security and the inherent trade-offs involved.

**Weaknesses**

W1. The definitions in Section 4 don't clearly connect to established notions of watermarking and adversarial defense, nor do they fully justify the various properties introduced. This makes it harder for readers to relate the new framework to existing knowledge in the field.

W2. The paper could benefit from more discussion on the practical implications of its findings for real-world machine learning systems and security practices.

W3. While not a major weakness, the paper focuses solely on classification tasks. The authors acknowledge this limitation and discuss potential generalizations to generative modeling tasks in Appendix E.

W4. The paper could benefit from a discussion of whether there are any real-world scenarios that aren't captured by the theory. Additionally, the authors could justify more thoroughly their choice to model Alice and Bob as circuits and discuss potential limitations of this approach.

**Minor**

- Incorrect cite style

- Fig 1: the caption refers to “adversaries” and “defenders”, but it’s not clear whether this refers to Alice or Bob. (I believe Alice corresponds to the adversary in all cases).

---

> ### Author Rebuttal · Authors · 2025-07-31
>
> # Response to Reviewer f8Ls
>
> Thank you for your thorough review and for acknowledging the contributions of our work. Let us address your concerns and questions one by one.
>
> ---
>
> ## Response to W1 – Clarifying the link between our Section 4 definitions and established notions.
>
> Below we map each element of our Section 4 framework to the corresponding, well‑studied concepts in the watermarking and adversarial‑defense literature and explain why every property is necessary.
>
> ---
>
> ## 1. Backdoor–based watermarks (Definition 3)
>
> | **Property in Def. 3** | **Classical analogue** | **Why it is needed?** |
> |------------------------|------------------------|------------------------|
> | Correctness | Standard accuracy requirement, e.g. *Adi et al., 2018* | Ensures watermarking does not degrade task performance. |
> | Uniqueness | *Verifiability* in black‑box watermarking | Prevents false positives on independently‑trained models. |
> | Robustness‑to‑Removal (“Unremovability”) | *Robustness* to pruning / fine‑tuning (*Namba and Sakuma, 2019*) | Captures the usual “cannot be scrubbed” criterion. |
> | Undetectability | *Stealth* requirement (*Le Merrer et al., 2017*) | Guarantees watermark triggers look like in‑distribution data. |
>
>
> ---
>
> ## 2. Adversarial defenses (Definition 4)
>
> | **Property in Def. 4** | **Classical analogue** | **Why it is needed?** |
> |------------------------|------------------------|------------------------|
> | Correctness | Baseline test‑error requirement in certified / detection‑based defences | Ensures the defended model remains useful. |
> | Completeness | “No false positive” guarantee in detection frameworks (*Goldwasser et al., 2020*) | Prevents trivial defences that reject everything. |
> | Soundness | Detection+robustness guarantee (rejection‑based defenses) | Formalises that attacks must both fool and stay indistinguishable. |
>
>
> ---
>
> ## 3. Transferable attacks (Definition 5)
>
> | **Property in Def. 5** | **Classical analogue** | **Why it is needed?** |
> |------------------------|------------------------|------------------------|
> | Correctness | Baseline learnability precondition | Ensures a meaningful low‑error model exists for the attacker to exploit, mirroring the “low‑error target” assumption in adversarial‑example studies. |
> | Transferability | Cross‑model adversarial transfer (*Tramèr et al., 2017*) | Captures worst‑case attacks that succeed regardless of defender architecture or training procedure. |
> | Undetectability | Stealth / indistinguishability (*Goldwasser et al., 2020*) | Guarantees defenders cannot filter the adversarial queries, aligning with cryptographic indistinguishability in EFID pairs. |
>
>
> ---
>
> ## 4. Why the explicit interactive protocol?
>
> - **Bridging literatures:**
>   Framing watermarks and defenses as verifier–prover games lets us use proof techniques from zero‑sum games and interactive proofs (*Goldwasser et al., 1986*), which is crucial for Theorem 1.
>
> - **Resource‑aware analysis:**
>   Circuit‑size parameters $(S_A, S_B)$ capture fine‑grained trade‑offs (e.g. $S_A \approx \sqrt{S_B}$ in Theorem 1) that earlier works omit.
>
> ---
>
> ## 5. Planned text changes to improve clarity
>
> 1. Add a bridging paragraph at the start of Sec. 4 explicitly mapping each property to its classical analogue (see tables above).
>
> ---
>
> We hope this clarifies how our formalisation extends, rather than deviates from, established definitions while retaining the new computational perspective.
>
> ## Response to W2/Q1 – Practical implications for real‑world ML security
>
> 1. ### Existing Results
>
> We would like to clarify that trade-offs between backdoor-based attacks and adversarial defenses have been studied in several prior works, primarily through empirical analysis. These works generally observe that models made robust to certain adversarial attacks often become more vulnerable to backdoor-based attacks, highlighting the nuanced and complex nature of such trade-offs.
>
> For example, (Pal and Vidal, 2020) show (first theoretically) that Fast Gradient Methods (as adversarial attacks) and Randomized Smoothing (as defenses) can form a Nash equilibrium under a restricted additive noise model.
> Additionally, they provide *experiments* showing that their theoretical result does hold for some datasets (e.g., MNIST).
> The experiments in Pal and Vidal, 2020 consider a game with a slightly different utility from the one used by us.
> If the experiments were run for our game, then depending on the utility obtained at the equilibrium, our approach would extract one of the three schemes (defense, watermark, transferable attack).
> Our theoretical contributions can be seen as a generalization of this result to a broader class of attacks and defenses.
>
>
> 2. ### Computational Resources
>
> Our theorem also offers insight into the computational resources required to successfully construct a watermark or a defense. Specifically, if one aims to defend against adversaries with a computational budget of
> $T$ units (interpretable as compute time), our result suggests that by increasing one’s own resources to $T^2$
>  , it is likely to succeed—provided the learning task admits a defense. Conversely, if such an effort fails, it may indicate that the task permits transferable attacks, which in turn could preclude the existence of a watermark as well. This perspective could serve as a practical guideline for how to allocate resources before concluding that a task may not admit a defense. To our best knowledge, we provide one of the first quantitive bounds on the resources of attackers and defenders in very general settings (e.g., beyond bounded VC-dim etc.).
>
> ## Response to W3 – Extending beyond classification
>
> Thank you for raising this point. While our main results are phrased for binary classification, the framework naturally extends to generative models as well.
>
> We already reference how our definitions apply to backdoor-based watermarking in generative models such as pre-trained language models (PLMs) in *Related Work* (Gu et al., 2022; Peng et al., 2023; Li et al., 2023; lines 98–102, 985–992), and outline how to adapt the framework to non-classification tasks in Appendix E.
>
> To make the presentation clearer, we will revise the *Related Work* section to distinguish between:
> - backdoor-based watermarks in generative models that can be framed as classification tasks (e.g., PLMs), and
> - watermarking settings that omit classification entirely.
>
> This addition highlights that our framework partly accommodates state-of-the-art generative systems without modifying the underlying formalism.
>
>
>
> ## Response to W4/L1 – Circuits and their Limitations
>
>  At a fundamental level, families of Boolean circuits—as used in our work—are Turing complete, meaning they can simulate any algorithm that a Turing machine can. This makes them expressive enough to capture any computable algorithm and a natural abstraction for studying the inherent properties of learning tasks, independent of specific algorithms. One of our contributions is to show that, for every learning task, at least one of the following must exist: a watermark, an adversarial defense, or a transferable attack.
>
> While, as the reviewer notes, this may not have immediate practical utility, our existence result can guide practical efforts by informing where to search. A key part of our proof is the formulation of a zero-sum game between a "watermarking agent" and a "defense agent," where the game's value indicates which of the three properties exists. Moreover, an optimal strategy in this game corresponds to an actual implementation of a watermark, defense, or transferable attack.
>
> Though finding a Nash equilibrium in such a game seems computationally challenging—since the action spaces involve all circuits of a given size—recent iterative algorithms for large-scale games [Lanctot et al., 2017; McAleer et al., 2021; Adam et al., 2021] offer promising approaches. These methods work by evaluating only parts of the game at each step, discovering good strategies over time. Recall that our model captures examples like (Pal and Vidal, 2020).
>
> In practical settings, circuits can be replaced with standard ML models (e.g., deep neural networks). This opens the door to algorithms that (1) determine whether a given task admits a defense, watermark, or transferable attack, and (2) produce an implementation accordingly. In summary, our theory shows that for a given task, one only needs to set up the appropriate loss and apply these iterative algorithms to extract the desired property. We believe this represents an exciting future direction at the intersection of large-scale algorithmic game theory and AI security.
>
> ## Response to minor comment – Fig. 1 caption
>
> Thank you for catching the ambiguity.
> We will edit the caption to read (new text in *italics*):
>
> > **Figure 1**:  [...]
> > *Throughout this figure, Alice (the verifier) plays the role of the adversary, while Bob (the prover) plays the role of the defender.*
>
>
> ---
>
> We sincerely thank the reviewer for their critical perspective and hope that these clarifications and planned revisions resolve the concerns raised and improve the paper’s accessibility and impact.
>
> ---
>
> ### References:
>
> Lanctot, Marc, et al. "A unified game-theoretic approach to multiagent reinforcement learning." NeurIPS 2017.
>
> McAleer, Stephen, et al. "XDO: A double oracle algorithm for extensive-form games." NeurIPS 2021.
>
> Adam, Lukáš, et al. "Double oracle algorithm for computing equilibria in continuous games." AAAI 2021.
>
> Pal, Ambar, and René Vidal. "A game theoretic analysis of additive adversarial attacks and defenses." NeurIPS 2020.

---

> > ### Comment · Reviewer_f8Ls · 2025-08-04
> >
> > Thank you for your detailed and thoughtful response. I appreciate the clarifications you've provided, particularly regarding points W1, 3, and 4. I encourage you to integrate these responses into the main body of the paper. I remain in favor of acceptance and maintain my current score.

---

> > > ### Author Response · Authors · 2025-08-08
> > > **Thank you for the Update**
> > >
> > > Thank you for providing useful suggestions to improve the clarity of the paper. For the final version, we will add the discussions related to points W1,3 and 4 as requested by the reviewer.

---

### Note · Authors · 2025-08-16

We thank the reviewers for their efforts in providing thoughtful feedback.

Overall, they appreciated the technical depth of our analysis, which integrates zero-sum games, cryptography, computational learning theory, and circuit complexity to meta-analyze backdoor-based watermarks, adversarial defenses, and transferable attacks, and to quantify the necessary resource bounds. Moreover, they liked the construction of the cryptography-based transferable attack, in which the attacker can succeed with fewer resources than the defender.

However, the reviewers raised concerns about the definitions, the main takeaways, the implications of our results, and practicality. Below, we summarize the key points of our rebuttal:

**Definitions**: We clarified how our definitions relate to existing, equivalent notions (see the table in the rebuttal for Reviewer f8Ls).

**Clarity and key takeaways**: Our main result is a trichotomy: every learning task admits at least one of (i) a defense, (ii) a watermark, or (iii) a transferable attack. Theorem 2 exhibits tasks where robustness and verifiability cannot both hold (via transferable attacks). Conversely, for structured families (e.g., bounded VC-dimension) we show that robustness and verifiability can coexist (App. I–J, Lemmas 3–4), in which case transferable attacks do not arise. We will make this clearer and expand the preliminaries for a broader ML audience.

**Practical guidance**: We state a compute rule of thumb for _any_ learning task: defending against an attacker with budget $T$ may require approximately $T^2$
 defender resources when a defense exists; failure at that scale could be evidence for transferable attacks (and may preclude watermarking). We will make this guidance explicit and add pointers to related work (e.g., Pal and Vidal, 2020).

**From theory to practice**: We cast a zero-sum game between watermarking and defense; the game’s value identifies which property holds, and equilibrium strategies instantiate it. Despite large action spaces, recent algorithmic developments for large-scale games (e.g., regret-based/PSRO) could help make this practical with standard ML models. We will add a detailed discussion of this.

We appreciate the reviewers’ insights and will incorporate these clarifications and edits in the final version.

---

### Decision · Program_Chairs · 2025-09-17

**Decision:**

Accept (poster)

**Comment:**

The paper studies trade-offs between three basic problems in ML in the context of classification tasks: (1) back-door based watermarks (for which limited modifications still keep the secret), (2) adversarial attacks/defenses (in which adversary does limited modification and hopes to evade), and (3) transferrable attacks (which are attacks that work against all defenses).

The paper proves a nice trichotomy theorem, basically showing that for any classification task, at least one of the 3 above will always happen. The proof is by first modeling a zero-sum game between two players, one wants to watermark and one wants to get an adversarial attack. They then show that either of them wins or else transferable attacks emerge. Of course the devil is in the details of how each of these three tasks are formalized. So, the definitions do not fully match with the main-stream variants of these tasks, but the definitions “make sense” as they are natural on their own. So, the final result is a very interesting theoretical trichotomy.

The paper uses tools across different fields of game theory, cryptography, and complexity theory. The reviewers find the theoretical contribution significant, though there was no agreement about the practical implications. Despite that, the theoretical contribution is major enough to attract interest from the NeurIPS audience.